# Bosonic and fermionic Gaussian states from Kähler structures

**Lucas Hackl[1,2]⋆ and Eugenio Bianchi[3,4]†**

**1** School of Physics, The University of Melbourne, Parkville, VIC 3010, Australia
**2** QMATH, Department of Mathematical Sciences, University of Copenhagen,
Universitetsparken 5, 2100 Copenhagen, Denmark
**3** Department of Physics, The Pennsylvania State University, University Park, PA 16802, USA
**4** Institute for Gravitation and the Cosmos, The Pennsylvania State University,
University Park, PA 16802, USA

⋆ lucas.hackl@unimelb.edu.au, † ebianchi@psu.edu

## Abstract

We show that bosonic and fermionic Gaussian states (also known as "squeezed coherent states") can be uniquely characterized by their linear complex structure $J$ which is a linear map on the classical phase space. This extends conventional Gaussian methods based on covariance matrices and provides a unified framework to treat bosons and fermions simultaneously. Pure Gaussian states can be identified with the triple $(G, \Omega, J)$ of compatible Kähler structures, consisting of a positive definite metric $G$, a symplectic form $\Omega$ and a linear complex structure $J$ with $J^2 = -\mathbb{1}$. Mixed Gaussian states can also be identified with such a triple, but with $J^2 \neq -\mathbb{1}$. We apply these methods to show how computations involving Gaussian states can be reduced to algebraic operations of these objects, leading to many known and some unknown identities. We apply these methods to the study of (A) entanglement and complexity, (B) dynamics of stable systems, (C) dynamics of driven systems. From this, we compile a comprehensive list of mathematical structures and formulas to compare bosonic and fermionic Gaussian states side-by-side.

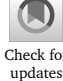

# 1 Introduction

Gaussian states play a distinguished role in quantum theory: they appear under various names (squeezed coherent states, squeezed vacua, quasi-free states, generalized Slater determinants, ground states of free Hamiltonians) and are used in vastly different research fields, from quantum information [1–3] to quantum field theory in curved spacetimes [4,5]. They are often used as testing ground, as many concepts can be studied analytically when they are applied to Gaussian states (*e.g.*, entanglement entropy, logarithmic negativity, circuit complexity). They also form the basis of many numerical or perturbative methods to study physical systems approximately (*e.g.*, Feynman diagrams, Bardeen–Cooper–Schrieffer theory, Hartree-Fock method, Bogoliubov theory) [6].

Gaussian states are completely characterized by the 2-point correlation functions of linear observables, from which all higher $n$-point functions can be computed via the famous Wick's theorem. For a system of $N$ bosonic degrees of freedom, one can choose coordinates $q_i, p_i$ in phase space and express linear observables as $\hat{\xi}^a \equiv (\hat{q}_1, \cdots, \hat{q}_N, \hat{p}_1, \cdots, \hat{p}_N)$. Similarly, for a system with $N$ fermionic degrees of freedom, one can choose Majorana modes $q_i, p_i$ (often denoted $\gamma_i, \eta_i$) at the classical level, and express linear observables again in a compact form as $\hat{\xi}^a$. In both cases, given a state $|\psi\rangle$, the correlation function takes the form[1]

$$\langle\psi|\hat{\xi}^a\hat{\xi}^b|\psi\rangle = \frac{1}{2}\left(G^{ab} + i\Omega^{ab}\right). \tag{1}$$

The symmetric and the antisymmetric part of the correlation function define two mathematical structures: a *positive definite metric* $G^{ab}$ and a *symplectic form* $\Omega^{ab}$. For bosons, the symplectic form $\Omega^{ab}$ is canonical and determined by the classical Poisson brackets, while the metric $G^{ab}$ depends on the state. On the other hand, for fermions it is the metric $G^{ab}$ that is canonical and fixed already at the classical level, while the symplectic form $\Omega^{ab}$ characterizes the quantum state. Remarkably, for a pure Gaussian state $|\psi\rangle$, the two structures $G^{ab}$ and $\Omega^{ab}$ satisfy a compatibility condition that defines a *Kähler structure* $(G, \Omega, J)$, (see figure 1). The third object $J^a{}_b$ is a complex structure and defines a notion of creation and annihilation operators. As in [8], we show how properties of Gaussian states for bosonic and for fermionic systems can be described in a compact unified way in terms of Kähler structures[2], tailoring the language and selecting aspects relevant for applications in quantum information and out-equilibrium quantum systems. We highlight the fact that various expressions for information-theoretic quantities (*e.g.*, the entanglement entropy) take the same form for bosons and fermions when written in terms of the complex structure $J$.

Due to their versatility, many properties of Gaussian states have been independently discovered in different research communities ranging from quantum optics and condensed matter physics to high energy theory and quantum gravity. Historically, Gaussian states were mostly used as a calculational tool which is only described indirectly in terms of correlations, Bogoliubov transformations, creation and annihilation operators or free Hamiltonians, but more recently they were recognized in quantum information theory and condensed matter theory as important families of pure and mixed quantum states with distinct properties. Consequently, there exist many reviews [1–3, 7–9] focusing on specific applications of Gaussian states relevant for these research fields. Kähler structure were first used to describe Gaussian states in the context of quantum fields in curved spacetimes [10–15].

---

[1] Here, we assume $\langle\psi|\hat{\xi}^a|\psi\rangle = 0$, but also treat the general case in section 3.

[2] In mathematics, one usually says that a manifold or a vector space has *a* Kähler structure if it is equipped with compatible objects $(G, \Omega, J)$. In this manuscript, we will refer to these objects $G$, $\Omega$ and $J$ commonly as Kähler structures, too.

The goal of the present manuscript is the systematic application of Kähler structures to (A) entanglement and circuit complexity, (B) dynamics of stable quantum systems and (C) dynamics of driven quantum systems, while previous work on Kähler structures for Gaussian states [7–9] has largely focused on quantization and the definition of free quantum fields. We further emphasize how Kähler structures can be used to unify the description of bosonic and fermionic systems by providing explicit formulas for both systems. We carefully distinguish the involved geometric structures (metric, symplectic form and complex structures) and treat them as independent of their matrix representation in a given basis. This allows us to re-derive existing results (such as the inner product between Gaussian states) in a simplified and often basis-independent way, but also enabled us to find some—to our knowledge—new formulas (such as some of the covariant Baker-Campbell-Hausdorff relations to combine squeezing and displacement). Several results and techniques of this manuscript have been implicitly used in some of our earlier works [16–25] and we are confident that other researchers can benefit from a thorough exposition of Gaussian states from Kähler structures.

*The content of this manuscript is complemented by two other recent papers that adopt the same formalism: First, the geometry of variational methods is studied in [24], where Gaussian states appear as a prime example of so-called Kähler manifolds, whose geometric properties are closely related to the Kähler structures on the classical phase space. Second, the geometry of pure Gaussian state manifolds is used in [25] to find local extrema of differentiable functions on these manifolds, for which a large number of different parametrizations is reviewed. The present manuscript provides a self-contained, yet rigorous derivation of many results that are used in the other two papers.*

This manuscript is structured as follows: In section 2, we review the classical theory of bosonic and fermionic phase spaces, discuss the properties of Kähler structures, and review the quantization of the bosonic and fermionic theories. In section 3, we define Gaussian states in a unified framework based on Kähler structures and we use this framework to derive compact formulas for properties of Gaussian state for bosons and fermions. In section 4 we discuss applications to entanglement and dynamics. Each of the previous sections contains a large summary table (namely, table 1, 4 and 5) that summarizes the main results and compares bosons and fermions side-by-side. We conclude in section 5 by summarizing our results and discussing further applications of our formalism.

## 2 Bosons and fermions from Kähler structures

We review the quantization of bosonic and fermionic quantum systems based on Kähler structures. We present this material in a condensed way with a unified notation in mind, which is particularly suitable for later applications in physics. More detailed treatments of this material can be found in the mathematical physics literature [7–9, 26].

### 2.1 Classical theory

Quantization can be understood as a deformation of the algebra of observables on the classical phase space. We therefore begin by constructing the respective classical theories.

### 2.1.1 Phase space

For both, bosonic and fermionic systems, we begin our construction with a classical phase space $V \simeq \mathbb{R}^{2N}$ given by a $2N$-dimensional real vector space for systems with $N$ degrees of freedom. We denote a phase space vector by $v^a$. We have a canonical notion of linear observables given by linear forms $w_a$ in the dual phase space $V^*$. More generally, we use upper Latin indices to denote phase space vectors and lower ones to denote linear forms, *i.e.*, dual vectors. Please see appendix A.1 for a brief review of this formalism inspired by Einstein's summation convention and Penrose's abstract index notation.

So far, we have not assumed any additional structure on phase space or its dual.

**Definition 1.** *A $2N$-dimensional real vector space $V$ is*

- *a **bosonic phase space** if it is equipped with a **symplectic form** $\omega_{ab}$, i.e., an antisymmetric and non-degenerate bilinear form, or*

- *a **fermionic phase space** if it is equipped with a **metric** $g_{ab}$, i.e., a symmetric positive-definite bilinear form.*

*We further define their inverses as the dual symplectic form $\Omega^{ab}$ and the dual metric $G^{ab}$, which are uniquely determined by the conditions*

$$G^{ac}g_{cb} = \delta^a{}_b \quad and \quad \Omega^{ac}\omega_{cb} = \delta^a{}_b. \tag{2}$$

### 2.1.2 Linear observables

The key difference in the definition of bosonic and fermionic systems lies in the different background structure that we equip the space of linear observables with. For bosons, we equip $V^*$ with the symplectic form $\Omega^{ab}$ and $V$ with the dual structure $\omega_{ab}$ satisfying $\Omega^{ac}\omega_{cb} = \delta^a{}_b$. For fermions, we choose a positive metric $G^{ab}$ on $V$ instead, which comes with the dual metric $g_{ab}$ satisfying $G^{ac}g_{cb} = \delta^a{}_b$. In both cases, our choice provides a natural isomorphism between $V$ and $V^*$, *i.e.*, we can identify the vector $v^a \in V$ either with the dual vector $w_a = \omega_{ab}v^b$ for bosons or $w_a = g_{ab}v^b$ for fermions.

### 2.1.3 Algebra of classical observables

General observables form an associative algebra $\mathcal{A}$ with identity that is generated by $V^*$. For bosons, we require that this algebra is symmetric (commutative) leading to the unique symmetric algebra $\text{Sym}(V^*)$.[3] For fermions, we require that the algebra is antisymmetric (anticommutative) leading to the unique Grassmann algebra $\text{Grass}(V^*)$. While the bosonic algebra is infinite dimensional, as we can take arbitrary powers of linear observables, we have that the fermionic algebra is finite dimensional, $\dim \text{Grass}(V^*) = 2^N$ due to anti-symmetry.

A general classical observable can be written as a power series of the form

$$f(\xi) = f_0 + (f_1)_a \xi^a + (f_2)_{ab} \xi^a \xi^b + \cdots, \tag{3}$$

where for bosons only the completely symmetric part and for fermions only the completely antisymmetric part of $(f_n)_{a_1 \ldots a_n}$ will matter, as powers of $\xi^a$ are symmetric or anti-symmetric, respectively, in the algebra.

---

[3]Technically, we then complete this algebra to allow for infinite power series leading to the space of smooth phase space functions that physicists usually use to describe observables. Such considerations are not necessary for fermions, where the Grassmann algebra stays finite dimensional for finitely many degrees of freedom.

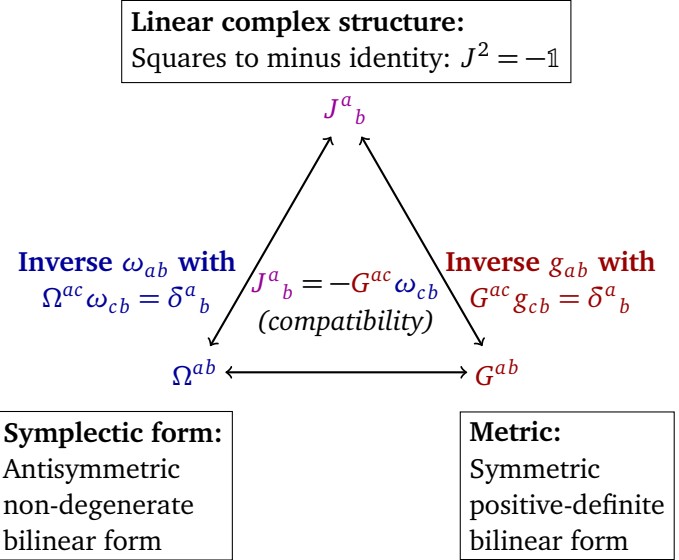

Figure 1: *Triangle of Kähler structures.* This sketch was reproduced from [24] and illustrates the triangle of Kähler structures, consisting of a symplectic form $\Omega$, a positive definite metric $G$ and a linear complex structure $J$. We also define the inverse symplectic form $\omega$ and the inverse metric $g$.

The background structures $\Omega$ and $G$ equip the algebra of observables with the additional structure of a Poisson bracket. This operation $\{\cdot,\cdot\}_{\pm} : \mathcal{A} \times \mathcal{A} \to \mathcal{A}$ satisfies

$$\{f,g\}_- = (\partial_a f)(\partial_b g)\Omega^{ab} \qquad \textbf{(bosons)}, \tag{4}$$

$$\{f,g\}_+ = (\partial_a f)(\partial_b g)G^{ab} \qquad \textbf{(fermions)} \tag{5}$$

2 with $\partial_a = \frac{\partial}{\partial \xi^a}$. Here, $\{\cdot,\cdot\}_-$ are the regular Poisson brackets known from classical mechanics, while $\{\cdot,\cdot\}_+$ are their analogue for fermions.

## 2.2 Kähler structures

We will now review the underlying mathematical structures that provide a unified description of bosonic and fermionic Gaussian states. It is based on the notion of Kähler structures and in particular on a so-called linear complex structure. These structures are well-studied in the context of Kähler manifolds, but for our purposes it suffices to study them on a single linear space, namely the classical phase space $V$ of the bosonic or fermionic theory. Gaussian states were, to our knowledge, first parametrized by linear complex structures in the context of quantum fields in curved spacetime [4, 10]. Here, linear complex structures naturally arise to distinguish unitarily inequivalent representations of the observable algebra. The role of linear complex structures has also been recognized in the mathematical physics literature on quantization [9] and to some extent in the field of quantum information [27].

**Definition 2.** *A real vector space is called Kähler space if it is equipped with the following three structures*

- **Metric**[4] $G^{ab}$, *which is symmetric and positive-definite with inverse* $g_{ab}$, *such that* $G^{ac}g_{cb} = \delta^a{}_b$,

---

[4]Here, "metric" refers to a metric tensor, *i.e.*, an inner product on a vector space. It should not be confused with the notion of metric spaces in analysis and topology.

- **Symplectic form** $\Omega^{ab}$, which is antisymmetric and non-degenerate[5] with inverse $\omega_{ab}$, such that $\Omega^{ac}\omega_{cb} = \delta^a{}_b$,

- **Complex structure** $J^a{}_b$, which satisfies $J^a{}_c J^c{}_b = -\delta^a{}_b$,

*such that they are related via the compatibility equations*

$$J^a{}_b = -G^{ac}\omega_{cb} \quad \Longleftrightarrow \quad J^a{}_b = \Omega^{ac}g_{cb}\,. \tag{6}$$

*We refer to $(G, \Omega, J)$ as* **Kähler structures**.

Note that (6) together with the requirements on metric, symplectic form and complex structure also implies

$$J^a{}_c\Omega^{cd}(J^{\mathsf{T}})_d{}^b = \Omega^{ab} \quad \text{and} \quad J^a{}_c G^{cd}(J^{\mathsf{T}})_d{}^b = G^{ab}\,, \tag{7}$$

which is often required separately. In practice, it is therefore sufficient to choose any two out of the three Kähler structures, solve equation (6) for the third and then require that this third Kähler structure satisfies the respective conditions.

### 2.2.1 Groups and algebras

Each structure defines a subgroup of the real general linear group $\mathrm{GL}(2N, \mathbb{R})$ of invertible linear maps $M^a{}_b$ on $V$ that preserves this specific structure. We have the orthogonal, the symplectic and the general linear groups given by

$$\begin{aligned}
\mathrm{O}(2N) &= \left\{ M \in \mathrm{GL}(2N, \mathbb{R}) \,\middle|\, MGM^{\mathsf{T}} = G \right\}, \\
\mathrm{Sp}(2N, \mathbb{R}) &= \left\{ M \in \mathrm{GL}(2N, \mathbb{R}) \,\middle|\, M\Omega M^{\mathsf{T}} = \Omega \right\}, \\
\mathrm{GL}(N, \mathbb{C}) &= \left\{ M \in \mathrm{GL}(2N, \mathbb{R}) \,\middle|\, MJ = JM \right\},
\end{aligned} \tag{8}$$

respectively, where $(M^{\mathsf{T}})_d{}^b = M^b{}_d$ and $(MGM^{\mathsf{T}})^{ab} = M^a{}_c G^{cd}(M^{\mathsf{T}})_d{}^b$ as explained in appendix A.1. Note that each subgroup depends on the respective Kähler structure, *i.e.*, $G$, $\Omega$ and $J$, respectively. Provided that two structures are compatible, the respective groups will intersect in a new subgroup isomorphic to $\mathrm{U}(N)$. We recall that compatibility between two Kähler structures is equivalent to requiring that the third structure defined via equation (6) satisfies the respective properties. Consequently, the subgroup associated to the third structure will necessarily intersect with the other structures exactly where those structures already overlap. This is known as the 2-out-of-3 property, because any two compatible Kähler structures already define the third. We visualize this in figure 2.

We can also represent the Lie algebras as linear maps on the classical phase space. They are the orthogonal, the symplectic and the general linear algebra given by

$$\mathfrak{so}(2N) = \{ K \in \mathfrak{gl}(2N, \mathbb{R}) \,|\, KG + GK^{\mathsf{T}} = 0 \}, \tag{9}$$

$$\mathfrak{sp}(2N) = \{ K \in \mathfrak{gl}(2N, \mathbb{R}) \,|\, K\Omega + \Omega K^{\mathsf{T}} = 0 \}, \tag{10}$$

$$\mathfrak{gl}(N, \mathbb{C}) = \{ K \in \mathfrak{gl}(2N, \mathbb{R}) \,|\, KJ = JK \}, \tag{11}$$

respectively, where we used that the Lie algebra of $\mathrm{O}(2N)$ is the same as for $\mathrm{SO}(2N)$. There is an important isomorphism that identifies the symplectic and the orthogonal algebra with

---

[5] A bilinear form $b_{ab}$ is called non-degenerate, if it is invertible. For this, we can check $\det(b) \neq 0$ in any basis of our choice.

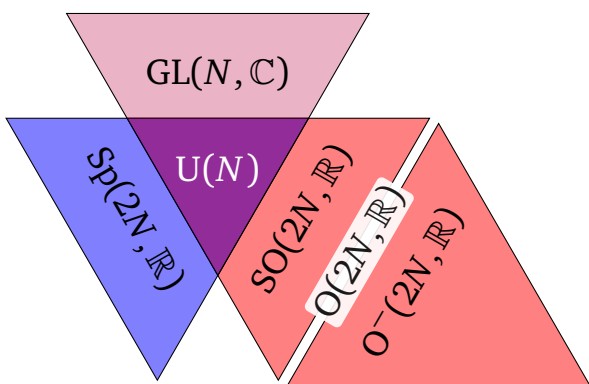

Figure 2: *Illustration of 2-out-of-3 property*. We show how the three groups $\mathrm{O}(2N,\mathbb{R})$, $\mathrm{Sp}(2N,\mathbb{R})$ and $\mathrm{GL}(N,\mathbb{C})$ intersect to form the unitary group $\mathrm{U}(N)$. In particular, we see that intersecting all three groups is equivalent to intersecting any two out of the three groups. Moreover, we see that only the component $\mathrm{SO}(2N,\mathbb{R}) \subset \mathrm{O}(2N,\mathbb{R})$ connected to the identity matters, while $\mathrm{O}^-(2N,\mathbb{R}) \subset \mathrm{O}(2N,\mathbb{R})$ is the subset (not subgroup) of group elements not connected to the identity.

symmetric and antisymmetric forms on $V$, respectively. More precisely, we can identify the Lie algebra element $K^a{}_b$ with the bilinear form $h_{ab}$ via

$$
\begin{aligned}
K^a{}_b &= \Omega^{ac} h_{cb} \iff h_{ab} = \omega_{ac} K^c{}_b, \quad \textbf{(bosons)} \\
K^a{}_b &= G^{ac} h_{cb} \iff h_{ab} = g_{ac} K^c{}_b. \quad \textbf{(fermions)}
\end{aligned}
\tag{12}
$$

The conditions (9) and (10) imply $h_{ab} = h_{ba}$ for bosons (symplectic Lie algebra) and $h_{ab} = -h_{ba}$ for fermions (orthogonal Lie algebra).

In the context of bosonic or fermionic Gaussian states, we are given either a symplectic form $\Omega^{ab}$ for bosons or a metric $G^{ab}$ for fermions, so that the respective groups and algebras will play a special role as being there without defining further structures. We will therefore refer to them simply as $\mathcal{G}$ and $\mathfrak{g}$ given by

$$
\begin{aligned}
\mathcal{G} &= \mathrm{Sp}(2N,\mathbb{R}), \quad \mathfrak{g} = \mathfrak{sp}(2N,\mathbb{R}), \quad \textbf{(bosons)} \\
\mathcal{G} &= \mathrm{O}(2N,\mathbb{R}), \quad \mathfrak{g} = \mathfrak{so}(2N,\mathbb{R}), \quad \textbf{(fermions)}
\end{aligned}
\tag{13}
$$

The symplectic and orthogonal Lie algebras can be equipped with the non-degenerate Killing form given by

$$
\mathcal{K}(K_1, K_2) = \begin{cases} 2(N+1)\,\mathrm{Tr}(K_1 K_2) \\ 2(N-1)\,\mathrm{Tr}(K_1 K_2) \end{cases},
\tag{14}
$$

where we represent algebra elements as linear maps on the phase space $V$ as introduced in section 2.1.1. The Killing form is negative definite for fermions, while for bosons it has the signature $(N(N+1), N^2)$, *i.e.*, $N(N+1)$ positive and $N^2$ negative directions.

### 2.2.2 2-out-of-3 property

By choosing the right basis, we can bring all three structures simultaneously into their standard form. In fact, bringing two structures into the standard form is sufficient, because relation (6) ensures that also the third structure will be in its standard form. Moreover, all transformations that preserve two of the three structures will actually preserve all three structures. First, we can

always choose an orthonormal basis with respect to $G$, such that $G \equiv \mathbb{1}$. Second, due to (6), $J$ and $\Omega$ will have the same matrix representations and thus $J$ is represented by an antisymmetric matrix. Third, due to $J^2 = -\mathbb{1}$, we can always apply an orthogonal transformation to bring $J$ into block diagonal form, such that

$$G \equiv \left( \begin{array}{c|c} \mathbb{1} & 0 \\ \hline 0 & \mathbb{1} \end{array} \right), \ \Omega \equiv \left( \begin{array}{c|c} 0 & \mathbb{1} \\ \hline -\mathbb{1} & 0 \end{array} \right), \ J \equiv \left( \begin{array}{c|c} 0 & \mathbb{1} \\ \hline -\mathbb{1} & 0 \end{array} \right). \tag{15}$$

These standard forms are preserved by the intersection

$$\mathrm{U}(N) = \mathrm{O}(2N) \cap \mathrm{Sp}(2N, \mathbb{R}) \cap \mathrm{GL}(N, \mathbb{C}), \tag{16}$$

where the RHS satisfies the 2-out-of-3 property, meaning that the intersection of any two out of the three groups is sufficient.

If we are given the group $\mathcal{G}$, *i.e.*, $\mathrm{Sp}(2N, \mathbb{R})$ for bosons or $\mathcal{O}(2N, \mathbb{R})$ for fermions, it suffices for a group element $M \in \mathcal{G}$ to preserve $J$ with $MJM^{-1} = J$ to lie in the unitary group $\mathrm{U}(N)$, *i.e.*,

$$\mathrm{U}(N) = \left\{ M \in \mathcal{G} \,\middle|\, [M, J] = 0 \right\}. \tag{17}$$

The same condition can also be used to restrict the Lie algebra $\mathfrak{g}$ of the group $\mathcal{G}$ to its unitary subalgebra

$$\mathfrak{u}(N) = \left\{ K \in \mathfrak{g} \,\middle|\, [K, J] = 0 \right\}. \tag{18}$$

We can define $\mathfrak{u}_\perp(N)$ as the orthogonal complement of $\mathfrak{u}(N)$ in $\mathfrak{g}$ with respect to the Killing form $\mathcal{K}$, *i.e.*,

$$\mathfrak{u}_\perp(N) = \left\{ K \in \mathfrak{g} \,\middle|\, \mathcal{K}(K, \tilde{K}) \, \forall \, \tilde{K} \in \mathfrak{u}(N) \right\}. \tag{19}$$

While this definition via the Killing form is rather indirect, we can find a much simpler characterization of $\mathfrak{u}_\perp(N)$ as the following proposition states.

**Proposition 1.** *The orthogonal complement* $\mathfrak{u}_\perp(N)$ *is*

$$\mathfrak{u}_\perp(N) = \left\{ K \in \mathfrak{g} \,\middle|\, \{K, J\} = 0 \right\}, \tag{20}$$

*where we use the Killing form* $\mathcal{K}$ *as in* (19).

*Proof.* We first prove that any element $K$ with $\{K, J\} = 0$ is in $\mathfrak{u}_\perp(N)$. For this, we note that any $\tilde{K} \in \mathfrak{u}(N)$ satisfies $[\tilde{K}, J] = 0$ and thus $\tilde{K} = \frac{1}{2}(\tilde{K} - J\tilde{K}J)$. We can therefore compute the Killing form as

$$\mathcal{K}(K, \tilde{K}) \propto \mathrm{Tr}\, K(\tilde{K} - J\tilde{K}J). \tag{21}$$

Clearly, if $K$ anti-commutes with $J$, we find $\mathcal{K}(K, \tilde{K}) \propto \mathrm{Tr}(K\tilde{K} + JK\tilde{K}J) = \mathrm{Tr}(K\tilde{K} - K\tilde{K}) = 0$, where we used cyclicity of the trace and $J^2 = -\mathbb{1}$.

Second, we prove vice versa that for $K$ satisfying $\mathcal{K}(K, \tilde{K}) = 0$ for all $\tilde{K} \in \mathfrak{u}(N)$, we have $\{J, K\} = 0$. For this, we define $\tilde{K} = (K - JKJ)$, which clearly satisfies $[\tilde{K}, J] = 0$ and thus $\tilde{K} \in \mathfrak{u}(N)$. We can now evaluate

$$\mathcal{K}(K, \tilde{K}) \propto \mathrm{Tr}\, K(K - JKJ) = \frac{1}{2}\mathrm{Tr}(K - JKJ)^2 \propto \mathcal{K}(\tilde{K}, \tilde{K}), \tag{22}$$

which only vanishes if $\tilde{K} = 0$ implying $\{K, J\} = 0$, as claimed. $\qquad\square$

### 2.2.3 Relation to complex vector spaces

Every finite-dimensional Kähler space is automatically a complex Hilbert space and vice versa. For a complex number $z = x + \mathrm{i}y$ with $x, y \in \mathbb{R}$, complex scalar multiplication $\odot : \mathbb{C} \times V \to V$ of a vector $v^a \in V$ is

$$(z \odot v)^a = (x\mathbb{1} + yJ)^a{}_b v^b\,, \tag{23}$$

where we have $(\mathrm{i}^2) \odot v = J^2 v = -v$, *i.e.*, $J$ plays the role of multiplication by the imaginary unit on the vector space. Moreover, we have the inner product

$$\langle u, v \rangle = u^a (g_{ab} - \mathrm{i}\omega_{ab}) v^b = u(g - \mathrm{i}\omega)v\,, \tag{24}$$

which we can verify to be anti-linear in the first and linear in the second component, *i.e.*,

$$\begin{aligned}
\langle z \odot u, v \rangle &= u(x + yJ^{\intercal})(g - \mathrm{i}\omega)v = z^* \langle u, v \rangle\,, \\
\langle u, z \odot v \rangle &= u(g + \mathrm{i}\omega)(x + yJ)v = z \langle u, v \rangle\,,
\end{aligned} \tag{25}$$

where we used $J^{\intercal}(g + \mathrm{i}\omega) = \mathrm{i}g + \omega$ and $(g + \mathrm{i}\omega)J = \mathrm{i}g - \omega$ following from (6).

Instead of treating $V$ as a $N$-dimensional complex vector space, where $J$ represents the imaginary unit i, we could also complexify $V$ to find the $2N$-dimensional complex vector space $V_{\mathbb{C}}$, where we allow complex linear combinations of our original phase space vectors and on which $J$ represents a complex-linear map with eigenvalues $\pm\mathrm{i}$. Consequently, we can decompose this complexified space $V_{\mathbb{C}}$ and its dual into the eigenspaces of $J$ and its adjoint $J^{\intercal}$, such that

$$V_{\mathbb{C}} = V^+ \oplus V^- \quad \text{and} \quad V_{\mathbb{C}}^* = (V^*)^+ \oplus (V^*)^-\,, \tag{26}$$

where $V^{\pm}$ and $(V^*)^{\pm}$ refers to the right and left eigenspaces of $J$ with eigenvalue $\pm\mathrm{i}$, respectively. We can define the respective projectors

$$P^{\pm} = \frac{1}{2}(\mathbb{1} \mp \mathrm{i}J)\,, \tag{27}$$

which project onto $V^{\pm}$. When applied to the real subspace $V$, $P^{\pm}$ actually provides an isomorphism between $V$ and $V^{\pm}$ as $2N$-dimensional real vector spaces. Applying complex conjugation (of $V_{\mathbb{C}}$) to an element of $V^+$ maps it to a respective element in $V^-$ and vice versa, which is the same as identifying $V^+$ and $V^-$ via $V$ through $P^{\pm}$. All of these structures naturally appear when constructing so-called creation and annihilation operators in the quantum theory, as will be discussed in Sec. 2.3.

In the case of an infinite dimensional vector space $V$, we can use the same construction to get a complex vector space with inner product, but typically we will need to complete the space using the induced norm to get a Hilbert space. Different Kähler structures induce potentially inequivalent norms leading to different completions, which is related to unitarily inequivalent representations of the algebra of observables. This is discussed in more detail in section 2.3.7 on field theories.

A linear map $K$ on $V$ is complex-linear if it commutes with $J$, *i.e.*, $[K, J] = 0$ and it is complex anti-linear if it anti-commutes with $J$, *i.e.*, $\{K, J\} = 0$. We can decompose any linear map $K$ uniquely into its linear and anti-linear parts $K_{\pm}$ given by

$$K_{\pm} = \frac{1}{2}(K \pm JKJ) \quad \text{with} \quad K = K_+ + K_-\,, \tag{28}$$

such that $\{K_+, J\} = 0$ and $[K_-, J] = 0$. We find

$$\mathrm{Tr}(K_- K_+) = \tfrac{1}{4}\mathrm{Tr}(K^2 + JK^2J + KJKJ - JKJK) = 0\,, \tag{29}$$

which is proportional to the Killing form $\mathcal{K}(K_+, K_-)$ on $\mathfrak{g}$, which we defined in (14). Therefore, the decomposition splits $K$ over $\mathfrak{u}(N)$ and its orthogonal complement $\mathfrak{u}_\perp(N)$, which corresponds exactly to this decomposition into complex linear algebra elements $K_-$ forming $\mathfrak{u}(N)$ and complex anti-linear algebra elements $K_+$ forming $\mathfrak{u}_\perp(N)$.

If we choose a basis, in which the Kähler structures take the standard form (15), we can identify real $2N$-dimensional vectors $v$ with $N$-dimensional complex vectors $\widetilde{v}$ via

$$v \equiv (v_2, v_1) \in \mathbb{R}^{2N} \quad \Longleftrightarrow \quad \widetilde{v} = v_1 + \mathrm{i}v_2 \in \mathbb{C}^N\,. \tag{30}$$

In this basis, the inner product defined in (25) is given by $\langle u, v \rangle = \widetilde{u}^\dagger \widetilde{v}$.

For a general linear map $K : V \to V$, we have the matrix representations

$$K \equiv \begin{pmatrix} A & B \\ C & D \end{pmatrix}, \quad K_\pm \equiv \begin{pmatrix} \mp A_\pm & B_\pm \\ \pm B_\pm & A_\pm \end{pmatrix}, \tag{31}$$

where we have $A_\pm = \tfrac{1}{2}(A \mp D)$ and $B_\pm = \tfrac{1}{2}(B \pm C)$. We can then define the complex $N$-by-$N$ matrices

$$\widetilde{K}_\pm \equiv A_\pm + \mathrm{i}B_\pm\,, \tag{32}$$

where $\widetilde{K}_+$ is complex anti-linear and $\widetilde{K}_-$ is complex linear, as explained previously. We therefore find the following matrix representation of complex $N$-by-$N$ matrices and $N$-dimensional complex vectors:

$$\widetilde{Kv} \equiv \widetilde{K}_+ \widetilde{v}^* + \widetilde{K}_- \widetilde{v}\,, \tag{33}$$

*i.e.*, when converting the $2N$-dimensional real vector $Kv$ into an $N$-dimensional complex vector $\widetilde{Kv}$ under the above identification, it is the same as acting according to (33) with $\widetilde{K}_\pm$ on $\widetilde{v}$ and its complex conjugate $\widetilde{v}^*$.

In summary, every $2N$-dimensional Kähler space is equivalent to an $N$-dimensional complex Hilbert space. Under this identification, the Kähler structures $(G, \Omega, J)$ correspond to the Hilbert space inner product and multiplication by i. With this, it also becomes apparent how our definitions of the groups $\mathrm{GL}(N, \mathbb{C})$ and $\mathrm{U}(N)$ from (8) are related to the standard definitions as complex $N$-by-$N$ matrices. There are several reasons why we describe Kähler spaces as $2N$-dimensional real vector spaces rather than using the complex formulation. First, the classical phase space considered for bosonic or fermionic systems does not start out as a Kähler space, but as a real vector space, where only one of the required structures ($\Omega$ for bosons, $G$ for fermions) is given. Second, we will consider various real linear maps $K$, which would need to be decomposed into $\widetilde{K}_+$ and $\widetilde{K}_-$ in the complex language. Third, we will later complexify phase space $V$ and its dual $V^*$ leading to $2N$-dimensional complex vector spaces $V_\mathbb{C}$ and $V_\mathbb{C}^*$, which only make sense when $V$ and $V^*$ are treated as real vector spaces.

### 2.2.4 Non-Kähler subspaces

Given a real subspace $A \subset V$ of a Kähler space $V$ equipped with $(G, \Omega, J)$, we can restrict the bilinear forms $g$ and $\omega$ onto $A$. We will denote these restrictions by $g_A$ and $\omega_A$. Due to the fact that $g$ is positive definite, also the restriction $g_A$ is positive definite and has an inverse $G_A$. Using this, we define the restricted linear complex structure as $J_A = -G_A \omega_A$. At this stage, we can ask what conditions on $A$ result in $(G_A, \Omega_A, J_A)$ being a Kähler space.

**Proposition 2.** *Given a Kähler space $V$ with structures $(G, \Omega, J)$, a real subspace $A \subset V$ with restricted structures $(G_A, \Omega_A, J_A = -G_A \omega_A)$ is a Kähler space if and only if $J_A^2 = -\mathbb{1}_A$.*

*Proof.* We need to check the condition for each structure and need to ensure that the three structures are related by (6). The latter is ensured by construction. The restriction $g_A$ continues to be positive definite and has an inverse $G_A$. The restriction $\omega_A$ continues to be antisymmetric, but may not be non-degenerate. However, if $\omega_A$ is non-degenerate, then the linear map $J_A$ could not have full rank. Therefore, $J_A^2 = -\mathbb{1}_A$ guarantees not only that $J_A$ satisfies the conditions to be a linear complex structure, but it also ensures that $\omega_A$ is non-degenerate, has an inverse $\Omega_A$ and thus satisfies the conditions of a symplectic form. $\square$

There are several ways how a subspace can fail to be a Kähler space. For example, any odd dimensional real subspace will not be a Kähler space. For our purpose, we will be interested in specific classes of subspaces which define bosonic and fermionic subsystems.

**Definition 3.** *Given a Kähler space $V$ with structure $(g, \omega, J)$, we refer to a subspace $A \in V$ as*

- **bosonic subsystem** *if $\omega_A$ on $A$ is non-degenerate,*

- **fermionic subsystem** *if $\dim(A)$ is even.*

*We also define the complementary subsystem $B$ as*

$$B = \begin{cases} \{v \in V \mid v^a \omega_{ab} u^b = 0 \,\forall\, u \in A\} & \textbf{(bosons)} \\ \{v \in V \mid v^a g_{ab} u^b = 0 \,\forall\, u \in A\} & \textbf{(fermions)} \end{cases}, \tag{34}$$

*which are commonly referred to as the symplectic and orthogonal complement of $A$ in $V$, respectively. The resulting decomposition $V = A \oplus B$ induces an equivalent dual decomposition $V^* = A^* \oplus B^*$.*

In essence, this definition ensures that the restrictions $G_A$ and $\Omega_A$ are a proper positive definite metric and a proper symplectic form, respectively. Therefore, bosonic or fermionic subsystems $A$ fail to be Kähler spaces if and only if these two structures are incompatible in the sense of definition 2, *i.e.*, they fail to give rise to proper linear complex structure $J_A = G_A \omega_A$, such that $J_A^2 = -\mathbb{1}_A$.

### 2.2.5 Cartan decomposition

For a Kähler space $V$, the Cartan decomposition provides a unique decomposition $M = Tu$ of every group element $M \in \mathcal{G}$ into a piece $u \in \mathrm{U}(N)$ that preserves the Kähler structures and another piece $T$ with $\{T, J\} = TJ + JT = 0$.

We begin by fixing compatible Kähler structures $(G, \Omega, J)$ on $V$. For every group element $M \in \mathcal{G}$, which is either the symplectic group $\mathrm{Sp}(2N, \mathbb{R})$ preserving $\Omega$ or the orthogonal group $\mathrm{O}(2N, \mathbb{R})$ preserving $G$, we find new Kähler structures

$$(G_M, \Omega_M, J_M) := (MGM^{\mathsf{T}}, M\Omega M^{\mathsf{T}}, MJM^{-1}), \tag{35}$$

of which $\Omega_M = \Omega$ for bosons and $G_M = G$ for fermions. We can multiply $M$ by an element $u \in \mathrm{U}(N)$, that preserves $(G, \Omega, J)$, without changing $(G_M, \Omega_M, J_M)$, *i.e.*,

$$(G_{Mu}, \Omega_{Mu}, J_{Mu}) = (G_M, \Omega_M, J_M). \tag{36}$$

It defines an equivalence relation on the group $\mathcal{G}$, namely

$$M \sim \tilde{M} \quad \Leftrightarrow \quad \exists u \in \mathrm{U}(N) : \tilde{M} = Mu, \tag{37}$$

where $J_M$ is the same for all $M \in [M]$ within a given equivalence class.

The Cartan decomposition attempts to fix a unique representative $T \in [M]$ in the equivalence class of $M$. This is always possible for bosons and almost always for fermions, namely if $M \in \mathrm{SO}(2N, \mathbb{R})$, *i.e.*, if $M$ is connected to the identity. The basic idea is to search for $T = e^{K_+}$, where $K_+ \in \mathfrak{u}_\perp(N)$ defined in (19). This ensures that $K_+$ anti-commutes with $J$, so that we find $e^{K_+}J = Je^{-K_+}$. With this, we can compute

$$TJT^{-1} = T^2 J = J_M \quad \Rightarrow \quad T^2 = -J_M J, \tag{38}$$

where we used $J^{-1} = -J$. It is useful to define the so-called *relative complex structure*

$$\Delta = -J_M J, \tag{39}$$

which encodes exactly the relative information between $J$ and $J_M$ and is thus independent of the representative $M \in [M]$. Based on the above calculation, we would like to set $T = \sqrt{\Delta}$, but the question is under which conditions this square root is well-defined and unique.

**Proposition 3.** *Given Kähler structures $(G, \Omega, J)$ and a group element $M \in \mathcal{G}$ being either symplectic or orthogonal, the relative complex structure $\Delta = -MJM^{-1}J$ has the following properties:*

- **Bosons.** *All eigenvalues of $\Delta$ are positive and come in pairs of the form $(e^\rho, e^{-\rho})$ with $\rho \in [0, \infty)$, such that its square root $T = \sqrt{\Delta}$ is unique, satisfies $TJ = JT^{-1}$ and has eigenvalues $(e^{\rho/2}, e^{-\rho/2})$.*

- **Fermions.** *Eigenvalues either come in quadruples $(e^{i\theta}, e^{i\theta}, e^{-i\theta}, e^{-i\theta})$ with $\theta \in (0, \pi)$ or in pairs $(1, 1)$ or $(-1, -1)$. If $-1$ is not an eigenvalue, we can define $T = \sqrt{\Delta}$ uniquely, such that it satisfies $TJ = JT^{-1}$ and has eigenvalues $(e^{i\theta/2}, e^{i\theta/2}, e^{-i\theta/2}, e^{-i\theta/2})$. If the eigenvalue pair $(-1, -1)$ appears an even number of times, we can still define an appropriate $T$, but it will not be unique. If the pair $(-1, -1)$ appears an odd number of times, there is no $T$, such that $T^2 = \Delta$ and $TJ = JT^{-1}$ hold at the same time.*

*Proof.* Using $J^{-1} = -J$, we find $\Delta^{-1} = J\Delta J^{-1}$, which implies that $\Delta$ and $\Delta^{-1}$ have the same spectrum, *i.e.*, all eigenvalues appear in pairs $(\lambda, \lambda^{-1})$.
**Bosons.** For bosons, we can use $M\Omega M^\intercal = \Omega$ from (8) and $J = -G\omega$ from (6) to show that $\Delta = MGM^\intercal g$. The matrix representations of $MGM^\intercal$ and $g$ are both symmetric and positive-definite. Consequently, $\Delta$ is diagonalizable with positive eigenvalues. Therefore, the spectrum of $\Delta$ must consist of pairs $(e^\rho, e^{-\rho})$, such that $T = \sqrt{\Delta}$ with eigenvalues $(e^{\rho/2}, e^{-\rho/2})$ is well-defined and unique. We can therefore choose a basis, where $(G, \Omega, J)$ decompose into 2-by-2 blocks in the standard form of (15), such that $\Delta \equiv \oplus_i \Delta^{(i)}$ and $T \equiv \oplus_i T^{(i)}$ are given by

$$\Delta^{(i)} = \begin{pmatrix} e^{\rho_i} & 0 \\ 0 & e^{-\rho_i} \end{pmatrix}, \quad T^{(i)} = \begin{pmatrix} e^{\rho_i/2} & 0 \\ 0 & e^{-\rho_i/2} \end{pmatrix}. \tag{40}$$

It also follows that we have $TJ = JT^{-1}$.
**Fermions.** For fermions, we can use $MGM^\intercal = G$ from and $J = \Omega g$ from (6) to show that $\Delta = M\Omega M^\intercal \omega$. The matrix representations of $M\Omega M^\intercal$ and $\omega$ are both anti-symmetric and non-degenerate, so the spectrum of $\Delta$ has the same properties as the product of two anti-symmetric matrices. As proven in [28], such products satisfy the Stenzel condition, which ensures that every eigenvalue appears an even number of times. Moreover, we have $\Delta \in \mathrm{O}(2N, )$, whose

group elements are known to be diagonalizable with eigenvalues of modulus 1. Therefore, the eigenvalues of $\Delta$ split into quadruples of the form $(e^{i\theta}, e^{i\theta}, e^{-i\theta}, e^{-i\theta})$ and possible pairs of the form $(1, 1)$ and $(-1, -1)$. Similar to the bosonic, we can decompose $(G, \Omega, J)$ into 4-by-4 and some 2-by-2 blocks of the standard form (15), such that $\Delta = \oplus_i \Delta^{(i)}$ and $T = \oplus_i T^{(i)}$. The respective 4-by-4 blocks then take the form

$$
\Delta^{(i)} \equiv \begin{pmatrix}
\cos\theta_i & \sin\theta_i & 0 & 0 \\
-\sin\theta_i & \cos\theta_i & 0 & 0 \\
0 & 0 & \cos\theta_i & -\sin\theta_i \\
0 & 0 & \sin\theta_i & \cos\theta_i
\end{pmatrix},
\tag{41}
$$

whose eigenvalues are $(e^{i\theta_i}, e^{i\theta_i}, e^{-i\theta_i}, e^{-i\theta_i})$, such that

$$
T^{(i)} \equiv \begin{pmatrix}
\cos\frac{\theta_i}{2} & \sin\frac{\theta_i}{2} & 0 & 0 \\
-\sin\frac{\theta_i}{2} & \cos\frac{\theta_i}{2} & 0 & 0 \\
0 & 0 & \cos\frac{\theta_i}{2} & -\sin\frac{\theta_i}{2} \\
0 & 0 & \sin\frac{\theta_i}{2} & \cos\frac{\theta_i}{2}
\end{pmatrix}
\tag{42}
$$

and we have $TJ = JT^{-1}$. Consequently, $T$ is unique if all $\theta_i \in [0, \pi)$. For $\theta_i = \pi$ associated to a $(-1, -1, -1, -1)$ eigenvalue quadruple, we have $\Delta^{(i)} \equiv -\mathbb{1}$, for which there is no unique square root $T^{(i)}$, but a whole family

$$
T^{(i)} \equiv \begin{pmatrix}
0 & \cos\phi & 0 & \sin\phi \\
-\cos\phi & 0 & -\sin\phi & 0 \\
0 & \sin\phi & 0 & -\cos\phi \\
-\sin\phi & 0 & \cos\phi & 0
\end{pmatrix}
\tag{43}
$$

parametrized by an angle $\phi \in [0, \pi]$. If there is a remaining eigenvalue pair $(-1, -1)$ that cannot be paired up with another one, we find that the candidates for $T^{(i)}$ satisfying $(T^{(i)})^2 \equiv -\mathbb{1}_2$ are given by

$$
T^{(i)} \equiv \begin{pmatrix}
a & b \\
-\frac{1+a^2}{b} & -a
\end{pmatrix},
\tag{44}
$$

of which none satisfies $T^{(i)} J^{(i)} T^{(i)} = J^{(i)}$. Therefore, such $T$ does not exist. A single eigenvalue block $(-1, -1)$ in $\Delta$ can be created by the group element

$$
M^{(i)} \equiv \begin{pmatrix}
1 & 0 \\
0 & -1
\end{pmatrix} \Rightarrow \Delta^{(i)} \equiv -MJM^{-1}J \equiv -\mathbb{1}_2,
\tag{45}
$$

which lies in the part of $O(2, \mathbb{R})$ that is not connected to the identity with $\det M^{(i)} = -1$. If we have an odd number of such blocks, the resulting matrix $M$ will also have $\det M = -1$ and thus lies in the part $O^-(2N, \mathbb{R})$ not connected to the identity. We therefore see that $T = \sqrt{\Delta}$ with $TJ = JT^{-1}$ does not exist if and only if $M \in O^-(2N, \mathbb{R})$, in which case $\Delta$ has an odd number of eigenvalue pairs $(-1, -1)$. $\square$

This allows us to define the Cartan decomposition of most group elements $M \in \mathcal{G}$.

**Definition 4.** *Given a Kähler space $V$ with structures $(G, \Omega, J)$ and a group element $M$ connected to the identity (i.e., $M \in SO(2N, \mathbb{R})$ for fermions), we define the Cartan decomposition as*

$$
M = Tu \text{ with } u \in U(N) \text{ and } TJT = J.
\tag{46}
$$

Table 1: *Classical theory and Kähler structures.* This table summarizes and compares our methods to describe bosonic and fermionic Gaussian states using Kähler structures covered in section 2.

| structure | bosons | fermions |
|---|---|---|
| classical phase space | $\xi^a \in V \simeq \mathbb{R}^{2N}$ | |
| dual phase space | $w_a \in V^* \simeq \mathbb{R}^{2N}$ | |
| defining structure | symplectic form $\Omega^{ab}$ | positive definite metric $G^{ab}$ |
| dual structure | $\omega_{ab}$ with $\Omega^{ac}\omega_{cb} = \delta^a{}_b$ | $g_{ab}$ with $G^{ac}g_{cb} = \delta^a{}_b$ |
| Poisson bracket | $\{f_a, g_b\}_- = f_a g_b \Omega^{ab}$ | $\{f_a, g_b\}_+ = f_a g_b G^{ab}$ |
| classical algebra | Symmetric algebra $\mathrm{Sym}(V^*)$ generated by $V^*$ | Grassmann algebra $\mathrm{Grass}(V^*)$ generated by $V^*$ |
| Kähler structures | $(G, \Omega, J)$ with $J^2 = -\mathbb{1}$ and $J = -G\omega = \Omega g$ | |
| Complex multiplication | $(z \odot v) = (x\mathbb{1} + yJ)v$ for $z = x + \mathrm{i}y$ | |
| Complex inner product | $\langle u, v \rangle = u^a(g_{ab} - \mathrm{i}\omega_{ab})v^b$ | |
| structure group $\mathcal{G}$ | $\mathrm{Sp}(2N, \mathbb{R}) = \{M \in \mathrm{GL}(2N, \mathbb{R}) \mid M\Omega M^{\mathsf{T}} = \Omega\}$ | $\mathrm{O}(2N, \mathbb{R}) = \{M \in \mathrm{GL}(2N, \mathbb{R}) \mid MGM^{\mathsf{T}} = G\}$ |
| structure algebra $\mathfrak{g}$ | $\mathfrak{sp}(2N, \mathbb{R}) = \{K \in \mathfrak{gl}(2N, \mathbb{R}) \mid K\Omega + \Omega K^{\mathsf{T}} = 0\}$ | $\mathfrak{so}(2N, \mathbb{R}) = \{K \in \mathfrak{gl}(2N, \mathbb{R}) \mid KG + GK^{\mathsf{T}} = 0\}$ |
| dimension | $N(2N+1)$ | $N(2N-1)$ |
| intersecting group | $\mathrm{U}(N) = \mathrm{Sp}(2N, \mathbb{R}) \cap \mathrm{O}(2N, \mathbb{R}) = \{M \in \mathcal{G} \mid [M, J] = 0\}$ | |
| intersecting algebra | $\mathfrak{u}(N) = \mathfrak{sp}(2N, \mathbb{R}) \cap \mathfrak{so}(2N, \mathbb{R}) = \{M \in \mathfrak{g} \mid [K, J] = 0\}$ | |
| dimension | $N^2$ | |
| orthogonal complement | $\mathfrak{u}_\perp(N) = \{K \in \mathfrak{g} \mid \{K, J\} = 0\}$ | |
| algebra decomposition | $\mathfrak{g} = \mathfrak{u}(N) \oplus \mathfrak{u}_\perp(N)$ | |
| element decomposition | $K_\pm = \frac{1}{2}(K \pm JKJ)$ with $K_+ \in \mathfrak{u}_\perp(N)$ and $K_- \in \mathfrak{u}(N)$ | |
| dimension of $\mathfrak{u}_\perp$ | $N(N+1)$ | $N(N-1)$ |
| Subsystems $A \subset V$ | $\omega_A$ non-degenerate | $\dim(A)$ even |
| Cartan decomposition $M = Tu$ | $T = \sqrt{\Delta}$ with $u = MT^{-1}$ | |
| relative complex structure $\Delta$ | $\Delta = -J_M J$ for $J_M = MJM^{-1}$ | |
| spectrum of $\Delta$ | $(e^{\rho_i}, e^{-\rho_i})$ | $(e^{\vartheta_i}, e^{\vartheta_i}, e^{-\vartheta_i}, e^{-\vartheta_i})$, $(1,1)$ or $(-1,-1)$ |
| Space of $J$ | $\mathcal{M}_b = \{J \in \mathrm{Sp}(2N, \mathbb{R}) \mid J^2 = -\mathbb{1}, J\Omega > 0\}$ | $\mathcal{M}_f = \{J \in \mathrm{O}(2N, \mathbb{R}) \mid J^2 = -\mathbb{1}\}$ |
| Symmetric space | type DIII: $\mathcal{M}_b \simeq \mathrm{Sp}(2N, \mathbb{R})/\mathrm{U}(N)$ | type CI: $\mathcal{M}_f \simeq \mathrm{O}(2N, \mathbb{R})/\mathrm{U}(N)$ |
| Manifold dimension | $N(N+1)$ | $N(N-1)$ |
| quantization procedure | $\mathrm{Sym}(V^*) \longrightarrow \mathrm{Weyl}(V^*, \Omega)$ | $\mathrm{Grass}(V^*) \longrightarrow \mathrm{Cliff}(V^*, G)$ |
| linear observables | $\hat{\xi}^a \overset{q,p}{\equiv} (q_1, \cdots, q_N, p_1, \cdots, p_N) \overset{a, a^\dagger}{\equiv} (\hat{a}_1, \ldots, \hat{a}_N, \hat{a}_1^\dagger, \ldots, \hat{a}_N^\dagger) \in V \simeq \mathbb{R}^{2N}$ | |
| algebra representation $\widehat{K}$ | $-\frac{\mathrm{i}}{2}\omega_{ac}K^c{}_b \hat{\xi}^a \hat{\xi}^b$ | $\frac{1}{2}g_{ac}K^c{}_b \hat{\xi}^a \hat{\xi}^b$ |
| group representation $\mathcal{U}(M, z)$ | $\mathcal{U}^\dagger(M, z)\hat{\xi}^a \mathcal{U}(M, z) = M^a{}_b \hat{\xi}^b + z^a$ | |
| total number operator | $\hat{N}_J = \frac{1}{2}(g_{ab} - \mathrm{i}\omega_{ab})\hat{\xi}^a \hat{\xi}^b$ | |
| unitary equivalence | $\langle \tilde{J} | \hat{N}_J | \tilde{J} \rangle = \frac{1}{4}(g_{ab} - \mathrm{i}\omega_{ab})(\tilde{G}^{ab} + \mathrm{i}\tilde{\Omega}^{ab}) < \infty$ | |
| of $\mathcal{F}_{(G,\Omega,J)}$ and $\mathcal{F}_{(\tilde{G},\tilde{\Omega},\tilde{J})}$ | $\langle \tilde{J} | \hat{N}_J | \tilde{J} \rangle = -\frac{1}{4}\mathrm{tr}(\mathbb{1} - \Delta)$ | $\langle \tilde{J} | \hat{N}_J | \tilde{J} \rangle = \frac{1}{4}\mathrm{tr}(\mathbb{1} - \Delta)$ |

This decomposition is unique for bosons and almost unique for fermions, as discussed in proposition 3. It further follows that $T$ can always be written as $T = e^{K_+}$ with $K_+ \in \mathfrak{u}_\perp(N)$. In summary, the Cartan decomposition $M = Tu$ is unique for all group elements $M \in \mathrm{Sp}(2N, \mathbb{R})$ and almost all group elements $M \in \mathrm{SO}(2N, \mathbb{R})$. Only in the special case, where $\Delta = -MJM^{-1}J$ has eigenvalue quadruples $(-1, -1, -1, -1)$, the square root is not unique. Finally, the Cartan decomposition does not exist if $\Delta = -MJM^{-1}J$ has an odd number of eigenvalue pairs $(-1, -1)$.

### 2.2.6 Symmetric spaces

We will see in section 3.1.1 that the manifolds of pure bosonic or fermionic Gaussian states are isomorphic to the inequivalent ways a bosonic or fermionic phase space can be turned into a Kähler space. In this section, we will construct the respective manifolds in purely geometric terms without making an explicit reference to Gaussian states or Hilbert spaces and show that they are so-called symmetric spaces.

Given a symplectic form $\Omega$ for bosons or a positive definite metric $G$ for fermions, we define the submanifolds

$$\mathcal{M}_b = \left\{ J \in \mathrm{Sp}(2N, \mathbb{R}) \,\middle|\, J^2 = -\mathbb{1}, J\Omega > 0 \right\}, \tag{47}$$

$$\mathcal{M}_f = \left\{ J \in \mathrm{O}(2N, \mathbb{R}) \,\middle|\, J^2 = -\mathbb{1} \right\} \tag{48}$$

of $\mathcal{G}$. This definition ensures that for every $J \in \mathcal{M}$, we have a triple of compatible Kähler structures $(G, \Omega, J)$, where $\Omega$ for bosons or $G$ for fermions is fixed a priori. We will now show that these manifolds are isomorphic to the quotient $\mathcal{G}/\mathrm{U}(N)$ and satisfy the conditions of what is known in mathematics as symmetric spaces.

**Proposition 4.** *Given a single element $J_0 \in \mathcal{M}$, we can generate the full manifold $\mathcal{M}$ as*

$$\mathcal{M} = \left\{ MJ_0M^{-1} \,\middle|\, M \in \mathcal{G} \right\}. \tag{49}$$

*For every element $J \in \mathcal{M}$, there exists a whole equivalence class of group elements $\{M \in \mathcal{G} \,|\, J = MJ_0M^{-1}\}$ that map $J_0$ to $J$. Therefore, the manifold $\mathcal{M}$ is isomorphic to the $\mathcal{G}/\mathrm{U}(N)$ with*

$$\mathrm{U}(N) = \{u \in \mathcal{G} \,|\, uJ_0u^{-1} = J_0\}. \tag{50}$$

*Proof.* Given $J_0$, let us show that for every $J \in \mathcal{M}$, there exists a group element $M$ with $J = MJ_0M^{-1}$. We define $\Delta = -JJ_0$ and use the same arguments as in proposition 3 to show that there exists a respective $T = \sqrt{\Delta}$, such that $J = TJ_0T^{-1}$ and we can just choose $M = T$. Only if $\Delta$ has an odd number of eigenvalue pairs $(-1, -1)$, we construct $M$ from blocks just like we would construct $T$, but in the last block associated to $(-1, -1)$, we choose $M^{(i)}$ as in (45), which does the job.

Having shown that for any $J \in \mathcal{M}$, there exists an $M \in \mathcal{G}$ with $J = MJ_0M^{-1}$, let us ask how many there are. Given two $M, \tilde{M}$ with $J = MJ_0M^{-1} = \tilde{M}J_0\tilde{M}^{-1}$, this relation implies that $u := \tilde{M}^{-1}M$ satisfies $uJ_0u^{-1} = J_0$ and thus $M \sim \tilde{M}$ in the sense of (37). This also shows that the set $\mathcal{M}$ is isomorphic to $\mathcal{G}/\sim = \mathcal{G}/\mathrm{U}(N)$. $\square$

In mathematics, the quotients $\mathcal{M} \simeq \mathcal{G}/\mathrm{U}(N)$ are known as symmetric spaces of type DIII (bosons: $\mathcal{M}_b$) and type[6] CI (fermions: $\mathcal{M}_f$). The fact that they are symmetric spaces follows from the following proposition.

---

[6]Note that a symmetric space CI is $\mathrm{SO}(2N)/\mathrm{U}(N)$, so $\mathcal{M}_f$ technically consists of two copies of CI.

**Proposition 5.** *The manifold of Gaussian states is a symmetric space* $\mathcal{M} = \mathcal{G}/\mathrm{U}(N)$.

*Proof.* A quotient manifold $\mathcal{G}/H$ constructed from a subgroup $H \subset \mathcal{G}$ is a symmetric space if and only if we can decompose the Lie algebra as $\mathfrak{g} = \mathfrak{h} \oplus \mathfrak{h}_\perp$, such that

$$[\mathfrak{h}, \mathfrak{h}] \subset \mathfrak{h}, \quad [\mathfrak{h}, \mathfrak{h}_\perp] \subset \mathfrak{h}_\perp, \quad [\mathfrak{h}_\perp, \mathfrak{h}_\perp] \subset \mathfrak{h}. \tag{51}$$

In the case of Gaussian states, we have $\mathfrak{h} = \mathfrak{u}(N)$ and $\mathfrak{h}_\perp = \mathfrak{u}_\perp(N)$ as defined in (19). This follows directly from the defining conditions $[K, J] = 0$ for $K \in \mathfrak{u}(N)$ and $\{K, J\} = 0$ for $K \in \mathfrak{u}_\perp(N)$. $\square$

In summary, we considered a bosonic or fermionic phase space, *i.e.*, the vector space $V$ with either a symplectic form $\Omega$ or a metric $G$, and asked: How many inequivalent ways are there to turn this vector space (with given $\Omega$ or $G$) into a Kähler space with compatible structures $(G, \Omega, J)$? The answer turned out to be the manifolds $\mathcal{M}_{b/f}$ which could either be embedded into $\mathcal{G}$ or written as quotient $\mathcal{G}/\mathrm{U}(N)$.

## 2.3 Quantum theory

We have seen that the classical bosonic and fermionic phase space is already equipped with one of the three structures that form a Kähler space, namely a symplectic form or a positive definite metric, respectively. We will use these structures to construct the quantum theory in two steps. First, we deform the classical algebra of observables to obtain a Weyl or Clifford algebra, promoting Poisson brackets to commutators and anti-commutators. Second, we build a representation of these algebras as Hermitian operators acting on a Hilbert space, defined in terms of Kähler structures.

### 2.3.1 Abstract algebra of observables

Using the ingredients introduced in the previous section, in particular the algebra of classical observables and the Poisson bracket, we can construct the abstract algebra of quantum observables. This is the first step of the quantization procedure, as at this point we have not yet chosen a representation of algebra elements as operators acting on some Hilbert space. We promote the Poisson brackets to canonical commutation relations (CCR) for bosons and to canonical anticommutation relations (CAR) for fermions, *i.e.*,

$$\begin{aligned} [\hat{\xi}^a, \hat{\xi}^b] &= \mathrm{i} \Omega^{ab} \mathbb{1}, \quad \textbf{(bosons)} \\ \{\hat{\xi}^a, \hat{\xi}^b\} &= G^{ab} \mathbb{1}, \quad \textbf{(fermions)} \end{aligned} \tag{52}$$

where $\mathbb{1}$ is the identity element in the algebra and we adopt units $\hbar = 1$. The commutator and the anticommutator are defined as usual: $[\hat{\xi}^a, \hat{\xi}^b] = \hat{\xi}^a \hat{\xi}^b - \hat{\xi}^b \hat{\xi}^a$ and $\{\hat{\xi}^a, \hat{\xi}^b\} = \hat{\xi}^a \hat{\xi}^b + \hat{\xi}^b \hat{\xi}^a$. This turns the symmetric algebra of bosonic observables into the Weyl algebra $\mathrm{Weyl}(V^*, \Omega)$ and the Grassmann algebra of fermionic observables into the Clifford algebra $\mathrm{Cliff}(V^*, G)$:

$$\begin{aligned} \mathrm{Sym}(V^*) &\longrightarrow \mathrm{Weyl}(V^*, \Omega), \quad \textbf{(bosons)} \\ \mathrm{Grass}(V^*) &\longrightarrow \mathrm{Cliff}(V^*, G). \quad \textbf{(fermions)} \end{aligned} \tag{53}$$

Throughout this manuscript, we will consistently present examples with respect to the two standard bases

$$\hat{\xi}^a \stackrel{q,p}{\equiv} (\hat{q}_1, \cdots, \hat{q}_N, \hat{p}_1, \cdots, \hat{p}_N), \tag{54}$$

$$\stackrel{a, a^\dagger}{\equiv} (\hat{a}_1, \cdots, \hat{a}_N, \hat{a}_1^\dagger, \cdots, \hat{a}_N^\dagger). \tag{55}$$

The first basis consists of Hermitian operators that are typically referred to as quadrature operators (bosons) and Majorana operators (fermions), while the second basis consists of bosonic or fermionic creation and annihilation operators, as summarized in table 2. The two bases are characterized by the property that the symplectic form (for bosons) and the metric (for fermions) takes the following real standard forms

$$
\begin{aligned}
\Omega^{ab} &\overset{q,p}{\equiv} \begin{pmatrix} 0 & \mathbb{1} \\ -\mathbb{1} & 0 \end{pmatrix} \overset{a,a^\dagger}{\equiv} \begin{pmatrix} 0 & -\mathrm{i}\mathbb{1} \\ \mathrm{i}\mathbb{1} & 0 \end{pmatrix}, \quad \textbf{(bosons)} \\
G^{ab} &\overset{q,p}{\equiv} \begin{pmatrix} \mathbb{1} & 0 \\ 0 & \mathbb{1} \end{pmatrix} \overset{a,a^\dagger}{\equiv} \begin{pmatrix} 0 & \mathbb{1} \\ \mathbb{1} & 0 \end{pmatrix}, \qquad \textbf{(fermions)}
\end{aligned}
\tag{56}
$$

where $\overset{q,p}{\equiv}$ and $\overset{a,a^\dagger}{\equiv}$ indicate that the RHS corresponds to the matrix representation with respect to one of the two standard bases (55) or (54). Note that these standard bases are only determined up to an overall group transformation in $\mathcal{G}$ that will preserve the respective structures.

### 2.3.2   Hilbert space and Fock basis

A Hilbert space representation of the algebra of observables is obtained via the Fock basis construction. Consider $N$ dual vectors $v_{ia} \in V_{\mathbb{C}}^*$, $(i = 1, \ldots, N)$ and define the associated annihilation and creation operators

$$
\hat{a}_i = v_{ia}\, \hat{\xi}^a\,, \qquad \hat{a}_i^\dagger = v_{ia}^*\, \hat{\xi}^a\,.
\tag{57}
$$

We impose canonical commutation and anticommutation relations for bosonic and for fermionic operators:

$$
\begin{aligned}
{[\hat{a}_i, \hat{a}_j]} &= 0\,, \quad {[\hat{a}_i, \hat{a}_j^\dagger]} = \delta_{ij}\,\mathbb{1}\,, \quad \textbf{(bosons)} \\
\{\hat{a}_i, \hat{a}_j\} &= 0\,, \quad \{\hat{a}_i, \hat{a}_j^\dagger\} = \delta_{ij}\,\mathbb{1}\,. \quad \textbf{(fermions)}
\end{aligned}
\tag{58}
$$

Due to (52), the dual vectors $v_{ia}$ satisfy the conditions

$$
\begin{aligned}
\Omega^{ab}\, v_{ia} v_{jb} &= 0\,, \quad \Omega^{ab}\, v_{ia}^* v_{jb} = \mathrm{i}\,\delta_{ij}\,, \quad \textbf{(bosons)} \\
G^{ab}\, v_{ia} v_{jb} &= 0\,, \quad G^{ab}\, v_{ia}^* v_{jb} = \delta_{ij}\,. \quad \textbf{(fermions)}
\end{aligned}
\tag{59}
$$

We can then define a state $|0,\ldots,0; v\rangle$ as the vacuum with respect to $v$ annihilated by $\hat{a}_i$,

$$
\hat{a}_i |0,\ldots,0; v\rangle = 0 \qquad i = 1,\ldots,N.
\tag{60}
$$

A unitary representation is constructed by defining the orthonormal Fock basis given by

$$
\begin{aligned}
\left\{ |n_1 \ldots n_N; v\rangle \,\big|\, n_i \in \mathbb{N} \right\}, &\quad \textbf{(bosons)} \\
\left\{ |n_1 \ldots n_N; v\rangle \,\big|\, n_i = 0, 1 \right\}, &\quad \textbf{(fermions)}
\end{aligned}
\tag{61}
$$

such that the action of the operators $\hat{a}_i$ and $\hat{a}_j^\dagger$ onto this basis satisfies

$$
\begin{aligned}
\hat{a}_i |\ldots n_i \ldots; v\rangle &= \sqrt{n_i}\, |\ldots n_i{-}1 \ldots; v\rangle\,, \\
\hat{a}_i^\dagger |\ldots n_i \ldots; v\rangle &= \sqrt{n_i{+}1}\, |\ldots n_i{+}1 \ldots; v\rangle\,.
\end{aligned}
\tag{62}
$$

Basis vectors can be obtained from the vacuum state $|0, \cdots, 0; v\rangle$ via

$$
|n_1, \ldots, n_N; v\rangle = \prod_{i=1}^{N} \left( \frac{(\hat{a}_i^\dagger)^{n_i}}{\sqrt{n_i!}} \right) |0, \cdots, 0; v\rangle\,,
\tag{63}
$$

where we have $n_i \in \mathbb{N}$ for bosons and $n_i \in \{0, 1\}$ for fermions. We denote $\mathcal{H}$ the Hilbert space of the system.

### 2.3.3 Algebra representation

Elements of the symplectic algebra $\mathfrak{sp}(2N,\mathbb{R})$ and of the orthogonal algebra $\mathfrak{so}(2N)$ can be represented as quadratic operators on the Hilbert space $\mathcal{H}$. We can represent a Lie algebra element $K$ as an operator $\widehat{K}$ via the identification

$$
K^a{}_b \quad \Longleftrightarrow \quad \widehat{K} = \begin{cases} -\frac{\mathrm{i}}{2}\omega_{ac}K^c{}_b\,\hat{\xi}^a\hat{\xi}^b & \textbf{(bosons)} \\ \frac{1}{2}g_{ac}K^c{}_b\,\hat{\xi}^a\hat{\xi}^b & \textbf{(fermions)} \end{cases}. \tag{64}
$$

Using the canonical commutation or anticommutation relations, one can verify that this is indeed a Lie algebra representation satisfying

$$
[\widehat{K}_1,\widehat{K}_2] = \widehat{[K_1,K_2]} = \begin{cases} -\frac{\mathrm{i}}{2}\omega_{ac}[K_1,K_2]^c{}_b\,\hat{\xi}^a\hat{\xi}^b & \textbf{(bosons)} \\ \frac{1}{2}g_{ac}[K_1,K_2]^c{}_b\,\hat{\xi}^a\hat{\xi}^b & \textbf{(fermions)} \end{cases}. \tag{65}
$$

Next, we will see that exponentiating operators $\widehat{K}$ gives rise to a projective representation of the respective Lie group.

### 2.3.4 Projective group representations

The bosonic and fermionic Fock spaces come naturally equipped with projective representations of the symplectic group $\mathrm{Sp}(2N,\mathbb{R})$ and of the orthogonal group $\mathrm{O}(2N)$. For bosons, we also have the (Abelian) group of phase space displacements given by $V$ with its vector addition as group operation, which leads to the extension of $\mathrm{Sp}(2N,\mathbb{R})$ as inhomogeneous symplectic group $\mathrm{ISp}(2N,\mathbb{R})$.

The projective representation $\mathcal{S} : \mathcal{G} \to \mathrm{Lin}(\mathcal{H})$ of the squeezing group $\mathcal{G}$ can be constructed by exponentiating quadratic operators $\widehat{K}$. Given a Lie algebra element $K \in \mathfrak{g}$, we represent the group element $M = e^K$ as

$$
\mathcal{S}(e^K) = \pm e^{\widehat{K}}, \tag{66}
$$

which is only defined up to an overall sign. This definition can be consistently extended to all $M \in \mathcal{G}$, *i.e.*, also those which cannot be written as $e^K$, by multiplication such that

$$
\mathcal{S}(M_1)\mathcal{S}(M_2) = \pm \mathcal{S}(M_1 M_2) \tag{67}
$$

holds, as proven in [26, 29]. Using the Baker-Campbell-Hausdorff formula, we can verify the relation

$$
\mathcal{S}^\dagger(M)\hat{\xi}^a\mathcal{S}(M) = M^a{}_b\,\hat{\xi}^b. \tag{68}
$$

For fermions[7], we also need to include the representation of a group element with $\det(M) = -1$, which defines the operator

$$
\mathcal{S}(M_w) = w_a\,\hat{\xi}^a, \qquad \textbf{(fermions)} \tag{69}
$$

where $(M_w)^a{}_b = w_c G^{ca} w_b - \delta^a{}_b$ and $w_a \in V^a$ is assumed to satisfy $w_a G^{ab} w_b = 2$.

---

[7]While the group $\mathcal{G} = \mathrm{Sp}(2N,\mathbb{R})$ for bosons is connected and completely generated by (66), the group $\mathcal{G} = \mathrm{O}(2N,\mathbb{R})$ for fermions consists of two disconnected components, of which only the subset $\mathrm{SO}(2N,\mathbb{R})$ connected to the identity is generated by (66). By including the operators (69), we can reach group elements $M^a{}_b = v_c G^{ca} v_b - \delta^a{}_b$ with $\det(M) = -1$. To give some intuition, let us note that $M \equiv \mathrm{diag}(1,-1,\cdots,-1)$ with respect to an orthonormal basis, in which $v \equiv (\sqrt{2},0,\cdots,0)$.

Furthermore, the set of all operators $\pm\mathcal{S}(M)$ can be understood as a faithful representation of the double cover of $\mathcal{G}$, called the metaplectic group $\mathrm{Mp}(2N,\mathbb{R})$ for bosons and the pin group $\mathrm{Pin}(2N)$ for fermions, where $\mathrm{Pin}(2N)$ relates to $\mathrm{Spin}(2N)$ just as $\mathrm{O}(2N)$ to $\mathrm{SO}(2N)$.

The group of phase space translations $V$ is represented as displacement operators $\mathcal{D}: V \to \mathrm{Lin}(\mathcal{H})$ satisfying

$$\mathcal{D}(z) = \begin{cases} e^{-\mathrm{i}z^a \omega_{ab}\hat{\xi}^b} & \textbf{(bosons)} \\ e^{-z^a g_{ab}\hat{\xi}^b} & \textbf{(fermions)} \end{cases}, \tag{70}$$

which satisfies the relations

$$\mathcal{D}(z_1)\mathcal{D}(z_2) = \begin{cases} e^{-\frac{\mathrm{i}}{2}z_1^a \omega_{ab} z_2^b}\,\mathcal{D}(z_1 + z_2) & \textbf{(bosons)} \\ e^{-\frac{1}{2}z_1^a g_{ab} z_2^b}\,\mathcal{D}(z_1 + z_2) & \textbf{(fermions)} \end{cases}, \tag{71}$$

and thus forms a projective representation. Note that for fermions, the phase space vector $z^a$ is Grassmann valued and thus not physical. Consequently, we will be mostly interested in the bosonic case, but fermionic displacements can still be used as a calculational tool, as we will see.

We can extend the group $\mathcal{G}$ to its inhomogeneous version $\mathrm{I}\mathcal{G}$, whose elements are pairs $(M,z)$ with $M \in \mathcal{G}$ and $z^a \in V$ with the group action

$$(M_1, z_1)\cdot(M_2, z_2) = (M_1 \cdot M_2, z_1 + M_1 z_2). \tag{72}$$

We can define $\mathcal{U}: \mathrm{I}\mathcal{G} \to \mathrm{Lin}(\mathcal{H})$ by

$$\mathcal{U}(M,z) = \mathcal{D}(z)\mathcal{S}(M), \tag{73}$$

which satisfies the relations

$$\mathcal{U}(M_1,z_1)\mathcal{U}(M_2,z_2) \simeq \mathcal{U}(M_1 \cdot M_2, z_1 + M_1 z_2), \tag{74}$$

of a projective representation. Again, we will be mostly interested in the bosonic case, where $\mathrm{I}\mathcal{G} = \mathrm{ISp}(2N,\mathbb{R})$ is the inhomogeneous symplectic group. We can use the Baker-Campbell-Hausdorff formula to show that the so constructed projective representation satisfies

$$\mathcal{U}^\dagger(M,z)\hat{\xi}^a\mathcal{U}(M,z) = M^a{}_b\,\hat{\xi}^b + z^a. \tag{75}$$

The following proposition shows the importance of this relation.

**Proposition 6.** *Condition* (75) *determines the unitary operator $\mathcal{U}$ uniquely up to its complex phase.*

*Proof.* First, let us note that any operator $\mathcal{O}: \mathcal{H} \to \mathcal{H}$ can be formally written as a function $\mathcal{O} = f(\hat{\xi}^a)$ satisfying[8] $\mathcal{U}^\dagger f(\hat{\xi}^a)\mathcal{U} = f(\mathcal{U}^\dagger\hat{\xi}^a\mathcal{U})$. Second, if we take $\mathcal{O} = |\psi\rangle\langle\psi|$, our previous observation shows that $\mathcal{U}^\dagger|\psi\rangle\langle\psi|\mathcal{U} = \tilde{\mathcal{U}}^\dagger|\psi\rangle\langle\psi|\tilde{\mathcal{U}}$ for all $|\psi\rangle \in \mathcal{H}$ implies $\mathcal{U}|\psi\rangle = e^{\mathrm{i}\varphi}\tilde{\mathcal{U}}|\psi\rangle$ and thus $\mathcal{U}^\dagger\hat{\xi}^a\mathcal{U} = \tilde{\mathcal{U}}^\dagger\hat{\xi}^a\tilde{\mathcal{U}}$. Third, we observe that $\tilde{\mathcal{U}} = e^{\mathrm{i}\varphi}\mathcal{U}$ satisfies (75), which is thus both necessary and sufficient to characterize $\mathcal{U}$ up to a complex phase. $\qquad\square$

---

[8]For a bosonic or fermionic system with annihilation operators $\hat{a}_i$ and associated vacuum $|0\rangle$, we have the formal function $|0\rangle\langle 0| = f(\hat{\xi}^a) = \lim_{\beta\to\infty} e^{-\beta\sum_i \hat{a}_i^\dagger \hat{a}}\left(\frac{e^\beta - 1}{e^\beta}\right)^N$ from which we can construct any other linear operator by applying creation operators from the left and annihilation operators from the right. Clearly, such functions satisfy $\mathcal{U}^\dagger f(\hat{\xi}^a)\mathcal{U} = f(\mathcal{U}^\dagger\hat{\xi}^a\mathcal{U})$.

### 2.3.5 Mode functions

Mode functions $u_i^a$ are defined by the expansion[9]

$$\hat{\xi}^a = z^a + \sum_{i=1}^{N} \left( u_i^a \, \hat{a}_i + u_i^{*a} \, \hat{a}_i^\dagger \right), \tag{76}$$

with $z^a = 0$ for fermions. The requirement that $\hat{\xi}^a$ and $\hat{a}_i$ satisfy the defining relations for bosons (CCR) and for fermions (CAR), results in the following conditions:

$$
\begin{aligned}
\omega_{ab} u_i^a u_j^b &= 0, \quad \omega_{ab} u_i^{*a} u_j^b = \mathrm{i}\,\delta_{ij}, \quad \textbf{(bosons)} \\
g_{ab} u_i^a u_j^b &= 0, \quad g_{ab} u_i^{*a} u_j^b = \delta_{ij}. \quad \textbf{(fermions)}
\end{aligned}
\tag{77}
$$

In section 2.3.2, we introduced vectors $v_{ia} \in V_{\mathbb{C}}^*$ to define annihilation operators (57). We have

$$\hat{a}_i = v_{ia} (\hat{\xi}^a - z^a) \tag{78}$$

with $v_{ia} \in V_{\mathbb{C}}^*$. The two dual basis $v_{ia}$ and $u_i^a$ satisfy the relations

$$v_{ia} u_j^{*a} = 0, \quad v_{ia} u_j^a = \delta_{ij}, \tag{79}$$

which allow us to express one basis in terms of the other using the canonical structures $\Omega^{ab}$ for bosons and $G^{ab}$ for fermions,

$$
\begin{aligned}
u_i^a &= \mathrm{i}\,\Omega^{ab} v_{ib}^*, \quad \textbf{(bosons)} \\
u_i^a &= G^{ab} v_{ib}^*. \quad \textbf{(fermions)}
\end{aligned}
\tag{80}
$$

Given the Fock vacuum $|v\rangle$ annihilated by $\hat{a}_i$, we can compute the correlation functions in terms of mode functions

$$\langle v | \hat{\xi}^a \hat{\xi}^b | v \rangle = \sum_{i=1}^{N} u_i^a u_i^{*b} = \frac{1}{2}(G^{ab} + \mathrm{i}\Omega^{ab}), \tag{81}$$

where the metric $G^{ab}$ is

$$G^{ab} = \sum_{i=1}^{N} \left( u_i^a u_i^{*b} + u_i^{*a} u_i^b \right) \tag{82}$$

and the symplectic structure $\Omega^{ab}$ is

$$\Omega^{ab} = -\mathrm{i} \sum_{i=1}^{N} \left( u_i^a u_i^{*b} - u_i^{*a} u_i^b \right). \tag{83}$$

We can also express the complex structure $J^a{}_b$ as

$$J^a{}_b = -\mathrm{i} \sum_{i=1}^{N} \left( u_i^a v_{ib} - u_i^{*a} v_{ib}^* \right). \tag{84}$$

Together with the expression of the identity,

$$\delta^a{}_b = \sum_{i=1}^{N} \left( u_i^a v_{ib} + u_i^{*a} v_{ib}^* \right), \tag{85}$$

---

[9]We use the conventions based on [4], while other authors switch the role of $u$ and $u^*$.

we find that a phase-space covariant version $\hat{\xi}_-^a$ of the annihilation operator $\hat{a}_i$ can be introduced:

$$\hat{\xi}_-^a \equiv \frac{1}{2}(\delta^a{}_b + \mathrm{i}J^a{}_b)(\hat{\xi}^b - z^b) = \sum_i u_i^a \, \hat{a}_i \,, \tag{86}$$

$$\hat{\xi}_+^a \equiv \frac{1}{2}(\delta^a{}_b - \mathrm{i}J^a{}_b)(\hat{\xi}^b - z^b) = \sum_i u_i^{a*} \, \hat{a}_i^\dagger \,. \tag{87}$$

Therefore we conclude that, up to a phase, the Gaussian state defined in (111) in terms of the complex structure $J$ and the Fock vacuum associated to the mode function $u$ coincide, $|v\rangle = |J, z\rangle$. A different choice $\tilde{u}_i^a$ of mode functions is associated to a different set of creation and annihilation operators $\hat{b}_i^\dagger, \hat{b}_i$,

$$\hat{\xi}^a = z^a + \sum_{i=1}^N \left( \tilde{u}_i^a \, \hat{b}_i + \tilde{u}_i^{*a} \, \hat{b}_i^\dagger \right). \tag{88}$$

The linear relation between the two sets of operators can be expressed in terms of Bogoliubov coefficients $\alpha_{ij}$ and $\beta_{ij}$,

$$\hat{b}_i = \sum_{j=1}^N (\alpha_{ij}\hat{a}_j + \beta_{ij}\hat{a}_j^\dagger), \tag{89}$$

where

$$\alpha_{ij} = -\mathrm{i}\,\omega_{ab}\,\tilde{u}_i^a u_j^b \,, \qquad \beta_{ij} = -\mathrm{i}\,\omega_{ab}\,\tilde{u}_i^{*a}u_j^b \,, \qquad \textbf{(bosons)}$$

$$\alpha_{ij} = g_{ab}\,\tilde{u}_i^a u_j^b \,, \qquad\quad \beta_{ij} = g_{ab}\,\tilde{u}_i^{*a}u_j^b \,. \qquad \textbf{(fermions)} \tag{90}$$

3 These expressions are equivalent to the relation (49) between two complex structures $J$ and $MJM^{-1}$.

### 2.3.6  Total number operator

As discussed in (2.2), the linear complex structure $J$ as linear map $J : V \to V$ on the classical phase space satisfies the conditions

$$J\Omega J^\mathsf{T} = \Omega \quad \text{and} \quad J\Omega = -\Omega J^\mathsf{T}, \qquad \textbf{(bosons)} \tag{91}$$

$$JGJ^\mathsf{T} = G \quad \text{and} \quad JG = -GJ^\mathsf{T}. \qquad \textbf{(fermions)} \tag{92}$$

This implies that $J$ represents both a group and an algebra element, so we can formally write $J \in \mathcal{G}$ and $J \in \mathfrak{g}$. The latter also implies that we can uniquely identify $J$ with the anti-Hermitian operator $\widehat{J}$ using (64). If we multiply by i, we can define

$$\hat{N}_J = \frac{1}{2}(g_{ab} - \mathrm{i}\omega_{ab})\hat{\xi}^a \hat{\xi}^b = \begin{cases} \mathrm{i}\widehat{J} - \frac{N}{2} & \textbf{(bosons)} \\ \mathrm{i}\widehat{J} + \frac{N}{2} & \textbf{(fermions)} \end{cases}, \tag{93}$$

which turns out to be a positive-definite Hermitian operator with integer spectrum and ground state $\langle J|\hat{N}_J|J\rangle = 0$. If we choose creation/annihilation operators $\hat{a}_i$ and number operators $\hat{n}_i = \hat{a}_i^\dagger \hat{a}_i$ associated to $|J\rangle$, we have

$$\hat{N}_J = \sum_{i=1}^N \hat{n}_i \,, \tag{94}$$

*i.e.*, we recognize $\hat{N}_J$ as the total number operator of the system, which is in one-to-one correspondence to $J$. While the choice of a Gaussian state $|J\rangle$ does not fix the individual creation/annihilation or number operators due to the allowed $\mathrm{U}(N)$ transformations that would mix them among themselves, the total number operator $\hat{N}_J$ is uniquely defined as the quadratic operator (up to a constant) with integer spectrum that has $|J\rangle$ as ground state.

### 2.3.7 Unitary equivalence

In the definition of the Fock representation we choose a basis $v_{ai}$, (57). If we had chosen a different basis $\tilde{v}$, it will be related to $v$ by some linear map $M$ that is symplectic or orthogonal, *i.e.*, satisfies $M\Omega M^{\intercal} = \Omega$ for bosons or $MGM^{\intercal} = G$ for fermions. We can relate the Fock basis $|\{n_i\}; \tilde{v}\rangle$ with the original one $|\{n_i\}; v\rangle$ using the unitary representation $\mathcal{S}(M)$. This leads to the identification

$$|\{n_i\}; \tilde{v}\rangle \cong \mathcal{S}(M)|\{n_i\}; v\rangle \,, \tag{95}$$

where we still have the choice of a complex phase. We can verify that this identification preserves all commutation relations. The vacuum state $|\tilde{J}\rangle = |0, \cdots, 0; \tilde{v}\rangle$ can be identified with the squeezed vacuum

$$|\tilde{J}\rangle = e^{i\varphi}\, \mathcal{S}(M)|J\rangle \,. \tag{96}$$

In the case of infinitely many degrees of freedom, $N \to \infty$, the Fock construction of the Hilbert space of states requires additional care as unitarily inequivalent representations arise. The phenomenon has a classical origin and can be described in terms of Kähler structures.

In quantum field theory, the Fock vacuum of free fields is often defined in terms of mode functions. Different Fock vacua are then related by Bogoliubov transformations [4–6]. We illustrate the relation between the formulation in terms of mode functions and the formulation in terms of Kähler structures discussed here and used in the context of quantum fields in curved spacetimes [10–15].

In the finite-dimensional case, defining symplectic transformations on a bosonic phase space simply requires the notion of a symplectic structure $\Omega$; similarly, defining orthogonal transformations on a finite-dimensional fermionic phase space simply requires the notion of a metric $G$. In the infinite-dimensional case however this is not enough: it is useful to introduce a Kähler structure $(G, \Omega, J)$ already at the classical level. First, we turn phase space in a real Hilbert space via Cauchy completion with respect to the metric $G$. This allows us to restrict linear observable $f(\xi) = w_a \xi^a$ to the ones with *normalizable* $w_a \in V^*$, *i.e.*, $G^{ab} w_a w_b < \infty$. Second, we restrict the class of symplectic and orthogonal transformations. Given a linear map $L : V \to V$, the adjoint with respect to the metric $G$ is $L^{\dagger} = GL^{\intercal}g$ and the Hilbert-Schmidt norm is $\|L\|_G^2 = \mathrm{tr}(LL^{\dagger})$. *Restricted symplectic* transformations $M \in \mathrm{Sp}_J(V)$ and *restricted orthogonal* transformations $M \in \mathrm{O}_J(V)$ are defined as linear transformations in $\mathrm{Sp}_J(V)$ and in $\mathrm{O}(V)$ that satisfy the condition[10]

$$\|MJ - JM\|_G^2 < \infty \,, \tag{97}$$

with respect to the Kähler structure $(G, \Omega, J)$. These restricted transformations play a central role in the Shale [30] and Shale-Stinespring theorems [31].

In the quantum theory, a Fock space $\mathcal{F}_{(G,\Omega,J)}$ associated to the Kähler structure $(G, \Omega, J)$ is constructed starting from a vacuum given by the Gaussian state $|J\rangle$. The two-point correlation

---

[10]This expression is equivalent to the condition $\|J - J_M\|_G^2 < \infty$ with $J_M = MJM^{-1}$ and $M$ bounded, as shown in [8].

function is

$$\langle J|\hat{\xi}^a\hat{\xi}^b|J\rangle = \frac{1}{2}(G^{ab} + i\,\Omega^{ab}),\tag{98}$$

and linear observables $w_a\hat{\xi}^a$ with normalizable $w_a$ have finite dispersion in the state $|J\rangle$. Moreover, in the Fock space $\mathcal{F}_{(G,\Omega,J)}$ we have a notion of total number operator

$$\hat{N}_J = \frac{1}{2}(g_{ab} - i\,\omega_{ab})\hat{\xi}^a\hat{\xi}^b.\tag{99}$$

Given a Gaussian state $|\tilde{J}\rangle$, we can express the expectation value of the total number operator in terms of the relative complex structure $\Delta = -\tilde{J}J$ introduced in (39). In the case of bosons and of fermions, we find

$$\langle \tilde{J}|\hat{N}_J|\tilde{J}\rangle = \frac{1}{4}(g_{ab} - i\,\omega_{ab})(\tilde{G}^{ab} + i\,\tilde{\Omega}^{ab}) = \begin{cases} -\frac{1}{4}\operatorname{tr}(\mathbb{1} - \Delta) & \textbf{(bosons)} \\ +\frac{1}{4}\operatorname{tr}(\mathbb{1} - \Delta) & \textbf{(fermions)} \end{cases}.\tag{100}$$

Two Fock representations $\mathcal{F}_{(G,\Omega,J)}$ and $\mathcal{F}_{(\tilde{G},\tilde{\Omega},\tilde{J})}$ are unitarily equivalent if and only if the expectation value of the number operator $\hat{N}_J$ in the vacuum $|\tilde{J}\rangle$ is finite [32],

$$\langle \tilde{J}|\hat{N}_J|\tilde{J}\rangle < \infty.\tag{101}$$

This condition coincides with the notion of restricted symplectic transformations and of restricted orthogonal transformations as we have the equality

$$\langle \tilde{J}|\hat{N}_J|\tilde{J}\rangle = \frac{1}{8}\|MJ - JM\|_G^2,\tag{102}$$

with $\tilde{J} = MJM^{-1}$. The condition of unitary equivalence between Fock space representations (101) can then be expressed in terms of Bogoliubov coefficients as

$$\langle \tilde{J}|\hat{N}_J|\tilde{J}\rangle = \sum_{ij}|\beta_{ij}|^2 < \infty.\tag{103}$$

In the bosonic case we can also consider Gaussian states with non-vanishing expectation value of linear observables, $|J, z\rangle$. In this case the number operator is

$$\hat{N}_{J,z} = \frac{1}{2}(g_{ab} - i\,\omega_{ab})(\hat{\xi}^a - z^a)(\hat{\xi}^b - z^b),\tag{104}$$

and the expectation value on the state $|\tilde{J}, \tilde{z}\rangle$ is

$$\langle \tilde{J}, \tilde{z}|\hat{N}_{J,z}|\tilde{J}, \tilde{z}\rangle = -\frac{1}{4}\operatorname{tr}(\mathbb{1} - \Delta) + \frac{1}{2}g_{ab}(z^a - \tilde{z}^a)(z^b - \tilde{z}^b).\tag{105}$$

Unitary equivalence of representations then results in the additional requirement that the shift $z - \tilde{z}$ has finite norm in the metric $G^{ab}$.

## 3 Gaussian states

We introduce Gaussian states in a unified formalism to describe bosons and fermions using Kähler structures. While the relationship between Kähler structures and Gaussian states (under the name of quasi-free states) is well known in the mathematical physics literature [8,9], the goal of the following section is to make these tools available to the broader physics community with particular emphasis on quantum information (entanglement theory) and non-equilibrium physics (quantum dynamics).

## 3.1 Pure Gaussian states

Having introduced bosonic and fermionic quantum systems and the mathematical notion of Kähler structures, we can now introduce a unified formalism to describe pure bosonic and fermionic Gaussian states in terms of Kähler structures on the classical phase space. We will then extend our formalism to also describe mixed Gaussian states by violating the Kähler condition in a controlled way. While we characterize bosonic and fermionic Gaussian states through their Kähler structures $(G, \Omega, J)$, there exists a large zoo of different representations ranging from characteristic functions and quasi-probability distributions to Bogoliubov transformations and wave functions. A comprehensive dictionary between different representations and conventions can be found in [25].

### 3.1.1 Definition

We consider a normalized state vector $|\psi\rangle \in \mathcal{H}$, for which we define the one- and two-point functions

$$
\begin{aligned}
z^a &= \langle \psi | \hat{\xi}^a | \psi \rangle , \\
C_2^{ab} &= \langle \psi | (\hat{\xi} - z)^a (\hat{\xi} - z)^b | \psi \rangle ,
\end{aligned}
\tag{106}
$$
(Requirement: $z^a = 0$ for **(fermions)**).

While there certainly exist fermionic states with $z^a \neq 0$, we only restrict to those $|\psi\rangle$ with $z^a = 0$, as we will later show that there are no physical fermionic Gaussian states with $z \neq 0$, *i.e.*, states are either non-Gaussian or only make sense if one takes $z$ to be Grassmann-valued in which case the Gaussian state does not live in the physical Hilbert space.[11]

We can decompose the two-point function $C_2^{ab}$ as

$$
C_2^{ab} = \frac{1}{2}(G^{ab} + \mathrm{i}\Omega^{ab}),
\tag{107}
$$

where $G^{ab}$ and $\Omega^{ab}$ are the symmetric or anti-symmetric parts, respectively, such that

$$
\begin{aligned}
G^{ab} &= C_2^{ab} + C_2^{ba} = \langle \psi | \hat{\xi}^a \hat{\xi}^b + \hat{\xi}^b \hat{\xi}^a | \psi \rangle - z^a z^b \\
\mathrm{i}\Omega^{ab} &= C_2^{ab} - C_2^{ba} = \langle \psi | \hat{\xi}^a \hat{\xi}^b - \hat{\xi}^b \hat{\xi}^a | \psi \rangle .
\end{aligned}
\tag{108}
$$

The properties of the Hermitian inner product imply that $G$ is symmetric and positive definite, while $\Omega$ must be antisymmetric. Note that this does not imply that $G$ and $\Omega$ are compatible Kähler structures. Further note, that for bosons $\Omega$ is already fixed by the canonical commutation relations of $\hat{\xi}^a$, while for fermions $G$ is fixed by the canonical anticommutation relations, such that our decomposition is compatible with our definition from (52). We will also see in footnote 12 that fermionic Gaussian states will require $z^a = 0$.

In summary, only one of the two structures will depend on the state, which we therefore define as the bosonic or fermionic covariance matrix

$$
\Gamma^{ab} = \begin{cases} G^{ab} & \textbf{(bosons)} \\ \Omega^{ab} & \textbf{(fermions)} \end{cases} .
\tag{109}
$$

With this in hand, we can now present two equivalent definitions of Gaussian states:

---

[11]One can make sense of $z^a \neq 0$ for fermionic Gaussian states, but it requires to extend Hilbert space by allowing the multiplication with Grassmann numbers. In this case, fermionic Gaussian states can have Grassmann valued displacements $z^a$. We will consider Grassmann displacements only as a calculational tool, but our formalism can be seamlessly extended to also include them for fermionic Gaussian states and we will comment on this in the following sections. See [33] for more details.

**Definition 5.** *A normalized state vector $|\psi\rangle$ is Gaussian*

(a) *if $J^a{}_b = \Omega^{ac} g_{cb}$ computed from (107) satisfies*

$$J^2 = -\mathbb{1}\,,\tag{110}$$

*or equivalently,*

(b) *if $|\psi\rangle$ is a solution to the equation*

$$\frac{1}{2}(\delta^a{}_b + \mathrm{i}J^a{}_b)(\hat{\xi} - z)^b |\psi\rangle = 0\,,\tag{111}$$

*for some $z^a \in V$ and a linear map $J^a{}_b : V \to V$, which turns out to imply $J^a{}_b = \Omega^{ac} g_{cb}$.*

*We denote $|\psi\rangle$ by $|J, z\rangle$, which is unique up to a complex phase. Note that $z^a = 0$ for fermions.*

*Proof.* In order to prove the equivalence of the two definitions and $z^a = 0$ for fermions, it is useful to introduce $\hat{\xi}^a_{\pm} = \frac{1}{2}(\delta^a{}_b \mp \mathrm{i}J^a{}_b)(\hat{\xi}^b - z^b)$, which satisfy $\hat{\xi}^a = \hat{\xi}^a_+ + \hat{\xi}^a_- + z^a$ and $\hat{\xi}^{\dagger}_{\pm} = \hat{\xi}_{\mp}$. To relate (110) and (111), we compute

$$\langle J, z | \hat{\xi}^a_+ \hat{\xi}^b_- | J, z \rangle = \frac{1}{4}(\mathbb{1} - \mathrm{i}J)^a{}_c C_2^{cd}(\mathbb{1} + \mathrm{i}J^{\mathsf{T}})_d{}^b\,,\tag{112}$$

whose real and imaginary parts are given by

$$\begin{aligned}
\mathrm{Re}\,\langle J, z | \hat{\xi}^a_+ \hat{\xi}^b_- | J, z \rangle &= G + JGJ^{\mathsf{T}} + J\Omega - \Omega J^{\mathsf{T}}\,,\\
\mathrm{Im}\,\langle J, z | \hat{\xi}^a_+ \hat{\xi}^b_- | J, z \rangle &= \Omega + J\Omega J^{\mathsf{T}} + GJ^{\mathsf{T}} - JG\,.
\end{aligned}\tag{113}$$

With this in hand, we can now show both directions:

$\Rightarrow$ The conditions of (a) imply $J(G + \mathrm{i}\Omega)J^{\mathsf{T}} = G + \mathrm{i}\Omega$, $J\Omega = -\Omega J^{\mathsf{T}} = -G$ and $GJ^{\mathsf{T}} = -JG = -\Omega$, which together imply (113) to vanish and thus (b).

$\Leftarrow$ The conditions of (b) imply that (113) vanishes, which we can solve for $\Omega J^{\mathsf{T}} = G + JGJ^{\mathsf{T}} + J\Omega$. Plugging this into $\mathrm{Im}\,\langle J, z | \hat{\xi}^a_+ \hat{\xi}^b_- | J, z \rangle = 0$ gives

$$(\mathbb{1} + J^2)(\Omega + GJ^{\mathsf{T}}) = 0\,.\tag{114}$$

Clearly, we either have $J^2 = -\mathbb{1}$ or $\Omega = -GJ^{\mathsf{T}}$. In the latter case, we can simplify the second equation from (113) to $\Omega = J\Omega J^{\mathsf{T}}$ and multiply by $J^{\mathsf{T}}$ to find $G = -J^2 G$, which finally implies $J^2 = -\mathbb{1}$, as $G$ is non-degenerate. We thus conclude $J^2 = -\mathbb{1}$.
In a second step, we can now compute

$$C_2^{ab} = \begin{cases} (\mathbb{1} + \mathrm{i}J)^a{}_c \Omega^{cd}(\mathbb{1} - \mathrm{i}J^{\mathsf{T}})_d{}^b & \textbf{(bosons)}\\ (\mathbb{1} + \mathrm{i}J)^a{}_c G^{cd}(\mathbb{1} - \mathrm{i}J^{\mathsf{T}})_d{}^b & \textbf{(fermions)} \end{cases}\tag{115}$$

implying $\Omega = J\Omega J^{\mathsf{T}}$ and $\Omega J^{\mathsf{T}} - J\Omega = 2G$ for bosons and $G = JGJ^{\mathsf{T}}$ and $JG - GJ^{\mathsf{T}} = 2\Omega$ for fermions. Together with $J^2 = -\mathbb{1}$, this leads in either case to $\Omega = JG$, which implies (a).

This proves the equivalence. $\qquad\square$

It is remarkable how (111) together with the canonical commutation or anticommutation relations of $\hat{\xi}^a$ suffices to prove (a). We already introduced

$$\hat{\xi}^a_\pm = \frac{1}{2}(\delta^a{}_b \mp \mathrm{i}J^a{}_b)(\hat{\xi}^b - z^b) \tag{116}$$

as first step in the above proof and in (86), but they turn out to be rather useful in general calculations. They can be defined with respect to any pure Gaussian state $|J, z\rangle$ and (115) implies the relations[12]

$$\begin{aligned}
[\hat{\xi}^a_\pm, \hat{\xi}^b_\pm] &= 0, \quad [\hat{\xi}^a_-, \hat{\xi}^b_+] = C^{ab}_2, \quad \textbf{(bosons)} \\
\{\hat{\xi}^a_\pm, \hat{\xi}^b_\pm\} &= 0, \quad \{\hat{\xi}^a_-, \hat{\xi}^b_+\} = C^{ab}_2. \quad \textbf{(fermions)}
\end{aligned} \tag{117}$$

Here, $\hat{\xi}^a_\pm$ represents the appropriately by $z^a$ shifted eigenvectors of $J$, i.e., we have

$$J^a{}_b\hat{\xi}^b_\pm = \pm\mathrm{i}\hat{\xi}^a_\pm. \tag{118}$$

Let us give some intuition on what the linear complex structure $J$ and the respective $\hat{\xi}_\pm$ actually do. As already discussed around (26), we can decompose the (complexified) classical phase space into the eigenspaces

$$V_{\mathbb{C}} = V^+ \oplus V^- \quad \text{and} \quad V^*_{\mathbb{C}} = (V^*)^+ \oplus (V^*)^-. \tag{119}$$

From the perspective of operators, the term $P_\pm = \frac{1}{2}(\mathbb{1} \mp \mathrm{i}J)$ in (111) is a projector $V_{\mathbb{C}} \to V^\pm$, which projects the operator-valued vector $(\hat{\xi} - z)^a$ onto the space of creation and annihilation operators $\hat{\xi}^a_\pm$, respectively. Put differently, the eigenspaces $(V^*)^\pm$ represent the $N$-dimensional complex spaces of creation or annihilation operators. While $z^a$ describes the displacement, $J$ encodes precisely which (complex) linear combinations of observables $\hat{\xi}^a$ form creation and annihilation operators. For a given state vector $|J\rangle = |J, 0\rangle$, it is illuminating to express $\hat{\xi}^a_\pm$ in a basis, in which both $\Omega$ and $G$ simultaneously take the standard forms (56), such that[13]

$$\begin{aligned}
\hat{\xi}_- &\stackrel{q,p}{\equiv} \left(\frac{\hat{a}_1}{\sqrt{2}}, \ldots, \frac{\hat{a}_N}{\sqrt{2}}, \frac{-\mathrm{i}\hat{a}_1}{\sqrt{2}}, \ldots, \frac{-\mathrm{i}\hat{a}_N}{\sqrt{2}}\right) \stackrel{a,a\dagger}{\equiv} (\hat{a}_1, \ldots, \hat{a}_N, 0, \ldots, 0), \\
\hat{\xi}_+ &\stackrel{q,p}{\equiv} \left(\frac{\hat{a}^\dagger_1}{\sqrt{2}}, \ldots, \frac{\hat{a}^\dagger_N}{\sqrt{2}}, \frac{\mathrm{i}\hat{a}^\dagger_1}{\sqrt{2}}, \ldots, \frac{\mathrm{i}\hat{a}^\dagger_N}{\sqrt{2}}\right) \stackrel{a,a\dagger}{\equiv} (0, \ldots, 0, \hat{a}^\dagger_1, \ldots, \hat{a}^\dagger_N),
\end{aligned} \tag{120}$$

where we see explicitly that $\hat{\xi}^a_\pm$ is spanned by creation or annihilation operators, respectively. If we had taken $|J, z\rangle$ instead, each component had been appropriately displaced by $(P_\pm z)^a$.

In summary, a normalized pure Gaussian state $|J, z\rangle$ is (up to a complex phase) uniquely characterized by its displacement vector $z^a \in V$ and either its complex structure $J$ or equivalently its covariance matrix

$$\Gamma^{ab} = \begin{cases} -J^a{}_c\Omega^{cb} & \textbf{(bosons)} \\ J^a{}_c G^{cb} & \textbf{(fermions)} \end{cases}, \tag{121}$$

---

[12]Note that (111) and (117) for fermions together imply

$$\hat{\xi}^a_+\hat{\xi}^b_+|J, z\rangle = z^a z^b|J, z\rangle = -z^b z^a|J, z\rangle = \hat{\xi}^a_-\hat{\xi}^b_-|J, z\rangle$$

and thus $z^a = 0$, unless $z^a$ is a Grassmann variable.

[13]Complex conjugation of the basis $\hat{\xi}^a$ satisfies $\hat{\xi}^{\dagger a} = C^a{}_b\hat{\xi}^b$ implying $\hat{\xi}^{\dagger a}_\pm = C^a{}_b\hat{\xi}^b_\mp$. We have the conjugation matrix

$$C \stackrel{q,p}{\equiv} \begin{pmatrix} \mathbb{1} & 0 \\ 0 & \mathbb{1} \end{pmatrix} \stackrel{a,a\dagger}{\equiv} \begin{pmatrix} 0 & \mathbb{1} \\ \mathbb{1} & 0 \end{pmatrix}.$$

where $\Omega$ and $G$ are fixed background structures for bosons or fermions, respectively.

The choice of a Fock space vacuum is equivalent to selecting a Gaussian state with $\langle \hat{\xi}^a \rangle = 0$. In the case of infinitely many degrees of freedom, two Gaussian states $|J\rangle$ and $|\tilde{J}\rangle$ give rise to unitarily equivalent Fock space representations if the Hilbert-Schmidt norm of $J - \tilde{J}$ is finite.[14]

**Example 1** (Single mode pure Gaussian bosonic states). *We consider a single bosonic mode with $\hat{\xi} \stackrel{q,p}{\equiv} (\hat{q}, \hat{p}) \stackrel{a,a^\dagger}{\equiv} (\hat{a}, \hat{a}^\dagger)$. With respect to the number eigenvectors $|n\rangle$, the most general Gaussian state vector with $z^a = 0$ is*

$$|J\rangle = \frac{1}{\sqrt{\cosh \frac{\rho}{2}}} \sum_{n=0}^\infty \frac{\sqrt{(2n)!}}{2^n n!} \left( -e^{i\phi} \tanh \frac{\rho}{2} \right)^n |2n\rangle, \tag{122}$$

*where $\phi \in [0, 2\pi]$ and $\rho \in [0, \infty)$. With respect to above bases, one finds*

$$G \stackrel{q,p}{\equiv} \begin{pmatrix} \cosh \rho + \cos \phi \sinh \rho & \sin \phi \sinh \rho \\ \sin \phi \sinh \rho & \cosh \rho - \cos \phi \sinh \rho \end{pmatrix} \stackrel{a,a^\dagger}{\equiv} \begin{pmatrix} e^{i\phi} \sinh \rho & \cosh \rho \\ \cosh \rho & -e^{-i\phi} \sinh \rho \end{pmatrix}, \tag{123}$$

$$J \stackrel{q,p}{\equiv} \begin{pmatrix} -\sin \phi \sinh \rho & \cos \phi \sinh \rho + \cosh \rho \\ \cos \phi \sinh \rho - \cosh \rho & \sin \phi \sinh \rho \end{pmatrix} \stackrel{a,a^\dagger}{\equiv} \begin{pmatrix} -i \cosh \rho & i e^{i\phi} \sinh \rho \\ -i e^{-i\phi} \sinh \rho & i \cosh \rho \end{pmatrix}. \tag{124}$$

*In summary, Gaussian states of a single bosonic mode form a two-dimensional plane parametrized by polar coordinates $(\rho, \phi)$.*

**Example 2** (Single and two mode pure Gaussian fermionic states). *We consider a single fermionic mode with $\hat{\xi} \stackrel{q,p}{\equiv} (\hat{q}, \hat{p}) \stackrel{a,a^\dagger}{\equiv} (\hat{a}, \hat{a}^\dagger)$. There are only two distinct pure Gaussian states, which are characterized by the state vectors*

$$\left\{ \begin{array}{l} |J_+\rangle = |0\rangle \\ |J_-\rangle = |1\rangle \end{array} \right\}, \tag{125}$$

*whose covariance matrix and complex structures*

$$\Omega_\pm \stackrel{q,p}{\equiv} \begin{pmatrix} 0 & \pm 1 \\ \mp 1 & 0 \end{pmatrix} \stackrel{a,a^\dagger}{\equiv} \begin{pmatrix} 0 & \mp i \\ \pm i & 0 \end{pmatrix}, \tag{126}$$

$$J_\pm \stackrel{q,p}{\equiv} \begin{pmatrix} 0 & \pm 1 \\ \mp 1 & 0 \end{pmatrix} \stackrel{a,a^\dagger}{\equiv} \begin{pmatrix} \mp i & 0 \\ 0 & \pm i \end{pmatrix}. \tag{127}$$

*In summary, there are only two distinct Gaussian pure states for a single fermionic mode. We consider also two fermionic modes with $\hat{\xi} \stackrel{q,p}{\equiv} (\hat{q}_1, \hat{q}_2, \hat{p}_1, \hat{p}_2) \stackrel{a,a^\dagger}{\equiv} (\hat{a}_1, \hat{a}_2, \hat{a}_1^\dagger, \hat{a}_2^\dagger)$, where the most general Gaussian state vectors are*

$$\left\{ \begin{array}{l} |J_+\rangle = \cos \frac{\theta}{2} |0, 0\rangle + e^{i\phi} \sin \frac{\theta}{2} |1, 1\rangle \\ |J_-\rangle = \cos \frac{\theta}{2} |1, 0\rangle + e^{i\phi} \sin \frac{\theta}{2} |0, 1\rangle \end{array} \right\}, \tag{128}$$

---

[14] Note that the definition of Hilbert-Schmidt norm requires an inner product on the classical phase space. For fermions, we can use the one induced by the background structure $G$, while for bosons we can equivalently use $G^{ab} = J^a{}_c \Omega^{cb}$ induced by $J$ or $\tilde{G}^{ab} = \tilde{J}^a{}_c \Omega^{cb}$ induced by $\tilde{J}$. See section 2.3.7 for further details.

*with $\theta \in [0, \pi]$ and $\phi \in [0, 2\pi]$. Their covariance matrix and complex structure are*

$$
\Omega_\pm \stackrel{q,p}{=}
\begin{pmatrix}
0 & \mp \sin\theta\sin\phi & \pm\cos\theta & \pm\sin\theta\cos\phi \\
\pm\sin\theta\sin\phi & 0 & -\sin\theta\cos\phi & \cos\theta \\
\mp\cos\theta & \sin\theta\cos\phi & 0 & \sin\theta\sin\phi \\
\mp\sin\theta\cos\phi & -\cos\theta & -\sin\theta\sin\phi & 0
\end{pmatrix}
$$
$$
\stackrel{a,a^\dagger}{=}
\begin{pmatrix}
0 & ie^{i\phi}\sin\theta & -i\cos\theta & 0 \\
-ie^{i\phi}\sin\theta & 0 & 0 & -i\cos\theta \\
i\cos\theta & 0 & 0 & -ie^{-i\phi}\sin\theta \\
0 & i\cos\theta & ie^{-i\phi}\sin\theta & 0
\end{pmatrix}, \tag{129}
$$

$$
J_\pm \stackrel{q,p}{=}
\begin{pmatrix}
0 & \mp\sin\theta\sin\phi & \pm\cos\theta & \pm\sin\theta\cos\phi \\
\pm\sin\theta\sin\phi & 0 & -\sin\theta\cos\phi & \cos\theta \\
\mp\cos\theta & \sin\theta\cos\phi & 0 & \sin\theta\sin\phi \\
\mp\sin\theta\cos\phi & -\cos\theta & -\sin\theta\sin\phi & 0
\end{pmatrix}
$$
$$
\stackrel{a,a^\dagger}{=}
\begin{pmatrix}
\mp i\cos\theta & i\delta_\mp e^{-i\phi}\sin\theta & 0 & i\delta_\pm e^{i\phi}\sin\theta \\
i\delta_\mp e^{i\phi}\sin\theta & -i\cos\theta & -i\delta_\pm e^{i\phi}\sin\theta & 0 \\
0 & -i\delta_\pm e^{-i\phi}\sin\theta & \pm i\cos\theta & -i\delta_\mp e^{-i\phi}\sin\theta \\
i\delta_\pm e^{-i\phi}\sin\theta & 0 & -i\delta_\mp e^{i\phi}\sin\theta & i\cos\theta
\end{pmatrix}, \tag{130}
$$

*with $\delta_\pm = \frac{1\pm1}{2}$, i.e., $\delta_+ = 1$ and $\delta_- = 0$. In summary, Gaussian states of two fermionic modes form two disconnected unit spheres parametrized by angles $(\theta, \phi)$, where we further distinguish the Gaussian state vectors of type $|J_+\rangle$ and $|J_-\rangle$. The two sets are distinguished by the parity operator $\hat{P} = \exp(i\pi\hat{N})$, as the total number operator $\hat{N} = \sum_i \hat{a}_i^\dagger \hat{a}_i$ is even for $|J_+\rangle$ and odd for $|J_-\rangle$*

The projective representations $\mathcal{U}(M, z)$ of group elements $M \in \mathcal{G}$ are called Gaussian transformations, because they map Gaussian states into Gaussian states, as we will prove next. They are also known as Bogoliubov transformations, where they are often written in terms of creation and annihilation operators. Any two Gaussian states are related by Gaussian transformations, from which we will uniquely identify a canonical one. This will also enable us to relate the manifold of pure Gaussian states with symmetric spaces, as introduced in section 2.2.6.

**Proposition 7.** *The unitary transformation $\mathcal{U}(M, z)$ defined section 2.3.4 applied to a Gaussian state $|J_0, z_0\rangle$ will map to another Gaussian state $|J_1, z_1\rangle = \mathcal{U}(M, z)|J_0, z_0\rangle = |MJ_0 M^{-1}, Mz_0 + z_1\rangle$, i.e., Gaussian transformations map Gaussian states to Gaussian states.*

*Proof.* Using (75), we compute the 1- and 2-point functions of the resulting state $|\psi\rangle = \mathcal{U}(M, z)|J_0, z_0\rangle$ as

$$
z^a = \langle\psi|\hat{\xi}^a|\psi\rangle = M^a{}_b z_0^b + z^a, \tag{131}
$$
$$
C_2^{ab} = \langle\psi|\hat{\xi}^a \hat{\xi}^b|\psi\rangle - z^a z^b = M^a{}_c C_2^{cd}(M^\mathsf{T})_d{}^b. \tag{132}
$$

Decomposing $C_2 = \frac{1}{2}(G + i\Omega)$ and computing $J = -\Omega g$ as in section 3.1.1 yields

$$
J = -M\Omega_0 M^\mathsf{T}(M^{-1})^\mathsf{T} g M^{-1} = MJ_0 M^{-1}. \tag{133}
$$

From this, it is easy to compute $J^2 = M(-J_0^2)M^{-1} = -\mathbb{1}$, which proves that the resulting state $|\psi\rangle \cong |J, z\rangle$ is Gaussian and thus implies that $\mathcal{U}(M, z)$ maps Gaussian states onto Gaussian states. $\square$

We can now reverse the argument and ask how to find the Gaussian transformation $\mathcal{U}(M, z)$ that transforms a fixed reference state $|J_0, z_0\rangle$ into an arbitrary Gaussian state $|J_1, z_1\rangle$ of our

choice, *i.e.*, $|J_1, z_1\rangle \cong \mathcal{U}(M, z)|J_0, z_0\rangle$. It is easy to see for the displacement as $z = z_1 - z_0$, which is unique. For $M \in \mathcal{M}$, we find the requirement

$$J_1 = M J_0 M^{-1}, \tag{134}$$

which does not determine $M$ uniquely, as we can recall from section 2.2.5. Instead, we find that there is an equivalence class $[M] \subset \mathcal{G}$ of group elements that transform $J_0$ into $J$. Applying $\mathcal{U}(M, z)|J_0, z_0\rangle$ for different $M \in [M]$ will only differ in its complex phase, so that the manifold $\mathcal{M}$ of pure Gaussian states is given by

$$\mathcal{M} = \begin{cases} \mathcal{M}_b \times V & \textbf{(bosons)} \\ \mathcal{M}_f & \textbf{(fermions)} \end{cases}, \tag{135}$$

where we also include displacements for bosons. The dimensions of these manifold can be deduced from the respective symmetric spaces $\mathcal{M}_{f/b}$ and $V$, *i.e.*,

$$\dim \mathcal{M} = \begin{cases} N(N+1) + 2N & \textbf{(bosons)} \\ N(N-1) & \textbf{(fermions)} \end{cases}. \tag{136}$$

We further recall that $\mathcal{M}_b$ is diffeomorphic to $\mathbb{R}^{N(N+1)}$, which implies $\mathcal{M} \simeq \mathbb{R}^{N(N+3)}$ for bosons. For fermions, we have $\mathcal{M} = \mathcal{M}_f \simeq \mathrm{O}(2N, \mathbb{R})/\mathrm{U}(N)$, which is a non-contractible and generally topologically non-trivial manifold consisting of two disconnected components (associated to the two parity sectors).

**Example 3** (Bosonic Gaussian single-mode pure states revisited)**.** *We reconsider Example 1 and choose the reference state vector $|J_0\rangle$ with*

$$G_0 \overset{q,p}{\equiv} \begin{pmatrix} 1 & 0 \\ 0 & 1 \end{pmatrix} \overset{a,a^\dagger}{\equiv} \begin{pmatrix} 0 & 1 \\ 1 & 0 \end{pmatrix}, J_0 \overset{q,p}{\equiv} \begin{pmatrix} 0 & 1 \\ -1 & 0 \end{pmatrix} \overset{a,a^\dagger}{\equiv} \begin{pmatrix} \mathrm{i} & 0 \\ 0 & -\mathrm{i} \end{pmatrix}. \tag{137}$$

*A general symplectic transformation $\mathcal{G} = \mathrm{Sp}(2, \mathbb{R})$ is*

$$M \overset{q,p}{\equiv} \begin{pmatrix} \cos\tau \cosh\frac{\rho}{2} - \sin\theta \sinh\frac{\rho}{2} & -\sin\tau \cosh\frac{\rho}{2} + \cos\theta \sinh\frac{\rho}{2} \\ \sin\tau \cosh\frac{\rho}{2} + \cos\theta \sinh\frac{\rho}{2} & \cos\tau \cosh\frac{\rho}{2} + \sin\theta \sinh\frac{\rho}{2} \end{pmatrix} \overset{a,a^\dagger}{\equiv} \begin{pmatrix} e^{\mathrm{i}\tau}\cosh\frac{\rho}{2} & \mathrm{i}e^{\mathrm{i}\theta}\sinh\frac{\rho}{2} \\ -\mathrm{i}e^{-\mathrm{i}\theta}\sinh\frac{\rho}{2} & e^{-\mathrm{i}\tau}\cosh\frac{\rho}{2} \end{pmatrix},$$

*for which we have $|J\rangle \cong \mathcal{S}(M)|J_0\rangle$ with $\Gamma$ from (123), where $\phi = \tau - \theta$. The stabilizer group of $|J_0\rangle$ consists of*

$$u \overset{q,p}{\equiv} \begin{pmatrix} \cos\varphi & \sin\varphi \\ -\sin\varphi & \cos\varphi \end{pmatrix} \overset{a,a^\dagger}{\equiv} \begin{pmatrix} e^{\mathrm{i}\varphi} & 0 \\ 0 & e^{-\mathrm{i}\varphi} \end{pmatrix}. \tag{138}$$

*From the relative complex structure $\Delta = T^2 = -J J_0$, we compute the generator*

$$K = \log T \overset{q,p}{\equiv} \frac{\rho}{2} \begin{pmatrix} \sin\phi & \cos\phi \\ \cos\phi & -\sin\phi \end{pmatrix} \overset{a,a^\dagger}{\equiv} \frac{\rho}{2} \begin{pmatrix} 0 & \mathrm{i}e^{-\mathrm{i}\phi} \\ -\mathrm{i}e^{\mathrm{i}\phi} & 0 \end{pmatrix}, \tag{139}$$

*such that $|J\rangle \cong e^{\widehat{K}}|J_0\rangle$. We can always change the basis to reach a standard forms $\phi = \frac{\pi}{2}$, where we can read off the eigenvalues $(e^\rho, e^{-\rho})$ of $\Delta$.*

**Example 4** (Fermions revisited)**.** *We reconsider Example 2. For a single fermionic mode, we choose the reference state vector $|J_0\rangle$ with*

$$\Omega_0 \overset{q,p}{\equiv} \begin{pmatrix} 0 & 1 \\ -1 & 0 \end{pmatrix} \overset{a,a^\dagger}{\equiv} \begin{pmatrix} 0 & -\mathrm{i} \\ \mathrm{i} & 0 \end{pmatrix}, \qquad J_0 \overset{q,p}{\equiv} \begin{pmatrix} 0 & 1 \\ -1 & 0 \end{pmatrix} \overset{a,a^\dagger}{\equiv} \begin{pmatrix} -\mathrm{i} & 0 \\ 0 & \mathrm{i} \end{pmatrix}. \tag{140}$$

*The stabilizer subgroup $\mathrm{U}(1)$ consists of the same elements as in (17), which coincides with the group $\mathrm{SO}(2, \mathbb{R})$. Consequently, the only group elements that transform $|J_0\rangle = |J_+\rangle$ into $|J_-\rangle$ lie in*

*the disconnected component. We also reconsider two fermionic modes with reference state vector*
$|J_0\rangle$ *given by*

$$\Omega_0 \overset{q,p}{\equiv} \begin{pmatrix} 0 & \mathbb{1} \\ -\mathbb{1} & 0 \end{pmatrix} \overset{a,a^\dagger}{\equiv} \begin{pmatrix} 0 & -\mathrm{i}\mathbb{1} \\ \mathrm{i}\mathbb{1} & 0 \end{pmatrix}, \qquad J_0 \overset{q,p}{\equiv} \begin{pmatrix} 0 & \mathbb{1} \\ -\mathbb{1} & 0 \end{pmatrix} \overset{a,a^\dagger}{\equiv} \begin{pmatrix} -\mathrm{i}\mathbb{1} & 0 \\ 0 & \mathrm{i}\mathbb{1} \end{pmatrix}. \tag{141}$$

*There is a 4-dimensional subspace of these generators also satisfying* $[K, J_0]$, *which generates*
$U(2) \subset O(4, \mathbb{R})$. *We can reach the most general complex structure* $J_+$ *by a continuous path*
*generated by*

$$K = \frac{1}{2}\log\Delta \overset{q,p}{\equiv} \frac{\theta}{2} \begin{pmatrix} 0 & \cos\phi & 0 & \sin\phi \\ -\cos\phi & 0 & -\sin\phi & 0 \\ 0 & \sin\phi & 0 & -\cos\phi \\ -\sin\phi & 0 & \cos\phi & 0 \end{pmatrix}, \tag{142}$$

*for* $\Delta = -J_+ J_0$. *To reach state vectors of the form* $|J_-\rangle$, *we must also apply an additional trans-*
*formation* $\mathcal{S}(M_v)$ *with* $v \overset{q,p}{\equiv} (\sqrt{2}, 0, 0, 0) \overset{a,a^\dagger}{\equiv} (1, 0, 1, 0)$ *to find* $|J_-\rangle = \mathcal{S}(M_v)|J_+\rangle$. *We can al-*
*ways change basis to reach a standard forms* $\phi = 0$, *where we can read off the eigenvalues*
$(e^{\mathrm{i}\theta}, e^{\mathrm{i}\theta}, e^{-\mathrm{i}\theta}, e^{-\mathrm{i}\theta})$ *of* $\Delta$.

The manifolds of bosonic and fermionic Gaussian states can be embedded into projective
Hilbert space $\mathcal{P}(\mathcal{H})$, from which it inherits the structures to make $\mathcal{M}$ itself a so-called Kähler
manifold. Such manifolds have various desirable properties, but for our purpose it is sufficient
to know that each tangent space $\mathcal{T}_{(J,z)}\mathcal{M}$ is a Kähler space equipped with compatible Kähler
structures $(\mathbf{G}, \mathbf{\Omega}, \mathbf{J})$ which are distinct from the ones $(G, \Omega, J)$ on the classical phase space $V$. A
detailed review can be found in the application section of [24], where it is also derived how the
two types of Kähler structures, *i.e.*, $(\mathbf{G}, \mathbf{\Omega}, \mathbf{J})$ and $(G, \Omega, J)$ are related. Treating the family of
Gaussian states as a Kähler manifold is particularly useful when one tries to approximate non-
Gaussian states, such as ground states of interacting Hamiltonians or the time evolution under
such Hamiltonians. This is known as the Gaussian time-dependent variational principle [34],
which is a special case of more general variational methods [24, 35]. Gaussian states can also
be understood as group theoretic coherent states [36–38] with respect to the symplectic or
orthogonal group. This concept recently led to generalizations [39, 40] relevant for variational
calculations.

At this stage, we have introduced pure Gaussian states in terms of Kähler structures and
in particular, by only specifying the complex structure $J$ (and $z$ in the case of bosons). This
specifies the quantum state uniquely, but leaves the complex phase of the state vector $|J, z\rangle$
undetermined, as this phase is not physical. Next, we will discuss how all relevant properties
of pure bosonic and fermionic Gaussian states can be expressed in terms of $J$ (and potentially
$z$ for bosons). In particular, we will see that many expressions are almost identical for bosons
and fermions, leading to a unified description.

### 3.1.2 Wick's theorem

One of the most important properties of Gaussian states is that we can compute the expectation
values of arbitrary operators (written in powers of $\hat{\xi}^a$) from the one- and two-point functions
$z^a$ and $C_2^{ab}$, which themselves are fixed by commutation or anti-commutation relations as well
as $J$ and $z$. Consequently, the evaluation of expectation values can be performed efficiently
using tensors on the classical phase space, instead of representing operators and states on
Hilbert space $\mathcal{H}$. This is particularly advantageous for bosonic systems, where $\mathcal{H}$ is infinite
dimensional, but also for fermions the Hilbert space dimension grows exponentially with the
number of degrees of freedom, which makes numerics on Hilbert space unfeasible.

We define the $n$-point correlation function of a state $|\psi\rangle$ with $z^a = \langle\psi|\hat{\xi}^a|\psi\rangle$ (requiring $z^a = 0$ for fermions) as

$$C_n^{a_1\cdots a_n} = \langle\psi|(\hat{\xi}-z)^{a_1}\cdots(\hat{\xi}-z)^{a_n}|\psi\rangle\,, \tag{143}$$

which can be efficiently computed as explained below.

**Proposition 8** (Wick's theorem)**.** *For a Gaussian state $|\psi\rangle = |J,z\rangle$, the n-point correlation function can be computes according to:*

*(a) Odd correlation functions vanish, i.e., $C_{2n+1} = 0$.*

*(b) Even correlation functions are given by the sum over all two-contractions*

$$C_{2n}^{a_1\cdots a_{2n}} = \sum_\sigma \frac{|\sigma|}{n!} C_2^{a_{\sigma(1)}a_{\sigma(2)}}\cdots C_2^{a_{\sigma(2n-1)}a_{\sigma(2n)}}\,, \tag{144}$$

*where the permutations $\sigma$ satisfy $\sigma(2i-1) < \sigma(2i)$ and $|\sigma| = 1$ for bosons and $|\sigma| = \mathrm{sgn}(\sigma)$, called parity, for fermions.*

*Proof.* A covariant proof of Wick's theorem is based on the previously introduced operators $\hat{\xi}_\pm^a$ from (116), such that

$$C_n^{a_1\cdots a_n} = \langle J,z|(\hat{\xi}_+ + \hat{\xi}_-)^{a_1}\cdots(\hat{\xi}_+ + \hat{\xi}_-)^{a_n}|J,z\rangle\,. \tag{145}$$

We now use their commutation and anticommutation relations (115) to normal-order the $\hat{\xi}_\pm^{a_i}$, i.e., to bring all $\hat{\xi}_-^{a_i}$ to the right and all $\hat{\xi}_+^{a_i}$ to the left. In doing so, we generate sums of products of $C_2^{ab}$. For odd $n$, every normalordered term contains at least one $\hat{\xi}_\pm^{a_i}$, which annihilates $|J,z\rangle$ or $\langle J,z|$ and $C_n = 0$. For even $n$, we find all possible pairings of the $a_i$, but for fermions we will pick up a minus sign for every anti-commutation, we perform in a given term. This gives an overall sign determined by the number of necessary adjacent transpositions, which is known as the parity of the permutation $\sigma$. Note that every commutation or anticommutation keeps the order of the $a_i$, i.e., we will never find $C_2^{a_i a_j}$ with $i > j$. $\qquad\square$

Note that this is a phase space covariant formulation of Wick's theorem, as it does not require to write $\hat{\xi}^a$ in any basis or expressing it in terms of creation and annihilation operators.

### 3.1.3 Baker-Campbell-Hausdorff

The philosophy of the present paper is formulate most results in a covariant manner, that are independent from any chosen basis of phase space. However, in order to prove these results, it is typically best to bring operators and matrices in certain standard forms, from which one can read off invariant information, such as eigenvalues and their generalization. For example, we saw in section 3.1.1 that any two Gaussian states $|J_0,z_0\rangle$ and $|J,z\rangle$ define a relative complex structure $\Delta$ whose eigenvalues give rise to invariant squeezing parameters $r_i$. In the following, we will present certain key formulas based on the famous Baker-Campbell-Hausdorff relations which will lay the foundations for deriving such covariant formulas.

We consider a system with $N$ bosonic or fermionic degrees of freedom. We have a Gaussian state $|J,0\rangle$ and an algebra element $K \in \mathfrak{g}$. We can decompose any such $K$ uniquely with respect to $J$ into the sum

$$K = K_+ + K_- \quad \text{with} \quad K_\pm = \tfrac{1}{2}(K \pm JKJ)\,, \tag{146}$$

as explained in the context of (28). Recall that we have

$$\widehat{K} = \begin{cases} -\frac{i}{2}k_{ab}\hat{\xi}^a\hat{\xi}^b & \textbf{(bosons)} \\ \frac{1}{2}k_{ab}\hat{\xi}^a\hat{\xi}^b & \textbf{(fermions)} \end{cases}, \tag{147}$$

where we have the definition $k_{ab} = \omega_{ac}K^c{}_b$ for bosons and $k_{ab} = g_{ac}K^c{}_b$ for fermions from (64). Using the definition of $\hat{\xi}^a_{\pm}$ with respect to $|J,0\rangle$ and some effort, we can compute

$$\widehat{K}_+ = \begin{cases} -\frac{i}{2}k_{ab}(\hat{\xi}^a_+\hat{\xi}^b_+ + \hat{\xi}^a_-\hat{\xi}^b_-) & \textbf{(bosons)} \\ \frac{1}{2}k_{ab}(\hat{\xi}^a_+\hat{\xi}^b_+ + \hat{\xi}^a_-\hat{\xi}^b_-) & \textbf{(fermions)} \end{cases}, \tag{148}$$

$$\widehat{K}_- = \begin{cases} -\frac{i}{2}k_{ab}(\hat{\xi}^a_+\hat{\xi}^b_- + \hat{\xi}^a_-\hat{\xi}^b_+) & \textbf{(bosons)} \\ \frac{1}{2}k_{ab}(\hat{\xi}^a_+\hat{\xi}^b_- + \hat{\xi}^a_-\hat{\xi}^b_+) & \textbf{(fermions)} \end{cases}. \tag{149}$$

We therefore see that $\widehat{K}_+$ is a pure squeezing operator, while $|J,0\rangle$ is eigenstate of $\widehat{K}_-$ with eigenvalue

$$\langle J,0|\widehat{K}_-|J,0\rangle = \begin{cases} -\frac{i}{4}\operatorname{Tr}(JK_-) & \textbf{(bosons)} \\ +\frac{i}{4}\operatorname{Tr}(JK_-) & \textbf{(fermions)} \end{cases}. \tag{150}$$

Technically, we can apply the same strategy to a bosonic Gaussian states $|J,z\rangle$ with displacement, in which case we will

**Normal-ordered squeezing.** As discussed previously, the simplest non-trivial example of squeezing requires one bosonic and two fermionic degrees of freedom. In both cases, we have two parameters to describe the precise squeezing operation, which correspond to polar coordinate $(r,\theta)$ for bosons (parametrizing a plane) and spherical angles $(r,\theta)$ for fermions (parametrizing a sphere). The following formulas are well-known in the literature [41, 42] for bosons and can be analogously derived for fermions

$$\exp\left[\frac{r}{2}(e^{i\theta}(\hat{a}^\dagger)^2 - e^{-i\theta}\hat{a}^2)\right] = \exp\left[\frac{1}{2}e^{i\theta}(\tanh r)(\hat{a}^\dagger)^2\right] \times \exp\left[-(\ln\cosh r)(\hat{n}+\tfrac{1}{2})\right] \\ \times \exp\left[-\frac{1}{2}(e^{-i\theta}\tanh r)\hat{a}^2\right], \qquad \textbf{(bosons)} \tag{151}$$

$$\exp\left[r(e^{i\theta}\hat{a}_1^\dagger\hat{a}_2^\dagger + e^{-i\theta}\hat{a}_1\hat{a}_2)\right] = \exp\left[e^{i\theta}\tan r\,\hat{a}_1^\dagger\hat{a}_2^\dagger\right] \times \exp\left[-(\ln\cos r)(\hat{n}_1+\hat{n}_2-1)\right] \\ \times \exp\left[e^{-i\theta}\tan r\,\hat{a}_1\hat{a}_2\right]. \qquad \textbf{(fermions)} \tag{152}$$

Such formulas are needed to compute expectation values of the form $\langle J,0|e^{\widehat{K}_+}|J,0\rangle$. We can use (151) and (152) to derive their covariant normal-ordered counter parts, where we express everything in terms of $\hat{\xi}^a_{\pm}$, namely

$$e^{\widehat{K}_+} = e^{-\frac{i}{2}\omega_{ac}L^c{}_b\hat{\xi}^a_+\hat{\xi}^b_+} \times e^{-\frac{i}{2}\omega_{ac}\log(\mathbb{1}-L^2)^c{}_b(\hat{\xi}^a_+\hat{\xi}^b_- + \frac{i}{4}\Omega^{ab})} \times e^{-\frac{i}{2}\omega_{ac}L^c{}_b\hat{\xi}^a_-\hat{\xi}^b_-}, \qquad \textbf{(bosons)} \tag{153}$$

$$e^{\widehat{K}_+} = e^{\frac{1}{2}g_{ac}L^c{}_b\hat{\xi}^a_+\hat{\xi}^b_+} \times e^{\frac{1}{2}g_{ac}\log(\mathbb{1}-L^2)^c{}_b(\hat{\xi}^a_+\hat{\xi}^b_- - \frac{1}{4}G^{ab})} \times e^{\frac{1}{2}g_{ac}L^c{}_b\hat{\xi}^a_-\hat{\xi}^b_-}, \qquad \textbf{(fermions)} \tag{154}$$

where we defined the linear map

$$L = \tanh K_+ = \tanh(\tfrac{1}{2}\log\Delta) = \mathbb{1} - \frac{2}{\mathbb{1}+\Delta}. \tag{155}$$

The fastest way to verify these relations is based on block-diagonalizing $K_+$ with eigenvalues $\pm r$ for bosons and $\pm ir$ for fermions. Using (155), we can conclude that $L$ has eigenvalues $\pm\tanh r_i$ for bosons and $i\tan r_i$ for fermions.

**Normal-ordered displacement.** We have

$$e^{\alpha\hat{a}^\dagger + \beta\hat{a}} = e^{\alpha\hat{a}^\dagger}e^{\alpha\beta/2}e^{\beta\hat{a}}, \tag{156}$$

which applies to both bosons and fermions (with $\alpha$ and $\beta$ being Grassman variables in the latter case). Relation (156) follows from $e^{X+Y} = e^X e^Y e^{-[X,Y]/2}$, which is valid if $X$ and $Y$ commute with $[X,Y]$. We consider $\hat{\xi}^a_\pm$ associated to a state $|J,0\rangle$, i.e., there is no displacement in $\hat{\xi}^a_\pm$, to derive the covariant form of (156) as

$$e^{v_a \hat{\xi}^a_+ + w_b \hat{\xi}^b_-} = e^{v_a \hat{\xi}^a_+} e^{\frac{1}{2} v_a (C_2^\mathsf{T})^{ab} w_b} e^{\hat{\xi}^b_- w_b}. \tag{157}$$

We can use this to normal-order the bosonic or fermionic displacement operator defined in (70) as

$$\mathcal{D}(z) = \begin{cases} e^{-\mathrm{i}z^a \omega_{ab} \hat{\xi}^b_+} e^{-\frac{1}{4} z^a g_{ab} z^b} e^{-\mathrm{i}z^a \omega_{ab} \hat{\xi}^b_-} & \textbf{(bosons)} \\ e^{-z^a g_{ab} \hat{\xi}^b_+} e^{\frac{\mathrm{i}}{4} z^a \omega_{ab} z^b} e^{-z^a g_{ab} \hat{\xi}^b_-} & \textbf{(fermions)} \end{cases}, \tag{158}$$

which will be crucial for many calculations related to displaced Gaussian states.

**Normal-ordered displacement and squeezing.** When we consider the interplay between displacement and squeezing, we need to normal-order combinations of them. This is based on

$$e^{\alpha \hat{a}} e^{\beta (\hat{a}^\dagger)^2} = e^{\beta(\hat{a}^\dagger)^2 + 2\alpha\beta \hat{a}^\dagger} e^{\alpha^2 \beta} e^{\alpha \hat{a}}, \quad \textbf{(bosons)}$$

$$\begin{aligned} e^{\alpha_1 \hat{a}_1 + \alpha_2 \hat{a}_2} e^{\beta \hat{a}_1^\dagger \hat{a}_2^\dagger} &= e^{\beta(\hat{a}_1^\dagger \hat{a}_2^\dagger + \alpha_1 \hat{a}_2^\dagger - \alpha_2 \hat{a}_1^\dagger)} \\ &\times e^{\alpha_1 \beta \alpha_2} e^{\alpha_1 \hat{a}_1 + \alpha_2 \hat{a}_2}, \end{aligned} \quad \textbf{(fermions)} \tag{159}$$

which can be used to find the general covariant form

$$e^{w_a \hat{\xi}^a_-} e^{v_{bc} \hat{\xi}^b_+ \hat{\xi}^c_+} = e^{v_{bc} \hat{\xi}^b_+ \hat{\xi}^c_+} e^{w_a \hat{\xi}^a_- + 2w_a C_2^{ab} v_{bc} \hat{\xi}^c_+} = e^{v_{bc} \hat{\xi}^b_+ \hat{\xi}^c_+ + 2w_a C_2^{ab} v_{bc} \hat{\xi}^c_+} \times e^{w_a C_2^{ab} v_{bc} (C_2^\mathsf{T})^{cd} w_d} e^{w_a \hat{\xi}^a_-}, \tag{160}$$

where $C_2^{ab}$ represents the 2-point function defined in (107). This allows us to normal-order the expression $\mathcal{D}(z) e^{\widehat{K}_+}$. We need to combine (153) or (154) with (158) and then apply (160) to the anti-normal ordered middle term, which can be reordered as

$$e^{-\mathrm{i}z^a \omega_{ab} \hat{\xi}^b_-} e^{-\frac{\mathrm{i}}{2} \omega_{ac} L^c{}_b \hat{\xi}^a_+ \hat{\xi}^b_+} = e^{-\frac{\mathrm{i}}{2} \omega_{ac} L^c{}_b \hat{\xi}^a_+ \hat{\xi}^b_+ + y_a \hat{\xi}^a_+} e^X e^{-\mathrm{i}z^a \omega_{ab} \hat{\xi}^b_-}, \qquad \textbf{(bosons)} \tag{161}$$

$$e^{-z^a g_{ab} \hat{\xi}^b_-} e^{\frac{1}{2} g_{ac} L^c{}_b \hat{\xi}^a_+ \hat{\xi}^b_+} = e^{\frac{1}{2} g_{ac} L^c{}_b \hat{\xi}^a_+ \hat{\xi}^b_+ - y_a \hat{\xi}^a_+} e^X e^{-z^a g_{ab} \hat{\xi}^b_-}, \qquad \textbf{(fermions)} \tag{162}$$

where we find $y_a = \frac{1}{2} z^b (g - \mathrm{i}\omega)_{bc} L^b{}_a$ based on (160). When computing $\langle J,0|\mathcal{D}(z) e^{\widehat{K}_+}|J,0\rangle$, the most important term is the complex number $X = z^a x_{ab} z^b$, which we compute separately for bosons and fermions. Using (155), (160), (160) and various Kähler relations as summarized in appendix A.3, one finds that $X$ is structurally the same for bosons and fermions and given by

$$X = -\frac{1}{4}\left[\frac{2}{G+\Delta G} - g + \mathrm{i}\left(\omega - \frac{2}{\Omega + \Delta\Omega}\right)\right]_{ab} z^a z^b = \begin{cases} -\frac{1}{4}\left[\frac{2}{G+\tilde{G}} - g - \frac{2\mathrm{i}}{\Omega + \Delta\Omega}\right]_{ab} z^a z^b & \textbf{(bosons)} \\ -\frac{1}{4}\left[\frac{2}{G+\Delta G} + \mathrm{i}\left(\omega - \frac{2}{\Omega + \tilde{\Omega}}\right)\right]_{ab} z^a z^b & \textbf{(fermions)} \end{cases} \tag{163}$$

with $\frac{1}{G+\Delta G} = (G+\Delta G)^{-1}$ and $\frac{1}{\Omega + \Delta\Omega} = (\Omega + \Delta\Omega)^{-1}$. Note that we have $\Delta G = \tilde{G}$ for bosons and $\Delta\Omega = \tilde{\Omega}$ for fermions, where this refers to the covariance matrix that is reached when applying the group transformation $T = \sqrt{\Delta} = e^{K_+}$ to the state with Kähler structures $(G, \Omega, J)$. This calculation will play an important role, when we want to evaluate the inner product between two Gaussian states.

**Combined squeezing and displacement.** We further require an important relation between linear and quadratic operators. We consider

$$\begin{aligned} \widehat{w} &= -\mathrm{i}w_a \hat{\xi}^a, \quad \widehat{K} = -\mathrm{i}\omega_{ac} K^c{}_b \hat{\xi}^a \hat{\xi}^b, \quad \textbf{(bosons)} \\ \widehat{w} &= -w_a \hat{\xi}^a, \quad \widehat{K} = g_{ac} K^c{}_b \hat{\xi}^a \hat{\xi}^b, \qquad \textbf{(fermions)} \end{aligned}, \tag{164}$$

where we assume $w_a$ to be Grassmann valued for fermions, as discussed in the previous paragraph. Note that we will allow for $K \in \mathfrak{g}_{\mathbb{C}}$ and $f \in V_{\mathbb{C}}^*$, i.e., the resulting operators may not be anti-Hermitian, such that their exponentials may not be unitary. The famous Baker-Campbell-Hausdorff relation allows us to find the operator expression

$$\log(e^{\widehat{K}} e^{\widehat{w}}) = \widehat{K} + \widehat{\eta} + w_a B^{ab} w_b \,, \tag{165}$$

where we have introduced the following objects

$$\eta_a = w_b \left( \frac{K}{e^K - \mathbb{1}} \right)^b{}_a \,, \qquad B^{ab} = \begin{cases} iF(K)^a{}_c \Omega^{cb} & \textbf{(bosons)} \\ F(K)^a{}_c G^{cb} & \textbf{(fermions)} \end{cases} \,, \qquad F(K) = \frac{1}{4} \frac{K - \sinh K}{\mathbb{1} - \cosh K} \,. \tag{166}$$

While these relations appear cumbersome at first, they will be crucial to evaluate characteristic functions of Gaussian states. We can prove (165) using the Dynkin formula [43], which gives a formal series of (165) in terms of nested commutators of $\widehat{K}$ and $\widehat{q}$. In our case, only two types of terms survive, namely $[\widehat{w}, [\widehat{K}, \ldots, [\widehat{K}, [\widehat{K}, \widehat{w}]] \ldots ]]$ and $[\widehat{K}, \ldots, [\widehat{K}, [\widehat{K}, \widehat{w}]] \ldots ]$. Using $[\widehat{K}, \widehat{w}] = \widehat{wK}$ with $(wK)_a = w_b K^b{}_a$, we can expand $\eta_a$ and $B^{ab}$ as a power series in $K$ to deduce above functional expressions.

### 3.1.4 Scalar product

We can also use the linear complex structures to compute the inner product $|\langle J, z | \tilde{J}, \tilde{z} \rangle|^2$ between two normalized Gaussian states. For this, we find again that the relative complex structure introduced in (39) provides a covariant way to encode this information.

**Proposition 9.** *The absolute value of the scalar product between two Gaussian states $|J, z\rangle$ and $|\tilde{J}, \tilde{z}\rangle$ is given by*

$$|\langle J, z | \tilde{J}, \tilde{z} \rangle|^2 = e^{-\left| \log \det \frac{\sqrt{\mathbb{1} + \Delta}}{\sqrt{2} \Delta^{1/4}} \right| - \frac{1}{2}(z - \tilde{z})^a (\Gamma + \tilde{\Gamma})^{-1}_{ab}(z - \tilde{z})^b} \,. \tag{167}$$

*This expression simplifies for $z = \tilde{z}$ to*

$$|\langle J, z | \tilde{J}, z \rangle|^2 = \begin{cases} \det \frac{\sqrt{2} \Delta^{1/4}}{\sqrt{\mathbb{1} + \Delta}} & \textbf{(bosons)} \\ \det \frac{\sqrt{\mathbb{1} + \Delta}}{\sqrt{2} \Delta^{1/4}} & \textbf{(fermions)} \end{cases} \,. \tag{168}$$

*Proof.* There are many different ways to prove formula (167), but we will rely on the decomposition already introduced in section 3.1.3. The relevant information is encoded in the squeezing parameters $r_i$ and the displacement parameters $z_i$ for bosons.
We consider the expectation value

$$|\langle J, z | \tilde{J}, z \rangle| = |\langle J, 0 | \tilde{J}, 0 \rangle| = |\langle J, 0 | e^{\widehat{K}_+} | J, 0 \rangle| \,, \tag{169}$$

where $K_+ = \log T = \frac{1}{2} \log \Delta$ with $\Delta = -\tilde{J}J$, i.e., we use the fact that $|\tilde{J}, 0\rangle \cong e^{\widehat{K}_+} |J, 0\rangle = |TJT^{-1}, 0\rangle$. Note that we intentionally only refer to the absolute value of this inner product, as we cannot determine the relative complex phase by only writing $|J, 0\rangle$ and $|\tilde{J}, 0\rangle$. We further write $K_+$ in reference to the decomposition $K = K_+ + K_-$ of (146), where our $K_+$ satisfies $\{K_+, J\} = 0$ and thus represents pure squeezing from $|J, 0\rangle$ to $|\tilde{J}, 0\rangle$. We can use (153) and (154) to compute

$$\langle J, 0 | e^{\widehat{K}_+} | J, 0 \rangle = \begin{cases} \det^{\frac{1}{8}}(\mathbb{1} - L^2) & \textbf{(bosons)} \\ \det^{-\frac{1}{8}}(\mathbb{1} - L^2) & \textbf{(fermions)} \end{cases} \,, \tag{170}$$

where we used $e^{\pm\frac{1}{8}\operatorname{tr}\log(\mathbb{1}-L^2)} = \det^{\pm\frac{1}{8}}(\mathbb{1}-L^2)$. We can now express everything in terms of $\Delta$ via $L = \tanh K_+ = \tanh\log T = \tanh(\frac{1}{2}\log\Delta)$ and simplify the resulting expression by using the identity

$$L = \tanh K_+ = \mathbb{1} - \tanh^2\left(\frac{\log\Delta}{2}\right) = \frac{4\Delta}{(1-\Delta^2)} \tag{171}$$

to find directly (168).

For bosons, we need to do a second step to also include displacement to find (167). For this, we first compute

$$|\langle J,z|\tilde{J},\tilde{z}\rangle| = |\langle J,z|\mathcal{D}^\dagger(z)\mathcal{D}(\tilde{z})\mathcal{S}(M)|J,0\rangle| = |\langle J,0|\mathcal{D}(\tilde{z}-z)e^{\widehat{K}_+}|J,0\rangle|, \tag{172}$$

where we ignored complex phases due to only considering the absolute value and where we have $K_+ = \log T = \frac{1}{2}\log\Delta$. At this stage, we normal order $\mathcal{D}(\tilde{z}-z)$ and $e^{\widehat{K}}$ based on (158) and (153) to find

$$
\begin{aligned}
|\langle J,z|\tilde{J},\tilde{z}\rangle| &= e^{-\frac{1}{4}(z-\tilde{z})^a g_{ab}(z-\tilde{z})^b} e^{\frac{1}{8}\operatorname{tr}\log(\mathbb{1}-L^2)} \\
&\quad \times |\langle J,0|e^{-\mathrm{i}(z-\tilde{z})^a \omega_{ab}\hat{\xi}_-^b} e^{-\frac{1}{2}\omega_{ac}L^c{}_b\hat{\xi}_+^a\hat{\xi}_+^b}|J,0\rangle|,
\end{aligned} \tag{173}
$$

where we encounter in the second line exactly the term discussed in (163), which we can normal-order to find $e^{\operatorname{Re}(x)_{ab}(z-\tilde{z})^a(z-\tilde{z})^b}$ with $x_{ab}$ from (163). This combines with the middle term in (158), so that we find exactly[15]

$$|\langle J,z|\tilde{J},\tilde{z}\rangle| = e^{-\frac{1}{4}(z-\tilde{z})^a(G+\tilde{G})_{ab}^{-1}(z-\tilde{z})^b} e^{\frac{1}{8}\operatorname{tr}\log(\mathbb{1}-L^2)}, \tag{174}$$

which leads to (167). $\qquad\square$

## 3.2 Mixed Gaussian states

In the previous sections, we focused on properties of pure Gaussian states. However, many applications in quantum theory also require the consideration of mixed Gaussian states. Mixed Gaussian states can either be considered as a larger class, which contains pure Gaussian states, or they can be considered as the states that arise if one restricts pure Gaussian states to smaller subsystems.

### 3.2.1 Definition

We recall that a mixed state $\rho : \mathcal{H} \to \mathcal{H}$ is a non-negative Hermitian operator, *i.e.*, $\rho \geq 0$, with $\operatorname{Tr}\rho = 1$. Only if the state is pure, we have $\operatorname{Tr}\rho^2 = 1$, in which case $\rho = |\psi\rangle\langle\psi|$ for some normalized $|\psi\rangle \in \mathcal{H}$ with a single non-zero eigenvalue equal to 1.

Given a mixed state $\rho$, we define its 1- and 2-point function in analogy to (106) as

$$
\begin{aligned}
z^a &= \operatorname{Tr}(\rho\,\hat{\xi}^a), \\
C_2^{ab} &= \operatorname{Tr}\left(\rho(\hat{\xi}-z)^a(\hat{\xi}-z)^b\right), \\
&\text{(Requirement: } z^a = 0 \text{ for \textbf{(fermions)}),}
\end{aligned} \tag{175}
$$

---

[15]For completeness, let us mention that we could use (163) for fermions to derive a similar expression for fermionic Gaussian states with Grassmann displacement $z^a$ and $\tilde{z}^b$ leading to

$$|\langle J,z|\tilde{J},\tilde{z}\rangle| = e^{\frac{\mathrm{i}}{4}(z-\tilde{z})^a(\Omega+\tilde{\Omega})_{ab}^{-1}(z-\tilde{z})^b} e^{-\frac{1}{8}\operatorname{tr}\log(\mathbb{1}-L^2)},$$

which is real in the Grassmann sense as $|\langle J,z|\tilde{J},\tilde{z}\rangle|^* = |\langle J,z|\tilde{J},\tilde{z}\rangle|$.

where we restrict once again to those states with $z^a = 0$ for fermions, as there are no physical fermionic Gaussian states with $z^a \neq 0$. We decompose $C_2^{ab} = \frac{1}{2}(G^{ab} + i\Omega^{ab})$ as in (108) to define the linear map

$$J = \begin{cases} -G\omega & \textbf{(bosons)} \\ +\Omega g & \textbf{(fermions)} \end{cases}, \tag{176}$$

which in general will not satisfy $J^2 = -\mathbb{1}$. Technically, $J$ is therefore not a complex structure, but we may abuse the language and call it a restricted complex structure (as any such $J$ can arise from restricting a complex structure to a subspace), while keeping to use the letter $J$.

We found in definition 5 that it suffices to compute $C_2^{ab}$ of an arbitrary pure state $|\psi\rangle$ and check if the resulting $J$ satisfies $J^2 = -\mathbb{1}$ to check if $|\psi\rangle$ is Gaussian. While we can still compute a $J$ via $C_2^{ab}$ for a mixed state $\rho$, In contrast there is no direct way to read off $J$ if the associated mixed state $\rho$ is Gaussian or not. Instead, we define mixed Gaussian states by the requirement that $\log \rho$ is a quadratic operator, as specified next.

**Definition 6.** *A mixed state $\rho$ is called Gaussian if and only if there exists a Hermitian quadratic operator*[16]

$$\hat{Q} = \begin{cases} q_{ab}(\hat{\xi} - z)^a (\hat{\xi} - z)^b + c & \textbf{(bosons)} \\ iq_{ab}\hat{\xi}^a\hat{\xi}^b + c & \textbf{(fermions)} \end{cases}, \tag{177}$$

*such that $\rho = e^{-\hat{Q}}$. In this case, we denote $\rho$ by $\rho_{(J,z)}$, where $z$ and $J$ are computed from (175).*

To derive properties of mixed Gaussian states, it is useful to bring $q_{ab}$ into block diagonal form, which can always be achieved by an appropriate group transformation $M \in \mathcal{G}$. Put differently, there always exist a basis, such that

$$q \stackrel{q,p}{\equiv} \begin{cases} \bigoplus_{i=1}^N \begin{pmatrix} \beta_i & 0 \\ 0 & \beta_i \end{pmatrix} & \textbf{(bosons)} \\ \bigoplus_{i=1}^N \begin{pmatrix} 0 & \beta_i \\ -\beta_i & 0 \end{pmatrix} & \textbf{(fermions)} \end{cases}, \tag{178}$$

which follows for bosons from the well-known Williamson's theorem [44] and for fermions from the block-diagonalization of anti-symmetric matrices under orthogonal transformations. This allows us to write

$$\rho_{(J,z)} \equiv \frac{e^{-2\sum_{i=1}^N \beta_i \hat{n}_i}}{\text{Tr} \exp(-2\sum_{i=1}^N \beta_i \hat{n}_i)}, \tag{179}$$

where the 2 in the exponent is convention. From (179), we can read off the spectrum as the diagonal form of $\rho$ is

$$\rho_{(J,z)} \equiv \sum_{n_1 \ldots n_N} \lambda_{n_1} \ldots \lambda_{n_N} |n_1, \ldots, n_N; v\rangle \langle n_1, \ldots, n_N; v|, \tag{180}$$

where we can read off the eigenvalues from (179) as

$$\lambda_{n_i} = \begin{cases} (1 - e^{-2\beta_i})e^{-2\beta_i n_i} & \textbf{(bosons)} \\ (1 + e^{-2\beta_i})^{-1}e^{-2\beta_i n_i} & \textbf{(fermions)} \end{cases}. \tag{181}$$

---

[16]We could extend the fermionic definition to include Grassmann displacements $z^a$ by having $\hat{Q} = iq_{ab}(\hat{\xi} - z)^a(\hat{\xi} - z)^b + c_0$.

Table 2: *Overview of notations for operator bases.* When treating bosonic or fermionic systems, there are two types of standard bases, namely the real one (bosonic quadrature operators, fermionic Majorana operators) and the complex one (creation/annihilation operators). In our unified notation, we use $\hat{\xi}$ independent of any basis, but will present many examples in both the real basis (indicated by $\overset{q,p}{\equiv}$) and the complex basis (indicated by $\overset{a,a\dagger}{\equiv}$).

|  | **Real basis** | **Complex basis** |
|---|---|---|
| **Bosons** | Quadratures $(\hat{q}_j, \hat{p}_k)$ <br> *Also:* $(\hat{x}_j, \hat{p}_k)$ | CCR operators $(\hat{b}_j, \hat{b}_k^\dagger)$ <br> *Also:* $(\hat{a}_j, \hat{a}_k^\dagger)$ |
| **Fermions** | Majorana modes $\hat{m}_a$ <br> *Also:* $\gamma_a$, $c_a$, $(c_j, \tilde{c}_k)$ | CAR operators $(\hat{f}_j, \hat{f}_k^\dagger)$ <br> *Also:* $(\hat{c}_j, \hat{c}_k^\dagger)$ |
| **Unified** | $\hat{\xi}^a \overset{q,p}{\equiv} (\hat{q}_j, \hat{p}_k)$ | $\hat{\xi}^a \overset{a,a\dagger}{\equiv} (\hat{a}_j, \hat{a}_k^\dagger)$ |

Table 3: *Mixed Gaussian states.* We list the standard forms of $J$, $G$, $\Omega$, $q$ and $c$ for a mixed Gaussian state $\rho_{(J,z)} = e^{-c-q_{ab}(\hat{\xi}_A - z_A)^a(\hat{\xi}_A - z_A)^b}$. This table matches the one in [25].

|  | **Bosons** | **Fermions** |
|---|---|---|
| $\rho$ | $\bigotimes_{i=1}^{N_A} \left( \dfrac{e^{-2\hat{n}_i \ln \coth r_i}}{\cosh r_i \sinh r_i} \right)$ | $\bigotimes_{i=1}^{N_A} \left( \cos r_i \sin r_i \, e^{-2\hat{n}_i \ln \tan r_i} \right)$ |
| $J \overset{q,p}{\equiv}$ | $\bigoplus_{i=1}^{N_A} \begin{pmatrix} 0 & \cosh 2r_i \\ -\cosh 2r_i & 0 \end{pmatrix}$ | $\bigoplus_{i=1}^{N_A} \begin{pmatrix} 0 & \cos 2r_i \\ -\cos 2r_i & 0 \end{pmatrix}$ |
| $J \overset{a,a\dagger}{\equiv}$ | $\bigoplus_{i=1}^{N_A} \begin{pmatrix} -i\cosh 2r_i & 0 \\ 0 & i\cosh 2r_i \end{pmatrix}$ | $\bigoplus_{i=1}^{N_A} \begin{pmatrix} -i\cos 2r_i & 0 \\ 0 & i\cos 2r_i \end{pmatrix}$ |
| $G \overset{q,p}{\equiv}$ | $\bigoplus_{i=1}^{N_A} \begin{pmatrix} \cosh 2r_i & 0 \\ 0 & \cosh 2r_i \end{pmatrix}$ | $\bigoplus_{i=1}^{N_A} \begin{pmatrix} 1 & 0 \\ 0 & 1 \end{pmatrix}$ |
| $G \overset{a,a\dagger}{\equiv}$ | $\bigoplus_{i=1}^{N_A} \begin{pmatrix} 0 & \cosh 2r_i \\ \cosh 2r_i & 0 \end{pmatrix}$ | $\bigoplus_{i=1}^{N_A} \begin{pmatrix} 0 & 1 \\ 1 & 0 \end{pmatrix}$ |
| $\Omega \overset{q,p}{\equiv}$ | $\bigoplus_{i=1}^{N_A} \begin{pmatrix} 0 & 1 \\ -1 & 0 \end{pmatrix}$ | $\bigoplus_{i=1}^{N_A} \begin{pmatrix} 0 & \cos 2r_i \\ -\cos 2r_i & 0 \end{pmatrix}$ |
| $\Omega \overset{a,a\dagger}{\equiv}$ | $\bigoplus_{i=1}^{N_A} \begin{pmatrix} 0 & -i \\ i & 0 \end{pmatrix}$ | $\bigoplus_{i=1}^{N_A} \begin{pmatrix} 0 & -i\cos 2r_i \\ i\cos 2r_i & 0 \end{pmatrix}$ |
| $q \overset{q,p}{\equiv}$ | $\bigoplus_{i=1}^{N_A} \begin{pmatrix} \ln \coth r_i & 0 \\ 0 & \ln \coth r_i \end{pmatrix}$ | $\bigoplus_{i=1}^{N_A} \begin{pmatrix} 0 & \ln \tan r_i \\ -\ln \tan r_i & 0 \end{pmatrix}$ |
| $q \overset{a,a\dagger}{\equiv}$ | $\bigoplus_{i=1}^{N_A} \begin{pmatrix} 0 & \ln \coth r_i \\ \ln \coth r_i & 0 \end{pmatrix}$ | $\bigoplus_{i=1}^{N_A} \begin{pmatrix} 0 & i\ln \tan r_i \\ -i\ln \tan r_i & 0 \end{pmatrix}$ |
| $c$ | $\sum_{i=1}^{N} \log \left( \cosh r_i \sinh r_i \right)$ | $-\sum_{i=1}^{N} \log \left( \cos r_i \sin r_i \right)$ |

This equation implies that mixed Gaussian states $\rho_{(J,z)}$ have a very particular spectrum constructed from powers of $e^{-\beta_i}$. This type of spectrum is called *Gaussian spectrum* and if we find a mixed state $\rho$ with such spectrum for appropriately chosen $\beta_i$, we can always find a Gaussian state $\rho_{(J,z)}$ and a unitary $U$, such that $\rho = U^\dagger \rho_{(J,z)} U$.

For bosons, we compute the 1-point correlation function to be

$$z^a = \mathrm{Tr}(\rho_{(J,z)}\hat{\xi}^a), \tag{182}$$

*i.e.*, the $z^a$ appearing in the definition $\rho$ is indeed its 1-point function. For both bosons and fermions, we can compute the 2-point correlation function

$$C_2^{ab} = \mathrm{Tr}\left(\rho_{(J,z)}(\hat{\xi}-z)^a(\hat{\xi}-z)^b\right) = \frac{1}{2}(G^{ab}+\mathrm{i}\Omega^{ab}), \tag{183}$$

*i.e.*, we perform just the same decomposition as for pure Gaussian states. Using the explicit form (179) for $\rho$, we can compute the respective bosonic and fermionic covariance matrix to be

$$\begin{aligned}
G &\overset{q,p}{\equiv} \bigoplus_{i=1}^{N} \begin{pmatrix} \coth\beta & 0 \\ 0 & \coth\beta \end{pmatrix}, \quad \textbf{(bosons)} \\
\Omega &\overset{q,p}{\equiv} \bigoplus_{i=1}^{N} \begin{pmatrix} 0 & \tanh\beta \\ -\tanh\beta & \end{pmatrix}. \quad \textbf{(fermions)}
\end{aligned} \tag{184}$$

Recall our definition (176), which only is the same for fermions and bosons if the Gaussian state is pure, *i.e.*, $J^2 = -\mathbb{1}$. We can use the explicit forms of $q$ from (178) and of the covariance matrices in (184) to deduce the covariant relation

$$J = \begin{cases} -\cot\Omega q = -\mathrm{i}\coth\mathrm{i}\Omega q & \textbf{(bosons)} \\ +\tan Gq = -\mathrm{i}\tanh\mathrm{i}Gq & \textbf{(fermions)} \end{cases}, \tag{185}$$

where the respective functions are applied as matrix functions, as explained in appendix A.2.

We see that the complex structure $J$ of the mixed Gaussian state characterized by $q_{ab}$ is computed from the Lie algebra generator

$$K = \begin{cases} \Omega q & \textbf{(bosons)} \\ Gq & \textbf{(fermions)} \end{cases}. \tag{186}$$

The mixed Gaussian state $\rho_{(J,z)}$ becomes pure in the limit where the eigenvalues of $K_q$ diverge, such that the eigenvalues of $J$ approach $\pm\mathrm{i}$. It is this limit, in which the density operator $\rho_{(J,z)}$ becomes a projector onto the ground state $|J,z\rangle$ of $\hat{Q}$.

A mixed state complex structure $J$ is characterized by the property that its eigenvalues appear in conjugate pairs $\pm\mathrm{i}\lambda_i$ with $\lambda_i \in [0,\infty)$ for bosons and $\lambda_i \in [0,1]$ for fermions. The choice of a mixed Gaussian state therefore corresponds to equipping the classical phase space with a metric $G$ and a symplectic form $\Omega$ that potentially violate the Kähler condition (6), *i.e.*, they do not give rise to a proper linear complex structure $J$ with $J^2 = -\mathbb{1}$. Instead, the more the eigenvalues of $J$ defined in (176) depart from $\pm\mathrm{i}$, the more mixed will the corresponding state $\rho_{(J,z)}$ be. From a geometric perspective, we can therefore think of mixed Gaussian states as equipping the classical phase space with specifically incompatible Kähler structures $(G,\Omega,J)$, where we have $-J^2 \geq \mathbb{1}$ for bosons and $-J^2 \leq \mathbb{1}$ for fermions. It is exactly the intersection of these two sectors that describes pure Gaussian states. Compatible Kähler structures $(G,\Omega,J)$ in this set can describe both, a bosonic or fermionic Gaussian state. Interestingly, the two

sectors (mixed bosonic Gaussian states vs. mixed fermionic Gaussian states) are related under the duality transformation

$$J \quad \Leftrightarrow \quad -J^{-1}, \tag{187}$$

which maps mixed bosonic complex structures onto fermionic ones and vice versa[17]. This relation can be used to relate the spectrum of bosonic and fermionic mixed states (and thus their entanglement) in supersymmetric systems [45].

**Example 5.** *We consider a single bosonic mode, for which the most general positive-definite quadratic Hamiltonian is characterized by*

$$q \overset{q,p}{\equiv} \beta \begin{pmatrix} \cosh\rho - \cos\phi \sinh\rho & -\sin\phi \sinh\rho \\ -\sin\phi \sinh\rho & \cosh\rho + \cos\phi \sinh\rho \end{pmatrix} \overset{a,a^\dagger}{\equiv} \beta \begin{pmatrix} -e^{i\phi} \sinh\rho & \cosh\rho \\ \cosh\rho & e^{-i\phi} \sinh\rho \end{pmatrix}. \tag{188}$$

*Using formula (185), we can deduce the respective mixed state complex structure $J$ and the associated covariance matrix $G$ to be given by*

$$J \overset{q,p}{\equiv} \coth\beta \begin{pmatrix} -\sin\phi \sinh\rho & \cos\phi \sinh\rho + \cosh\rho \\ \cos\phi \sinh\rho - \cosh\rho & \sin\phi \sinh\rho \end{pmatrix} \overset{a,a^\dagger}{\equiv} \coth\beta \begin{pmatrix} -i\cosh\rho & ie^{i\phi} \sinh\rho \\ -ie^{-i\phi} \sinh\rho & i\cosh\rho \end{pmatrix}, \tag{189}$$

$$G \overset{q,p}{\equiv} \coth\beta \begin{pmatrix} \cosh\rho + \cos\phi \sinh\rho & \sin\phi \sinh\rho \\ \sin\phi \sinh\rho & \cosh\rho - \cos\phi \sinh\rho \end{pmatrix} \overset{a,a^\dagger}{\equiv} \coth\beta \begin{pmatrix} e^{i\phi} \sinh\rho & \cosh\rho \\ \cosh\rho & -e^{-i\phi} \sinh\rho \end{pmatrix}, \tag{190}$$

*For a single mode, covariance matrix and complex structure are proportional to the ones of a pure state, but rescaled with $\coth\beta$, which approaches 1 for $\beta \to \infty$. For mixed states of several modes, each eigenvalue pair in $J$ is appropriately rescaled by a factor $\coth\beta_i$. Requiring that $\hat{Q}$ must be bounded from below implies that $\beta \in (0, \infty)$, such that the mixed Gaussian state becomes more and more mixed in the limit $\beta \to 0$, but there is no maximally mixed state in an infinite Hilbert space. The manifold of mixed bosonic Gaussian states of a single mode (assuming $z^a = 0$ here) is diffeomorphic to a three-dimensional half-space, i.e., $\mathbb{R}^2 \times \mathbb{R}_{\geq 0}$, where the boundary plane represents pure states with $\beta \to \infty$.*

**Example 6.** *We consider a single fermionic mode. The most general quadratic Hamiltonian is here given by*

$$q \equiv \begin{pmatrix} 0 & \beta \\ -\beta & 0 \end{pmatrix}. \tag{191}$$

*The resulting mixed Gaussian state $\rho = e^{\hat{Q}}$ is characterized by the following complex structure $J$ and covariance matrix $\Omega$:*

$$J \overset{q,p}{\equiv} \tanh\beta \begin{pmatrix} 0 & 1 \\ -1 & 0 \end{pmatrix} \overset{a,a^\dagger}{\equiv} \tanh\beta \begin{pmatrix} -i & 0 \\ 0 & i \end{pmatrix}, \tag{192}$$

$$\Omega_\pm \overset{q,p}{\equiv} \tanh\beta \begin{pmatrix} 0 & 1 \\ -1 & 0 \end{pmatrix} \overset{a,a^\dagger}{\equiv} \tanh\beta \begin{pmatrix} 0 & -i \\ i & 0 \end{pmatrix}. \tag{193}$$

*In contrast to bosons, we can choose the parameter $\beta \in \mathbb{R}$, as the respective $\hat{Q}$ will always be bounded from below. Choosing $\beta = 0$ corresponds to the maximally mixed state in the fermionic Hilbert space with $J = 0$. This shows that the family of mixed Gaussian states connects the two*

---

[17]For fermions, there exist complex structures $J$ with vanishing eigenvalues that are mapped to infinity under this duality. This relates a maximally mixed fermionic mode to a maximally mixed bosonic mode, which only makes sense in the limit, as the bosonic Hilbert space is infinite dimensional.

*parity sectors of pure Gaussian states, as every mixed Gaussian state can be connected to the maximally mixed state by rescaling $J \to 0$. The manifold of mixed fermionic Gaussian states of a single bosonic mode is diffeomorphic to an interval, i.e., $[-1, 1]$, where the boundary points represent the two fermionic Gaussian states of different parity.*

The geometry of mixed Gaussian state is more intricate than the one of pure Gaussian states. Generically, all eigenvalue pairs $\pm i\lambda_i$ of $J$ will be different, such that the subgroup

$$\mathrm{Sta}_J = \{M \in \mathcal{G} \,|\, MJM^{-1} = J\} \tag{194}$$

is isomorphic to $\mathrm{U}(1)^{\otimes N}$. More specifically, if we have $s$ distinct eigenvalue pairs $\pm i\lambda_i$ with degeneracies $d_i$, the stabilizer subgroup of $J$ is isomorphic to

$$\mathrm{Sta}_J = \bigoplus_{i=1}^{s} \mathrm{U}(d_i), \tag{195}$$

such that $\sum_{i=1}^{s} d_i = N$. One can repeat the same arguments as in section 2.2.6 to find that $\mathcal{M}_{b/f} = \mathcal{G}/\mathrm{Sta}_J$, which consists of all mixed Gaussian states characterized by the respective spectrum of $\lambda_i$ and their degeneracies. The full manifold can be foliated by $\mathcal{M}_{b/f}$ to form the manifold $\mathcal{M}_{\mathrm{mixed}}$ of mixed Gaussian states with

$$\dim \mathcal{M}_{\mathrm{mixed}} = \begin{cases} N(2N+1) + 2N & \textbf{(bosons)} \\ N(2N-1) & \textbf{(fermions)} \end{cases}. \tag{196}$$

This manifold has a complicated boundary consisting of various lower dimensional surfaces, corners etc. In particular, pure Gaussian states form a small corner of this manifold, just like pure quantum states form a small corner of the convex set of mixed states.

### 3.2.2 Characteristic function

We introduce the characteristic function of an operator $\mathcal{O}$ given by

$$\chi(w) = \begin{cases} \mathrm{Tr}(\mathcal{O} e^{-i w_a \hat{\xi}^a}) & \textbf{(bosons)} \\ \mathrm{Tr}(\mathcal{O} e^{-w_a \hat{\xi}^a}) & \textbf{(fermions)} \end{cases}, \tag{197}$$

which is defined for both bosonic and fermionic systems. For fermionic systems, $w_a$ is Grassmann valued, which anti-commutes with itself and with linear operators $\hat{\xi}^a$, i.e., we have $\{w_a, w_b\} = \{w_a, \hat{\xi}^b\} = 0$. Note that $e^{-w_a \hat{\xi}^a}$ behaves similar to a fermionic displacement operator from (70), such that $e^{w_a \hat{\xi}^a} \hat{\xi}^a e^{-w_a \hat{\xi}^a} = \hat{\xi}^a + G^{ab} w_b$ with $w_a$ being Grassmann-valued.

Let us further discuss an important subtlety about traces of $e^{\widehat{K}}$ for fermions. If we consider the fermionic operator $e^{\widehat{w}}$ satisfying $e^{-\widehat{w}} \hat{\xi}^a e^{\widehat{w}} = \hat{\xi}^a + G^{ab} w_b$, we find

$$\mathrm{Tr}(e^{-\widehat{w}} e^{\widehat{K}} e^{\widehat{w}}) = \mathrm{Tr}(e^{\widehat{K}}) e^{w_a \left(\tanh \frac{K}{2}\right)^a{}_b G^{bc} w_c}, \qquad \textbf{(fermions)} \tag{198}$$

which can be derived by block-diagonalizing $K$ and then expanding $e^{\pm \widehat{w}}$ for individual degrees of freedom.

With this in hand, we can compute the characteristic function $\chi(w)$ of general mixed Gaussian states.

**Proposition 10.** *The characteristic function of a mixed Gaussian state $\rho_{(J,z)}$ is given by*

$$\chi(w) = \begin{cases} e^{-\frac{1}{4}w_a G^{ab} w_b - iw_a z^a} & \textbf{(bosons)} \\ e^{-\frac{i}{4}w_a \Omega^{ab} w_b} & \textbf{(fermions)} \end{cases}, \tag{199}$$

*where $G$ and $\Omega$ are the respective covariance matrices.*

*Proof.* We consider bosons and fermions separately.
**Bosons.** We have

$$\chi(w) = \mathrm{Tr}\big(\frac{e^{-(\hat{\xi}-z)^a q_{ab}(\hat{\xi}-z)^b}}{Z} e^{-iw_a \hat{\xi}^a}\big) = \mathrm{Tr}\big(\frac{e^{-\hat{\xi}^a q_{ab}\hat{\xi}^b}}{Z} e^{-iw_a(\hat{\xi}+z)^a}\big), \tag{200}$$

where we used the displacement operators satisfying $\mathcal{D}^\dagger(z)\hat{\xi}^a \mathcal{D}(z) = (\hat{\xi}+z)^a$ to apply a shift to the whole expression without changing its trace. We can define $K = -2i\Omega q$, such that $\widehat{K} = -q_{ab}\hat{\xi}^a \hat{\xi}^b$ which is Hermitian and thus represents a complexified algebra element. This allows us to write

$$\chi(w) = \frac{e^{-iw_a z^a}}{Z}\mathrm{Tr}(e^{\widehat{K}} e^{\widehat{w}}) = \frac{e^{-iw_a z^a}}{Z}\mathrm{Tr}(e^{\widehat{K}+\widehat{\eta}+w_a B^{ab} w_b}), \tag{201}$$

where we applied (165). The exponent reads

$$-q_{ab}\hat{\xi}^a \hat{\xi}^b - i\eta_a \hat{\xi}^a + w_a B^{ab} w_b, \tag{202}$$

where we can complete the square to rewrite it as

$$-q_{ab}(\hat{\xi}-y)^a(\hat{\xi}-y)^b + \underbrace{q_{ab}y^a y^b + w_a B^{ab} w_b}_{=:w_a \tilde{B}^{ab} w_b}, \tag{203}$$

where we have $y^a = \frac{i}{2}Q^{ab}\eta_b$ with $Q^{ab} = (q^{-1})^{ab}$, which is invertible for a mixed Gaussian state. Using the explicit form (166) of $\eta$ in terms of $w$, we find

$$\chi(w) = e^{-iw_a z^a + w_a \tilde{B}^{ab} w_b}\mathrm{Tr}\big(\frac{e^{-q_{ab}(\hat{\xi}-y)^a(\hat{\xi}-y)^b}}{Z}\big) = e^{-iw_a z^a + w_a \tilde{B}^{ab} w_b}, \tag{204}$$

where we used that the shift in $y^a$ does not change the trace. More precisely, we argue that

$$\mathrm{Tr}\big(\frac{e^{-q_{ab}(\hat{\xi}-y)^a(\hat{\xi}-y)^b}}{Z}\big) = \mathrm{Tr}\big(\mathcal{D}^{-1}\frac{e^{-q_{ab}(\hat{\xi}-y)^a(\hat{\xi}-y)^b}}{Z}\mathcal{D}\big) = \mathrm{Tr}\big(\frac{e^{-q_{ab}(\hat{\xi})^a(\hat{\xi})^b}}{Z}\big) = \mathrm{Tr}\,\rho = 1, \tag{205}$$

where we used that the operator $\mathcal{D} = e^{-iy^a \omega_{ab}\hat{\xi}^b}$ with $\mathcal{D}^{-1}\hat{\xi}^a \mathcal{D} = \hat{\xi}^a + y^a$ does not change the trace[18] (which will turn out to be not true for fermions!). The new bilinear form $\tilde{B}$ from (204) is

$$\begin{aligned} \tilde{B}^{ab} &= -\frac{1}{4}\left(\frac{K}{e^K-\mathbb{1}}Q\frac{K^\mathsf{T}}{e^{K^\mathsf{T}}-\mathbb{1}} - i\frac{K-\sinh K}{\mathbb{1}-\cosh K}\Omega\right)^{ab} \\ &= -\frac{i}{4}\left(\frac{2K}{(e^K-\mathbb{1})(e^{-K}-\mathbb{1})} - \frac{K-\sinh K}{\mathbb{1}-\cosh K}\right)^a{}_c \Omega^{cb} \\ &= \frac{i}{4}\coth(-i\Omega q)^a{}_c \Omega^{cb} = \frac{1}{4}(J\Omega)^{ab} = -\frac{1}{4}G^{ab}, \end{aligned} \tag{206}$$

where we used $KQ = -2i\Omega$ and $\Omega K^\mathsf{T} = -K\Omega$ in the second step, combined the functions to find $\coth(K/2)$ and then used the expressions (185) to express everything in terms of $J$ and eventually $G$.

---

[18]Note that $y^a$ is a vector in $V_\mathbb{C}$, such that $\mathcal{D}$ is not a unitary displacement operator satisfying $\mathcal{D}^\dagger = \mathcal{D}^{-1}$. However, our argument does not require this.

**Fermions.** The derivation for fermions follows the one for bosons closely with $\widehat{K} = -\mathrm{i}q_{ab}\hat{\xi}^a\hat{\xi}^b$, but we now have $z^a = 0$ and $w_a$ is a Grassmann number. We can largely follow the same strategy, but need to replace $\mathrm{i}w_a \to w_a$, $q_{ab} \to \mathrm{i}q_{ab}$ and $Q^{ab} \to -\mathrm{i}Q^{ab}$. With this, we arrive at the analogue of (204) given by

$$\chi(w) = e^{w_a \tilde{B}^{ab} w_b} \operatorname{Tr}\left(\frac{e^{-\mathrm{i}q_{ab}(\hat{\xi}-y)^a(\hat{\xi}-y)^b}}{Z}\right), \tag{207}$$

where we use $K = -2\mathrm{i}Gq$ and $QK^{\mathsf{T}} = 2\mathrm{i}G$ to get

$$
\begin{aligned}
\tilde{B}^{ab} &= \frac{1}{4}\left(\frac{K}{e^K-\mathbb{1}}\mathrm{i}Q\frac{K^{\mathsf{T}}}{e^{K^{\mathsf{T}}-\mathbb{1}}} + \frac{K-\sinh K}{\mathbb{1}-\cosh K}G\right)^{ab} = \frac{1}{4}\left(\frac{-2K}{(e^K-\mathbb{1})(e^{-K}-\mathbb{1})} + \frac{K-\sinh K}{\mathbb{1}-\cosh K}\right)^a{}_c G^{cb} \\
&= \frac{1}{4}\coth(\tfrac{K}{2})^a{}_c G^b \,.
\end{aligned} \tag{208}
$$

As discussed around in the context of (198), we have

$$\operatorname{Tr}\left(\frac{e^{-\mathrm{i}q_{ab}(\hat{\xi}-y)^a(\hat{\xi}-y)^b}}{Z}\right) = e^{y^a g_{ab}\left(\tanh\frac{K}{2}\right)^b{}_c y^c} = e^{-\frac{1}{2}w_a \sinh^{-1}(K)^a{}_c G^{cb} w_b}, \tag{209}$$

where we used $y^a = \frac{1}{2}Q^{ab}\eta_b = \frac{1}{2}Q^{ab}\left(\frac{K^{\mathsf{T}}}{e^{K^{\mathsf{T}}}-\mathbb{1}}\right)_b{}^c w_c$. Consequently, we can combine the different terms to find $\chi(w) = e^{w_a(\tilde{B}+\tilde{C})^{ab}w_b}$ with

$$(\tilde{B}+\tilde{C})^{ab} = \frac{1}{4}(\coth\tfrac{K}{2} - 2\sinh^{-1}K)^a{}_c G^{cb} = \frac{1}{4}\tanh(-\mathrm{i}Gq)^a{}_c G^{cb} = \tfrac{1}{4}(-\mathrm{i}JG)^{ab} = -\tfrac{\mathrm{i}}{4}\Omega^{ab}, \tag{210}$$

where we followed the same strategy as for bosons to finally arriv at $\chi(w)$ from (199). $\qquad\square$

Characteristic functions are closely related to quasi-probability distribution on the classical phase space, which can be used as an alternative description of the quantum theory. Translation recipes to describe Gaussian states with such quasi-probability distributions can be found in [25]. However, phase space distributions can also be used to describe general quantum states and allow for more efficient calculations in certain settings [46,47], such as boson sampling.

### 3.2.3 Wick's theorem

In the previous section, we derived the representation of mixed Gaussian states as characteristic functions $\chi(w)$ defined on the dual phase space. This will enable us to prove Wick's theorem for mixed Gaussian states.

**Proposition 11.** *Given a mixed Gaussian state $\rho_{(J,\hat{z})}$, a general $n$-point function $C_n^{a_1\dots a_n}$ is computed in the same way as for pure Gaussian states, as explained in proposition 8. The 2-point function $C_2^{ab} = \frac{1}{2}(G^{ab} + \mathrm{i}\Omega^{ab})$ is related to the mixed state complex structure $J$ via*

$$
\begin{aligned}
G^{ab} &= -J^a{}_c \Omega^{cb} \quad \textbf{(bosons)} \\
\Omega^{ab} &= J^a{}_c G^{cb} \quad \textbf{(fermions)}
\end{aligned}, \tag{211}
$$

*where $\Omega$ for bosons and $G$ for fermions is fixed.*

*Proof.* We recall the definition of the characteristic function (197), where $w_a$ is Grassmann-valued for fermions. If we define the derivative operator

$$\mathcal{F}_w^{a_1\dots a_n} = \begin{cases} \left(\frac{\mathrm{i}\partial}{\partial w_{a_1}}\right)\cdots\left(\frac{\mathrm{i}\partial}{\partial w_{a_n}}\right) & \textbf{(bosons)} \\ \left(\frac{\partial}{\partial w_{a_1}}\right)\cdots\left(\frac{\partial}{\partial w_{a_n}}\right) & \textbf{(fermions)} \end{cases}, \tag{212}$$

we find

$$
\mathcal{F}_w^{a_1\ldots a_n}\, \chi(w)\big|_{w=0} =
\begin{cases}
\mathrm{Tr}\left(\rho\, \hat{\xi}^{(a_1}\ldots \hat{\xi}^{a_n)}\right) & \textbf{(bosons)} \\
\mathrm{Tr}\left(\rho\, \hat{\xi}^{[a_1}\ldots \hat{\xi}^{a_n]}\right) & \textbf{(fermions)}
\end{cases},
\tag{213}
$$

where $\hat{\xi}^{(a_1}\ldots \hat{\xi}^{a_n)} = \mathrm{SYM}(\hat{\xi}^{a_1}\ldots \hat{\xi}^{a_n})$ represents the totally symmetrized and $\hat{\xi}^{[a_1}\ldots \hat{\xi}^{a_n]} = \mathrm{ASYM}(\hat{\xi}^{a_1}\ldots \hat{\xi}^{a_n})$ represents the totally anti-symmetrized tensor, $e.g.$, $\hat{\xi}^{(a}\hat{\xi}^{b)} = \frac{1}{2}(\hat{\xi}^a\hat{\xi}^b + \hat{\xi}^b\hat{\xi}^a)$ and $\hat{\xi}^{[a}\hat{\xi}^{b]} = \frac{1}{2}(\hat{\xi}^a\hat{\xi}^b - \hat{\xi}^b\hat{\xi}^a)$ etc.

When we apply (213) to the characteristic functions derived in (199), we find that $C_n^{(a_1\ldots a_n)}$ for bosons and $C_n^{[a_1\ldots a_n]}$ satisfy Wick's theorem for $C_2^{(ab)} = \frac{1}{2}G^{ab}$ and $C_2^{[ab]} = \frac{i}{2}\Omega^{ab}$, respectively. Note that for bosons, the displacement of $z^a$ is automatically removed from $C_n^{(a_1\ldots a_n)}$ by the linear term $-iw_a z^a$ in the exponential. Finally, if we are interested in computing the regular ($i.e.$, neither symmetrized nor anti-symmetrized) $n$-point correlation functions, we just need to commute or anti-commute the respective terms of $\hat{\xi}^{a_i}$ in the symmetrized or anti-symmetrized expressions, which will yield additional commutators $i\Omega^{ab}$ for bosons and anti-commutators $G^{ab}$ for bosons, such that $C_n^{a_1\ldots a_n}$ will satisfy Wick's theorem in the same way as pure states with 2-point function $C_2 = \frac{1}{2}(G^{ab} + i\Omega^{ab})$. $\qquad\square$

We found that $n$-point correlation functions for mixed Gaussian states are computed in the same way as for pure Gaussian states via Wick's theorem. The only difference is that the respective $J$ does not satisfy $J^2 = -\mathbb{1}$, which can be used distinguish pure and mixed Gaussian states. Next, we will see how this relation can be used to show that mixed Gaussian states arise when we reduce pure Gaussian states to subsystems.

# 4 Applications

The goal of this section is to demonstrate how the formalism of Kähler structures can be used for applications in quantum information and non-equilibrium physics.

## 4.1 Entanglement and complexity

We derive a number of compact formulas to describe quantum-information properties, such as entanglement and complexity, of Gaussian states in terms of their complex structure $J$. While Gaussian states have been heavily used in quantum information [3,48,49], so far Kähler structures have been rarely used to describe their properties.

### 4.1.1 Algebraic definition of a subsystem

The observables of a quantum system form an algebra $\mathcal{A}$, given by the Weyl algebra $\mathrm{Weyl}(V^*, \Omega)$ in the bosonic case and by the Clifford algebra $\mathrm{Cliff}(V^*, G)$ in the fermionic case.

A subalgebra $\mathcal{A}_A \subset \mathcal{A}$ defines a subsystem $A$ in terms of its observables. In general, the subsystem $A$ and its complement $B$ share a set of observables, corresponding to the fact that the subalgebra $\mathcal{A}_A$ has a center in $\mathcal{A}$. We identify sufficient conditions for the absence of a center.

The set of observables that commute with all elements of $\mathcal{A}_A$, $i.e.$, its commutant, define a subsystem $B$ with algebra

$$
\mathcal{A}_B = \left\{ b \in \mathcal{A} \,\middle|\, [b, a] = 0 \,\forall\, a \in \mathcal{A}_A \right\}.
\tag{214}
$$

Table 4: *Gaussian states.* This table summarizes and compares our methods to describe bosonic and fermionic Gaussian states using Kähler structures covered in section 3.

| structure | bosons | fermions |
|---|---|---|
| 1-point function | $z^a = \langle\psi|\hat{\xi}^a|\psi\rangle = \mathrm{Tr}(\rho\,\hat{\xi}^a)$ | Requirement: $z^a = \langle\psi|\hat{\xi}^a|\psi\rangle = \mathrm{Tr}(\rho\,\hat{\xi}^a) = 0$ |
| 2-point function | $\begin{aligned}C_2^{ab} &= \langle\psi|(\hat{\xi}-z)^a(\hat{\xi}-z)^b|\psi\rangle\\ &= \mathrm{Tr}\,\rho(\hat{\xi}-z)^a(\hat{\xi}-z)^b\end{aligned}$ | $C_2^{ab} = \langle\psi|\hat{\xi}^a\hat{\xi}^b|\psi\rangle = \mathrm{Tr}(\rho\,\hat{\xi}^a\hat{\xi}^b)$ |
| decomposition | $C_2^{ab} = \frac{1}{2}(G^{ab} + \mathrm{i}\Omega^{ab})$ | |
| covariance matrix $\Gamma^{ab}$ | $\begin{aligned}\Gamma^{ab} = G^{ab} &= \langle\psi|\hat{\xi}^a\hat{\xi}^b + \hat{\xi}^b\hat{\xi}^a|\psi\rangle - 2z^az^b\\ &= \mathrm{Tr}\,\rho(\hat{\xi}^a\hat{\xi}^b + \hat{\xi}^b\hat{\xi}^a) - 2z^az^b\end{aligned}$ | $\begin{aligned}\Gamma^{ab} = \Omega^{ab} &= \langle\psi|\hat{\xi}^a\hat{\xi}^b - \hat{\xi}^b\hat{\xi}^a|\psi\rangle\\ &= \mathrm{Tr}\,\rho(\hat{\xi}^a\hat{\xi}^b - \hat{\xi}^b\hat{\xi}^a)\end{aligned}$ |
| relation to $J$ | $\Gamma^{ab} = -J^a{}_c\Omega^{cb}$ | $\Gamma^{ab} = J^a{}_c G^{cb}$ |
| pure Gaussian $|J,z\rangle$ | $\frac{1}{2}(\delta^a{}_b - \mathrm{i}J^a{}_b)(\hat{\xi}^b - z^b)|J,z\rangle = 0$ with $J^2 = -\mathbb{1}$ | |
| dimension | $N(N+1)$ plus $2N$ displacements | $N(N-1)$ |
| covariant ladder operators | $\hat{\xi}^a_\pm = \frac{1}{2}(\delta^a{}_b \mp \mathrm{i}J^a{}_b)(\hat{\xi}^b - z^b)$ with $J^a{}_b\hat{\xi}^b_\pm = \pm\mathrm{i}\hat{\xi}^a_\pm$ | |
| | $[\hat{\xi}^a_\pm, \hat{\xi}^b_\pm] = 0,\ [\hat{\xi}^a_-, \hat{\xi}^b_+] = C_2^{ab}$ | $\{\hat{\xi}^a_\pm, \hat{\xi}^b_\pm\} = 0,\ \{\hat{\xi}^a_-, \hat{\xi}^b_+\} = C_2^{ab}$ |
| $n$-point function | $C_n^{a_1\cdots a_n} = \langle\psi|(\hat{\xi}-z)^{a_1}\cdots(\hat{\xi}-z)^{a_n}|\psi\rangle$ | |
| Wick's theorem | $C_{2n+1} = 0$ and $C_{2n}^{a_1\cdots a_{2n}} = \sum_\sigma \frac{|\sigma|}{n!} C_2^{a_{\sigma(1)}a_{\sigma(2)}} \cdots C_2^{a_{\sigma(2n-1)}a_{\sigma(2n)}}$ | |
| normal-ordered squeezing (explicit) | $\begin{aligned}&e^{\frac{r}{2}(e^{\mathrm{i}\theta}(\hat{a}^\dagger)^2 - e^{-\mathrm{i}\theta}\hat{a}^2)} = e^{\frac{1}{2}e^{\mathrm{i}\theta}(\tanh r)(\hat{a}^\dagger)^2}\\ &\times e^{-(\ln\cosh r)(\hat{n}+\frac{1}{2})} e^{-\frac{1}{2}(e^{-\mathrm{i}\theta}\tanh r)\hat{a}^2}\end{aligned}$ | $\begin{aligned}&e^{r(e^{\mathrm{i}\theta}\hat{a}_1^\dagger\hat{a}_2^\dagger + e^{-\mathrm{i}\theta}\hat{a}_1\hat{a}_2)} = e^{e^{\mathrm{i}\theta}\tan r\,\hat{a}_1^\dagger\hat{a}_2^\dagger}\\ &\times e^{-(\ln\cos r)(\hat{n}_1+\hat{n}_2-1)} e^{-e^{-\mathrm{i}\theta}\tan r\,\hat{a}_1\hat{a}_2}\end{aligned}$ |
| normal-ordered squeezing (covariant) | $\begin{aligned}&e^{\hat{K}_+} = e^{-\frac{1}{2}\omega_{ac}L^c{}_b\hat{\xi}^a_+\hat{\xi}^b_+}\\ &\times e^{-\frac{1}{2}\omega_{ac}\log(\mathbb{1}-L^2)^c{}_b(\hat{\xi}^a_+\hat{\xi}^b_- + \frac{\mathrm{i}}{4}\Omega^{ab})} e^{-\frac{1}{2}\omega_{ac}L^c{}_b\hat{\xi}^a_-\hat{\xi}^b_-}\end{aligned}$ | $\begin{aligned}&e^{\hat{K}_+} = e^{\frac{1}{2}g_{ac}L^c{}_b\hat{\xi}^a_+\hat{\xi}^b_+}\\ &\times e^{\frac{1}{2}g_{ac}\log(\mathbb{1}-L^2)^c{}_b(\hat{\xi}^a_+\hat{\xi}^b_- - \frac{1}{4}G^{ab})} e^{\frac{1}{2}g_{ac}L^c{}_b\hat{\xi}^a_-\hat{\xi}^b_-}\end{aligned}$ |
| normal-ordered displacement (explicit) | $e^{\alpha\hat{a}^\dagger + \beta\hat{a}} = e^{\alpha\hat{a}^\dagger} e^{\alpha\beta/2} e^{\beta\hat{a}}$ | |
| normal-ordered displacement (covariant) | $e^{v_a\hat{\xi}^a_+ + w_b\hat{\xi}^b_-} = e^{v_a\hat{\xi}^a_+} e^{\frac{1}{2}v_a(C_2^\mathsf{T})^{ab}w_b} e^{\hat{\xi}^b_- w_b}$ | |
| normal-ordered middle term (explicit) | $e^{\alpha\hat{a}} e^{\beta(\hat{a}^\dagger)^2} = e^{\beta(\hat{a}^\dagger)^2 + 2\alpha\beta\hat{a}^\dagger} e^{\alpha^2\beta} e^{\alpha\hat{a}}$ | $\begin{aligned}&e^{\alpha_1\hat{a}_1 + \alpha_2\hat{a}_2} e^{\beta\hat{a}_1^\dagger\hat{a}_2^\dagger} = e^{\beta(\hat{a}_1^\dagger\hat{a}_2^\dagger + \alpha_1\hat{a}_2^\dagger - \alpha_2\hat{a}_1^\dagger)}\\ &\times e^{\alpha_1\beta\alpha_2} e^{\alpha_1\hat{a}_1 + \alpha_2\hat{a}_2}\end{aligned}$ |
| normal-ordered middle term (covariant) | $e^{w_a\hat{\xi}^a_-} e^{v_{bc}\hat{\xi}^b_+\hat{\xi}^c_+} = e^{v_{bc}\hat{\xi}^b_+\hat{\xi}^c_+ + 2w_a C_2^{ab}v_{bc}\hat{\xi}^c_+} e^{w_a C_2^{ab}v_{bc}(C_2^\mathsf{T})^{cd}w_d} e^{w_a\hat{\xi}^a_-}$ | |
| combining squeezing and displacement | $\begin{aligned}&\log(e^{\hat{K}}e^{\hat{w}}) = \hat{K} + \hat{\eta} + w_a B^{ab}w_b \text{ with } \eta_a = w_b\left(\frac{K}{e^K-1}\right)^b{}_a,\ F(K) = \frac{1}{4}\frac{K-\sinh K}{1-\cosh K},\\ &\hat{w} = -\mathrm{i}w_a\hat{\xi}^a,\ \hat{K} = -\mathrm{i}\omega_{ac}K^c{}_b\hat{\xi}^a\hat{\xi}^b \text{ and}\\ &B^{ab} = \mathrm{i}F(K)^a{}_c\Omega^{cb}\end{aligned}$ | $\begin{aligned}&\hat{w} = -w_a\hat{\xi}^a,\ \hat{K} = g_{ac}K^c{}_b\hat{\xi}^a\hat{\xi}^b \text{ and}\\ &B^{ab} = F(K)^a{}_c G^{cb}\end{aligned}$ |
| scalar product $|\langle J,z|\tilde{J},\tilde{z}\rangle|^2$ | $\det\left(\frac{\sqrt{2}\Delta^{1/4}}{\sqrt{\mathbb{1}+\Delta}}\right) e^{-\frac{1}{2}(z-\tilde{z})^a(\Gamma+\tilde{\Gamma})^{-1}_{ab}(z-\tilde{z})^b}$ | $\det\left(\frac{\sqrt{\mathbb{1}+\Delta}}{\sqrt{2}\Delta^{1/4}}\right)$ |
| mixed Gaussian $\rho_{(J,z)} = e^{-\hat{Q}}$ | $\hat{Q} = q_{ab}(\hat{\xi}-z)^a(\hat{\xi}-z)^b + c$ | $\hat{Q} = \mathrm{i}q_{ab}\hat{\xi}^a\hat{\xi}^b + c$ |
| dimension | $N(2N+1)$ plus $2N$ displacements | $N(2N-1)$ |
| finding $q$ | $q = -\omega\,\mathrm{arccot}\,J = -\mathrm{i}\omega\,\mathrm{arccoth}\,\mathrm{i}J$ | $q = g\arctan J = -\mathrm{i}g\,\mathrm{arctanh}\,\mathrm{i}J$ |
| finding $J$ | $J = -\cot\Omega q = -\mathrm{i}\coth\mathrm{i}\Omega q$ | $J = \tan Gq = -\mathrm{i}\tanh\mathrm{i}Gq$ |
| finding $c$ | $c = \frac{1}{4}\log\det\left(\frac{\mathbb{1}+J^2}{4}\right)$ | $c = -\frac{1}{4}\log\det\left(\frac{\mathbb{1}+J^2}{4}\right)$ |
| eigenvalues $\pm\mathrm{i}\lambda_i$ of $J$ | $\lambda_i \in [1,\infty)$ | $\lambda_i \in [0,1]$ |
| characteristic function | $\chi(w) = e^{-\frac{1}{4}w_a G^{ab}w_b - \mathrm{i}w_a z^a}$ | $\chi(w) = e^{-\frac{1}{4}w_a\Omega^{ab}w_b}$ |
| $n$-point function | $C_n^{a_1\cdots a_n} = \mathrm{Tr}\left(\rho(\hat{\xi}-z)^{a_1}\cdots(\hat{\xi}-z)^{a_n}\right)$ | |
| Wick's theorem | $C_{2n+1} = 0$ and $C_{2n}^{a_1\cdots a_{2n}} = \sum_\sigma \frac{|\sigma|}{n!} C_2^{a_{\sigma(1)}a_{\sigma(2)}} \cdots C_2^{a_{\sigma(2n-1)}a_{\sigma(2n)}}$ | |

In general, the subsystems $A$ and $B$ have a center

$$\mathcal{Z} = \mathcal{A}_A \cap \mathcal{A}_B. \tag{215}$$

As a result, the Hilbert space of the system decomposes as a direct sum of tensor products [50, 51],

$$\mathcal{H} = \bigoplus_\zeta \left( \mathcal{H}_A(\zeta) \otimes \mathcal{H}_B(\zeta) \right), \tag{216}$$

where the sum is over the spectrum of $\mathcal{Z}$.

Here, we consider subsystems defined by a Weyl algebra $\mathcal{A}_A = \text{Weyl}(V_A^*, \Omega_A)$ in the bosonic case and by a Clifford algebra $\mathcal{A}_A = \text{Cliff}(V_A^*, G_A)$ in the fermionic case. This restriction results in a trivial center $\mathcal{A}_A \cap \mathcal{A}_B = \{\mathbb{1}\}$ and a tensor-product decomposition of the Hilbert space of the system, $\mathcal{H} = \mathcal{H}_A \otimes \mathcal{H}_B$.

### 4.1.2 Subsystem decomposition

Given a bosonic or fermionic system with $N$ degrees of freedom, we can always decompose the classical phase space $V$ into two complementary subsystems $A$ and $B$ with $V = A \oplus B$ satisfying the conditions of section 2.2.4. A decomposition $V = A \oplus B$ into symplectic or orthogonal complements for bosons or fermions, respectively, induces a dual decomposition $V^* = A^* \oplus B^*$. More precisely, we have

$$\begin{aligned} \omega_{ab} \xi_A^a \xi_B^b = 0 \ \forall \ \xi_A \in A, \xi_B \in B, \quad &\textbf{(bosons)} \\ g_{ab} \xi_A^a \xi_B^b = 0 \ \forall \ \xi_A \in A, \xi_B \in B. \quad &\textbf{(fermions)} \end{aligned} \tag{217}$$

We further have $A^* = \{\omega_{ab}\xi_A^b | \xi \in A\}$ and $B^* = \{\omega_{ab}\xi_A^b | \xi \in B\}$ for bosons and $A^* = \{g_{ab}\xi_A^b | \xi \in A\}$ and $B^* = \{g_{ab}\xi_A^b | \xi \in B\}$ for fermions.

Any phase space decomposition $V = A \oplus B$, such that $A$ and $B$ are either symplectic complements for bosonic systems or orthogonal complements for fermionic systems, induces a tensor product decomposition

$$\mathcal{H} = \mathcal{H}_A \otimes \mathcal{H}_B. \tag{218}$$

It is induced by quantizing $A$ and $B$ (with the respective restricted symplectic form $\Omega$ or metric $G$) individually and then naturally identifying tensor products of states with elements in $\mathcal{H}$.

Of course, there are infinitely many other ways, one can write an infinite dimensional Hilbert space as a tensor product of two other infinite dimensional Hilbert spaces. However, for physical applications, we typically use above subsystem definition constructed from a subset $A^* \subset V^*$ of linear observables, which naturally gives rise to the decomposition described above.

**Proposition 12.** *Given a pure Gaussian state $|J, z\rangle$ and a subsystem decomposition $V = A \oplus B$ according to definition 3, we can decompose $J$ according to*

$$J = \left( \begin{array}{c|c} J_A & J_{AB} \\ \hline J_{BA} & J_B \end{array} \right) \quad with \quad \begin{array}{llll} J_A : & A \to A : & a \mapsto \mathbb{P}_A(Ja), \\ J_B : & B \to B : & b \mapsto \mathbb{P}_B(Jb), \\ J_{AB} : & B \to A : & b \mapsto \mathbb{P}_A(Jb), \\ J_{BA} : & A \to B : & a \mapsto \mathbb{P}_B(Ja), \end{array} \tag{219}$$

*where $\mathbb{P}_A$ and $\mathbb{P}_B$ are the respective projections onto $A$ and $B$, respectively, such that $\mathbb{1} = \mathbb{P}_A + \mathbb{P}_B$. We can then always choose the bases $\hat{\xi}_A^a \equiv (\hat{q}_1^A, \hat{p}_1^A, \dots, \hat{q}_{N_A}^A, \hat{p}_{N_A}^A)$ and $\hat{\xi}_B^a \equiv (\hat{q}_1^B, \hat{p}_1^B, \dots, \hat{q}_{N_A}^B, \hat{p}_{N_B}^A)$,*

*such that the linear complex structure is*

$$
J \equiv \left(\begin{array}{ccc|ccc|ccc}
\cosh(2r_1)\mathbb{A}_2 & \cdots & 0 & \sinh(2r_1)\mathbb{S}_2 & \cdots & 0 & 0 & \cdots & 0 \\
\vdots & \ddots & \vdots & \vdots & \ddots & \vdots & \vdots & \ddots & \vdots \\
0 & \cdots & \cosh(2r_{N_A})\mathbb{A}_2 & 0 & \cdots & \sinh(2_{N_A})\mathbb{S}_2 & 0 & \cdots & 0 \\
\hline
\sinh(2r_1)\mathbb{S}_2 & \cdots & 0 & \cosh(2r_1)\mathbb{A}_2 & \cdots & 0 & 0 & \cdots & 0 \\
\vdots & \ddots & \vdots & \vdots & \ddots & \vdots & \vdots & \ddots & \vdots \\
0 & \cdots & \sinh(2r_{N_A})\mathbb{S}_2 & 0 & \cdots & \cosh(2r_{N_A})\mathbb{A}_2 & 0 & \cdots & 0 \\
0 & \cdots & 0 & 0 & \cdots & 0 & \mathbb{A}_2 & \cdots & 0 \\
\vdots & \ddots & \vdots & \vdots & \ddots & \vdots & \vdots & \ddots & \vdots \\
0 & \cdots & 0 & 0 & \cdots & 0 & 0 & \cdots & \mathbb{A}_2
\end{array}\right) \quad \textbf{(bosons)}
$$

$$
J \equiv \left(\begin{array}{ccc|ccc|ccc}
\cos(2r_1)\mathbb{A}_2 & \cdots & 0 & \sin(2r_1)\mathbb{S}_2 & \cdots & 0 & 0 & \cdots & 0 \\
\vdots & \ddots & \vdots & \vdots & \ddots & \vdots & \vdots & \ddots & \vdots \\
0 & \cdots & \cos(2r_{N_A})\mathbb{A}_2 & 0 & \cdots & \sin(2r_{N_A})\mathbb{S}_2 & 0 & \cdots & 0 \\
\hline
-\sin(2r_1)\mathbb{S}_2 & \cdots & 0 & \cos(2r_1)\mathbb{A}_2 & \cdots & 0 & 0 & \cdots & 0 \\
\vdots & \ddots & \vdots & \vdots & \ddots & \vdots & \vdots & \ddots & \vdots \\
0 & \cdots & -\sin(2r_{N_A})\mathbb{S}_2 & 0 & \cdots & \cos(2r_{N_A})\mathbb{A}_2 & 0 & \cdots & 0 \\
0 & \cdots & 0 & 0 & \cdots & 0 & \mathbb{A}_2 & \cdots & 0 \\
\vdots & \ddots & \vdots & \vdots & \ddots & \vdots & \vdots & \ddots & \vdots \\
0 & \cdots & 0 & 0 & \cdots & 0 & 0 & \cdots & \mathbb{A}_2
\end{array}\right) \quad \textbf{(fermions)}
$$

$$(220)$$

*with matrices $\mathbb{A}_2$ and $\mathbb{S}_2$ written as*

$$
\mathbb{A}_2 \overset{q,p}{\equiv} \begin{pmatrix} 0 & 1 \\ -1 & 0 \end{pmatrix} \overset{a,a\dagger}{\equiv} \begin{pmatrix} -i & 0 \\ 0 & i \end{pmatrix}, \qquad \mathbb{S}_2 \overset{q,p}{\equiv} \begin{pmatrix} 0 & 1 \\ 1 & 0 \end{pmatrix} \overset{a,a\dagger}{\equiv} \begin{pmatrix} 0 & i \\ -i & 0 \end{pmatrix}. \tag{221}
$$

*In particular, we find that $J_A$ and $J_B$ have eigenvalues $\pm i\lambda_i$ with $\lambda_i \in [1,\infty)$ for bosons and $\lambda_i \in [0,1]$ for fermions.*

*Proof.* A detailed proof can be found in the appendices of [23] split over propositions 2 to 10. Equivalent results have been well-known in terms of the covariance matrices [52].
The idea is to first show that $J_A^2$ and $J_B^2$ are diagonalizable and have the same spectrum except for eigenvalues $-1$ (corresponding to eigenvalues $\pm i$ of $J_A$ and $J_B$). In a second step, one then needs to distinguish between bosonic and fermionic systems to show that $J_A$ and $J_B$ are diagonalizable with eigenvalues $\pm i\lambda_i$ of $J_A$ and $J_B$ satisfy $\lambda_i \in [1,\infty)$ for bosons and $\lambda_i \in [0,1]$ for fermions. At this stage, the block forms of $J_A$ and $J_B$ follow from the fact that any matrix with imaginary eigenvalues can be brought into block-diagonal form. In the third and last step, one then shows that $J_{AB}$ and $J_{BA}$ relate those eigenvectors of $J_A$ and $J_B$ whose eigenvalues are not $\pm i$ with a prescribed rescaling to ensure that $J$ as a whole only has eigenvalues $\pm i$. $\qquad\square$

We can now show that the reduction of a pure Gaussian state to such a subsystem gives rise to a mixed Gaussian state.

**Proposition 13.** *Given a pure Gaussian state $|J,z\rangle$ and a system decomposition $V = A \oplus B$ inducing the tensor product $\mathcal{H} = \mathcal{H}_A \otimes \mathcal{H}_B$, the reduced state*

$$
\rho_A(J,z) = \mathrm{Tr}_{\mathcal{H}_B} |J,z\rangle \langle J,z| \tag{222}
$$

*is Gaussian and explicitly given by $\rho_A(J,z) = \rho_{(J_A,z_A)}$, where $J_A$ was defined in proposition 12 and $z_A = \mathbb{P}_A z$ is the projection of $z$ onto A.*

*Proof.* This result follows from the fact that $\rho_{(J_A,z_A)}$ and $\rho_A(J,z)$ satisfy the same Wick's theorem, so that all their $n$-point functions agree, so they must be equal. $\qquad\square$

The restricted covariance matrix satisfies

$$\mathrm{Tr}\left(\rho_A(J,z)\,\hat{\xi}_A^r\hat{\xi}_A^s\right) = \frac{1}{2}\left(G_A^{rs} + \mathrm{i}\Omega_A^{rs}\right). \tag{223}$$

The real bilinear form $q_{rs}$ is symmetric for bosons and anti-symmetric for fermions. It can be compactly written in terms of the restricted linear complex structure as

$$q = \begin{cases} -\mathrm{i}\omega_A\,\mathrm{arccoth}\,(\mathrm{i}J_A) \\ +\mathrm{i}g_A\,\mathrm{arctanh}\,(\mathrm{i}J_A) \end{cases} = \begin{cases} +\omega_A\,\mathrm{arccot}\,(J_A) & \textbf{(bosons)} \\ -g_A\,\mathrm{arctanh}\,(J_A) & \textbf{(fermions)} \end{cases}, \tag{224}$$

where $\omega_A$ and $g_A$ are the restrictions of $\omega$ and $g$ to $A$. This follows from the respective structures discussed in section 3.2.

We can use this basis to find an explicit representation of the states with respect to the number operators $\hat{n}_i$ associated to this basis, namely

$$\rho_A = \begin{cases} \sum_{n_1,\cdots,n_{N_A}=0}^{\infty}\left(\prod_{i=1}^{N_A}\frac{(\tanh r_i)^{n_i}}{\cosh r_i}\right)^2 |n_1,\ldots,n_{N_A}\rangle\langle n_1,\ldots,n_{N_A}| & \textbf{(bosons)} \\ \sum_{n_1,\cdots,n_{N_A}=0}^{1}\left(\prod_{i=1}^{N_A}\frac{(\tan r_i)^{n_i}}{\sec r_i}\right)^2 |n_1,\ldots,n_{N_A}\rangle\langle n_1,\ldots,n_{N_A}| & \textbf{(fermions)} \end{cases}. \tag{225}$$

### 4.1.3 Entanglement entropy

Given a mixed Gaussian state $\rho_{(J,z)}$, we can compute the von Neumann-entropy

$$S(\rho_{(J,z)}) = -\mathrm{Tr}(\rho_{(J,z)}\log\rho_{(J,z)}) \tag{226}$$

using the explicit representation of $\rho_{(J,z)}$ from (177) to find

$$S(\rho) = \left|\mathrm{Tr}(\mathrm{i}J\,\mathrm{argh}\,\mathrm{i}J) + \tfrac{1}{4}\log\det\left(\frac{\mathbb{1}+J^2}{4}\right)\right|, \tag{227}$$

where we introduced the matrix function

$$\mathrm{argh}(x) = \tfrac{1}{4}\log\left(\frac{1+x}{1-x}\right)^2 = \begin{cases} \mathrm{arctanh}(x) & x \in [0,1] \\ \mathrm{arccoth}(x) & x \in [1,\infty) \end{cases} \tag{228}$$

applied to the restricted complex structure $\mathrm{i}J$. We can read off the entanglement spectrum as eigenvalues of $\rho_A$ which allows the computation of entanglement entropy $S_A$ and the Rényi entropy $R_A^{(\alpha)}$ of order $\alpha$ as

$$S_A = \sum_{i=1}^{N_A}\left(\cosh^2 r_i\log\cosh^2 r_i - \sinh^2 r_i\ln\sinh^2 r_i\right),$$

$$R_A^{(\alpha)} = \tfrac{1}{\alpha-1}\sum_{i=1}^{N_A}\log\left(\cosh^{2\alpha} r_i - \sinh^{2\alpha} r_i\right), \qquad \textbf{(bosons)}$$

$$S_A = -\sum_{i=1}^{N_A}\left(\cos^2 r_i\log\cos^2 r_i + \sin^2 r_i\log\sin^2 r_i\right), \tag{229}$$

$$R_A^{(\alpha)} = \tfrac{1}{1-\alpha}\sum_{i=1}^{N_A}\log\left(\cos^{2\alpha} r_i + \sin^{2\alpha} r_i\right). \qquad \textbf{(fermions)}$$

We can use the restricted complex structure $J_A$ to find a particularly compact trace formula for the entanglement entropy valid for both bosons and fermions, namely [53]

$$S_A = \left| \mathrm{Tr}\left( \frac{\mathbb{1}_A + \mathrm{i}J_A}{2} \log \left| \frac{\mathbb{1}_A + \mathrm{i}J_A}{2} \right| \right) \right| . \tag{230}$$

Similarly, we can express the Rényi entropies of order 2 as simple determinants

$$R_A^{(2)} = \begin{cases} \frac{1}{2}\log|\det \mathrm{i}J_A| & \textbf{(bosons)} \\ -\frac{1}{2}\log\det\left(\frac{\mathbb{1}_A - J_A^2}{2}\right) & \textbf{(fermions)} \end{cases} . \tag{231}$$

The entanglement entropy is bounded from above for fermions, due to the fact that the fermionic Hilbert space is finite-dimensional. The maximally entangled state is characterized by $J_A = 0$, i.e., all eigenvalues $\lambda_i$ vanish, and we have $S_A = N_A \log 2$ (assuming $N_A \leq N_B$). For bosons, the entanglement entropy is not bounded from above and the maximally mixed state can only be reached asymptotically, as it does not exist as a proper mixed state. When we consider time-evolution, these properties are also reflected by the fact that fermionic Gaussian states form a *compact* manifold, while bosonic Gaussian states form a *non-compact* manifold. This leads to interesting questions in the context of producing entanglement through time evolution [17, 18, 53, 53, 54].

The entanglement entropy of a non-Gaussian state will in general also depend on higher $n$-point functions, so we cannot use (230) anymore. Interestingly, if we perturb a Gaussian state in a non-Gaussian way, the entanglement entropy will at linear order only feel the Gaussian part of the perturbation [55], so that we can use the linearization of (230) to deduce the linear change

$$\delta S_A = \mathrm{Tr}\left( \frac{\delta S_A(J)}{\delta J} \delta J \right) , \tag{232}$$

via the first law of entanglement entropy [56–58].

For bosons, let us note that the entanglement entropy does not depend on the displacement $z$ of a state $|J, z\rangle$. For fermions, we can expand formula (230) in $J_A$ to find the power series

$$S_A = N_A \log 2 - \sum_{n=1}^{\infty} \frac{\mathrm{Tr}(\mathrm{i}J_A)^{2n}}{2n(2n-1)} , \qquad \textbf{(fermions)} \tag{233}$$

where we used that $\mathrm{Tr}(\mathrm{i}J_A)^{2n+1} = 0$. This series converges monotonously and absolutely. Moreover, any truncation of this series provides both an upper and a lower bound given by

$$S_A^{m+} = N_A \log 2 - \sum_{n=1}^{m} \frac{\mathrm{Tr}(\mathrm{i}J_A)^{2n}}{2n(2n-1)} ,$$

$$S_A^{m-} = N_A \left( \log 2 - \sum_{m+1}^{\infty} \frac{1}{n(2n-1)} \right) - \sum_{n=1}^{m} \frac{\mathrm{Tr}(\mathrm{i}J_A)^{2n}}{2n(2n-1)} ,$$

which one deduces from the inequality $0 \leq \mathrm{Tr}[\mathrm{i}J]_A^{2n} \leq 2N_A$. This inequality is a direct consequence from the fact that the restricted complex structure $[J]_A$ of a fermionic state has purely imaginary eigenvalues $\pm \mathrm{i}\lambda$ with $0 \leq \lambda \leq 1$, which we derived in [19].

The inequalities $S_A^{m-} \leq S_A \leq S_A^{m+}$ have been used in the context of typical entanglement of energy eigenstates [19, 21, 22, 59–61], but are likely also useful in other contexts.

### 4.1.4 Relative entropy

Given two mixed states $\rho$ and $\sigma$, the relative entropy $S(\rho\|\sigma)$ is defined as

$$S(\rho\|\sigma) = \text{Tr}\,\rho(\log\rho - \log\sigma)\,. \tag{234}$$

If both states are Gaussian states with respective complex structures $J_\rho$ and $J_\sigma$, we can use (177) to find

$$S(\rho\|\sigma) = \left| \text{Tr}\, \mathrm{i}J_\rho(\text{argh}\,\mathrm{i}J_\sigma - \text{argh}\,\mathrm{i}J_\rho) + \tfrac{1}{4}\log\det\left(\tfrac{\mathbb{1}+J_\sigma^2}{\mathbb{1}+J_\rho^2}\right) \right|, \tag{235}$$

where the ordering within the determinant does not matter, as the equation can be understood in terms of eigenvalues.

### 4.1.5 Circuit complexity

Circuit complexity is another quantum information-theoretic quantity which has recently emerged as an interesting field of research in the context of and holography. Holography provides an approach to quantum gravity where quantum field theory states on the boundary of a spacetime can be related geometry inside the bulk of the spacetime. This is also known as bulk-boundary correspondence or AdS-CFT, as the spacetime is typically assumed to be asymptotically anti-De Sitter (AdS) space and the quantum field theory on the boundary is taken to be a conformal field theory (CFT).

In this setting, it was noticed in [62–67] that certain geometric quantities computed in the bulk (such as codimension-one boundary-anchored maximal volumes and codimension-zero boundary-anchored causal diamonds) behave similar as the difficulty of preparing quantum states by applying a sequence of quantum operations to a reference state [68]. So far, it has been an open problem to make this observation concrete by identifying dual quantities on the boundary field theory that match those computed in the bulk. However, there has been some partial progress [69] by defining circuit complexity for free quantum fields based on the number of Gaussian transformations $e^{\epsilon\widehat{K}_i}$ with $K_i$ applied to a spatially unentangled Gaussian reference state $|J_\text{R}\rangle$ to reach the entangled field theory vacuum

$$|J_\text{T}\rangle = \left(\prod_{i=1}^{n} e^{\epsilon\widehat{K}_i}\right)|J_\text{R}\rangle\,, \tag{236}$$

as target state. The idea behind this definition is that the circuit complexity (or circuit depth) is given by the number of elementary gates $e^{\epsilon\widehat{K}_i}$ applied to the reference state. For this, it is important to require the normalization condition

$$\|K_i\|^2 = \frac{1}{2}\text{Tr}(K_i G_\text{R} K_i^\mathsf{T} g_\text{R}) = 1\,, \tag{237}$$

for the generators $K_i$, where $G_\text{R}$ and $g_\text{R}$ are the metric associated to the reference state $|J_\text{R}\rangle$. In the limit $\epsilon \to 0$ and $n \to \infty$, this becomes a path ordered exponential $\mathcal{S}(M) = \mathcal{P}\exp\int_0^1 \widehat{K}(t)dt$ and we can approximate $n\epsilon \approx \int_0^1 \|K(t)\|dt$ by the length of the path. The circuit complexity $\mathcal{C}(|J_\text{T}\rangle, |J_\text{R}\rangle)$ is then defined as the minimum over all paths, *i.e.*, the geodesic distance between the identity group element $\mathbb{1}$ and the closest point in the equivalence class $[M]$ with $J_\text{T} = MJ_\text{R}M^{-1}$.

As the above setup only describes the preparation of Gaussian states, it can be understood as Gaussian circuit complexity whose generalization to genuinely interacting field theories has

not been accomplished, so far. Above minimization can be carried out analytically to find the Gaussian circuit complexity to be given by

$$\mathcal{C}(|J_\mathrm{T}\rangle,|J_\mathrm{R}\rangle) = \sqrt{\frac{|\operatorname{Tr}\log^2(\Delta)|}{8}} \tag{238}$$

in terms their relative complex structure $\Delta = J_\mathrm{T} J_\mathrm{R}^{-1}$, as proven in [70] for fermions and in [71] for bosons. Interestingly, formula (238) also makes sense when defining circuit complexity for mixed Gaussian states using the Fisher information geometry, as derived for bosons in [72]. The geometry of Gaussian states was also used to define the so-called complexity of purification (CoP), where formula (238) is minimized over all Gaussian purifications of a given mixed Gaussian state [25,73,74].

## 4.2 Dynamics of stable quantum systems

We present compact equations for the full dynamics of bosonic and fermionic Gaussian states under the evolution of time-independent quadratic Hamiltonians.

### 4.2.1 Time-independent quadratic Hamiltonians

We consider the most general time-independent quadratic Hamiltonian,

$$\hat{H} = \begin{cases} \frac{1}{2}h_{ab}\hat{\xi}^a\hat{\xi}^b + f_a\hat{\xi}^a & \textbf{(bosons)} \\ \frac{i}{2}h_{ab}\hat{\xi}^a\hat{\xi}^b & \textbf{(fermions)} \end{cases}. \tag{239}$$

Due to commutation or anti-commutation relations, for bosons and fermions respectively, only the symmetric or antisymmetric part of $h_{ab}$ will contribute to the physics, while the other part only leads to a shift of the zero point energy. We can define the Lie algebra generator associated to the Hamiltonian as

$$K^a{}_b = \begin{cases} \frac{1}{2}\Omega^{ac}(h_{cb}+h_{bc}) \in \mathfrak{sp}(2N,\mathbb{R}) & \textbf{(bosons)} \\ \frac{1}{2}G^{ac}(h_{cb}-h_{bc}) \in \mathfrak{so}(2N) & \textbf{(fermions)} \end{cases}. \tag{240}$$

In the bosonic case, the Hamiltonian is bounded from below and the system is stable if $h_{ab}$ is positive definite. In the fermionic case, as the Hilbert space is finite dimensional and the system is always stable.

In the stable case, the generator $K$ can be put in standard form. One chooses a basis where $\Omega$ for bosons and $G$ for fermions is in its standard form (15) and then use the group $\mathcal{G}$, i.e., $\mathrm{Sp}(2N,\mathbb{R})$ for bosons and $\mathrm{O}(2N,\mathbb{R})$ for fermions, to change to a new basis $\hat{\xi}^a \equiv (\hat{q}_1,\hat{p}_1,\cdots,\hat{q}_N,\hat{p}_N)$ without modifying $\Omega$ or $G$ to bring $K$ into the standard form

$$K \overset{q,p}{\equiv} \bigoplus_{i=1}^{N} \begin{pmatrix} 0 & \epsilon_i \\ -\epsilon_i & 0 \end{pmatrix}, \tag{241}$$

where $\epsilon_i > 0$. This is obviously possible for fermions, because $K \in \mathfrak{so}(2N,\mathbb{R})$ is antisymmetric with respect to $G$, but it is also well-known that it can be done for bosons if $h_{ab}$ is positive definite as consequence of Williamson's theorem [44] (see App. B of [16] for a constructive proof). The eigenvalues of $K$ are thus $\pm i\epsilon_i$.

### 4.2.2 Dynamics of a Gaussian state

Under time evolution, quadratic Hamiltonians send Gaussian states into Gaussian states (See Sec. 3.1). Given an initial Gaussian state $|J_0, z_0\rangle$, the unitary time evolution $|J(t), z(t)\rangle = U(t) |J_0, z_0\rangle$ with $U(t) = e^{-i\hat{H}t}$ is completely determined by the evolution of the two-point correlation function,

$$\langle J(t), z(t)| \hat{\xi}^a \hat{\xi}^b |J(t), z(t)\rangle = \frac{1}{2}(G^{ab}(t) + i\Omega^{ab}(t)) + z^a(t)z^b(t). \tag{242}$$

Taking its time derivative, using the Schrödinger equation $i\partial_t |J(t), z(t)\rangle = \hat{H} |J(t), z(t)\rangle$ and the relation between Kähler structures, one finds

$$\begin{aligned}
\dot{J}(t) &= [K, J(t)] = KJ(t) - J(t)K, \\
\dot{z}^a(t) &= K^a{}_b z^b(t) + \Omega^{ab}f_b,
\end{aligned} \tag{243}$$

which has solution

$$\begin{aligned}
J(t) &= M(t)J_0 M^{-1}(t), \\
z(t) &= M(t)z_0 + M(t)\int_0^t M^{-1}(t')\Omega f \, dt',
\end{aligned} \tag{244}$$

with $M(t) = \exp(Kt)$ the symplectic or orthogonal transformation associated to the bosonic or fermionic dynamics.

Time evolution is an example of the natural group action of an element $M \in \mathcal{G}$ onto any Gaussian state $|J\rangle$ leading to $|MJM^{-1}\rangle$. This forms a natural representation of the group $\mathcal{G}$, but every Gaussian state $|J\rangle$ selects an invariant subgroup

$$\mathrm{Sta}_{|J\rangle} = \left\{ M \in \mathcal{G} \middle| MJM^{-1} = J \right\} \tag{245}$$

isomorphic to $U(N)$. This group arises naturally as the intersection

$$U(N) = \mathrm{Sp}_\Omega(2N, \mathbb{R}) \cap O_G(2N) \cap \mathrm{GL}_J(N, \mathbb{C}), \tag{246}$$

for any triple $(\Omega, G, J)$ of Kähler structures. Technically, this is only a proper representation on the space of Gaussian quantum states $\rho(J) = |J\rangle\langle J|$, while for Gaussian state vectors $|J\rangle$ we need to take complex phases into account. The unitary subgroup generated by hermitian operators $\hat{H}$ is in fact not given by $\mathcal{G}$, but by its double cover $\overline{\mathcal{G}}$ which is given by metaplectic group $\mathrm{Mp}(2N, \mathbb{R})$ for bosonic systems and the spin group $\mathrm{Spin}(N)$ for fermionic systems.

The expressions (244), together with the results of Sec. 4.1, allow one to compute the time evolution of information theoretic quantities such as the entanglement entropy.

### 4.2.3 Expectation value of the energy

The expectation value of the Hamiltonian on a Gaussian state $|J, z\rangle$ can be easily computed using Wick's theorem (See Sec. 3.1.2),

$$\langle J, z|\hat{H}|J, z\rangle = c_0 - \tfrac{1}{4}\mathrm{Tr}(KJ) + \tfrac{1}{2}h_{ab}z^a z^b + f_a z^a. \tag{247}$$

The term $c_0$ is independent of the state and is due to the definition of the Hamiltonian (239),

$$c_0 = \begin{cases} -\frac{1}{4}h_{ab}\Omega^{ab} & \textbf{(bosons)} \\ \frac{i}{4}h_{ab}G^{ab} & \textbf{(fermions)} \end{cases}. \tag{248}$$

The term $E_{\mathrm{cl}} = \frac{1}{2}h_{ab}z^a z^b + f_a z^a$ represents the energy of a classical system with phase-space configuration $z^a$. Lastly, the term $E_J = \frac{1}{4}\mathrm{Tr}(KJ)$ has purely quantum origin and depends on the complex structure defining the Gaussian state.

#### 4.2.4 Ground state and vacuum correlations

Provided that $h_{ab}$ is a positive definite bilinear form on $V$ for bosons and non-degenerate for fermions, the system has a unique ground state $|J_0, z_0\rangle$. The complex structure $J_0$ and the shift $z_0$ of the ground state can be determined by minimizing the expectation value of the energy (247) with respect to $J$ and $z$. One finds

$$(J_0)^a{}_b = |K^{-1}|^a{}_c K^c{}_b \quad \text{and} \quad z_0^a = -(h^{-1})^{ab} f_b, \tag{249}$$

where $|K|^a{}_b$ is the absolute value of $K$, which is best defined in an eigenbasis[19] Furthermore, we can plug this into expression (247) to find the vacuum energy

$$E_0 = c_0 + \tfrac{1}{4}\mathrm{Tr}(|K|) - \tfrac{1}{2} f_a (h^{-1})^{ab} f_b. \tag{250}$$

As the eigenvalues of $K$ are $\pm i \epsilon_i$ and the ones of $|K|$ are $\epsilon_i$ appearing in pairs, such that $\tfrac{1}{4}\mathrm{Tr}(|K|) = \tfrac{1}{2}\sum_{i=1}^{N} \epsilon_i$.

It is immediate to check that the ground state $|J_0, z_0\rangle$ is an eigenstate of the Hamiltonian as it is stationary: using (243) and (249) we see that $\dot{J} = 0$ as $[K, J_0] = 0$ and $\dot{z} = 0$. Note that for fermionic systems, the condition of stationarity is not sufficient to determine the ground state as all energy eigenstates are Gaussian.

Having determined the vacuum associated to the stable Hamiltonian (239), we can now express vacuum correlations directly in terms of the Hamiltonian as

$$\langle J_0, z_0| \hat{\xi}^a \hat{\xi}^b |J_0, z_0\rangle = \begin{cases} \tfrac{1}{2}(\mathbb{1} + i|K|^{-1}K)^a{}_c \, i\Omega^{cb} + z_0^a z_0^b & \textbf{(bosons)} \\ \tfrac{1}{2}(\mathbb{1} + i|K|^{-1}K)^a{}_c \, G^{cb} & \textbf{(fermions)} \end{cases}, \tag{251}$$

with $K$ given in (240).

### 4.3 Dynamics of driven quantum systems

We extend our formalism to driven quantum systems to describe the dynamics of bosonic and fermionic Gaussian states for time-dependent quadratic Hamiltonians. This also allows us to describe instantaneous and adiabatic vacua, which play an important role in driven quantum systems and quantum field theory in curved spacetime.

#### 4.3.1 Quadratic time-dependent Hamiltonians

We consider the most general time-dependent quadratic Hamiltonian,

$$\hat{H}(t) = \begin{cases} \tfrac{1}{2} h_{ab}(t) \hat{\xi}^a \hat{\xi}^b + f_a(t) \hat{\xi}^a & \textbf{(bosons)} \\ \tfrac{i}{2} h_{ab}(t) \hat{\xi}^a \hat{\xi}^b & \textbf{(fermions)} \end{cases}, \tag{252}$$

where both $h_{ab}(t)$ and $f_a(t)$ depend on time. We assume $h_{ab}(t)$ to be symmetric for bosons and antisymmetric for fermions, therefore dropping an unimportant time-dependent function of time that can be added to the Hamiltonian. We can then define the time-dependent Lie algebra generator associated to the Hamiltonian as

$$K^a{}_b(t) = \begin{cases} \Omega^{ac} h_{cb}(t) \in \mathfrak{sp}(2N, \mathbb{R}) & \textbf{(bosons)} \\ G^{ac} h_{cb}(t) \in \mathfrak{so}(2N) & \textbf{(fermions)} \end{cases}. \tag{253}$$

---

[19]As the eigenvalues of $K$ are $\pm i \epsilon_i$, we have also $|K|^2 = -K^2 > 0$.

We assume that the Hamiltonian is instantaneously stable, i.e., in the bosonic case $h_{ab}(t)$ is positive definite for all $t$. As a result the eigenvalues of $K^a{}_b(t)$ come in pairs $\pm i\epsilon_i(t)$. Note that in general both the eigenvalues and the eigenvectors of $K^a{}_b(t)$ have a non-trivial time dependence and a transformation that puts $K^a{}_b(t)$ in the standard form (241) at a time, fails to do it at a different time.

### 4.3.2 Dynamics of a Gaussian state

The unitary time evolution of an initial Gaussian state,

$$|J(t), z(t)\rangle = U(t, t_0)|J_0, z_0\rangle, \tag{254}$$

with

$$U(t, t_0) = \mathcal{T} e^{-i\int_{t_0}^t \hat{H}(t')dt'}, \tag{255}$$

is completely determined by the evolution of the two-point correlation function defined as in (242). Taking its time derivative, using the Schrödinger's equation $i\partial_t|J(t), z(t)\rangle = \hat{H}(t)|J(t), z(t)\rangle$ and the relation between Kähler structures, one finds

$$\dot{J}(t) = [K(t), J(t)],$$
$$\dot{z}^a(t) = K^a{}_b(t)z^b(t) + \Omega^{ab}f_b(t), \tag{256}$$

which has solution

$$J(t) = M(t, t_0)J(t_0)M^{-1}(t, t_0), \tag{257}$$
$$z(t) = M(t, t_0)z(t_0) + M(t, t_0)\int_{t_0}^t M^{-1}(t', t_0)k(t')\,dt',$$

where

$$M(t, t_0) = \mathcal{T}\exp(\int_{t_0}^t K(t')dt') \tag{258}$$

is the symplectic or orthogonal transformation associated to the bosonic or fermionic dynamics, expressed as a time-ordered exponential. Furthermore, for bosons $k^a(t) = \Omega^{ab}f_b(t)$.

### 4.3.3 Instantaneous and adiabatic vacua

In the general time-dependent case there is no absolute notion of vacuum. However, as we have assumed that the system is instantaneously stable, we can define the instantaneous vacuum at the time $t$ as the Gaussian state $|J_t^{(0)}, z_t^{(0)}\rangle$ with complex structure and shift defined as in (249),

$$J_t^{(0)} = |K(t)|^{-1}K(t) \text{ and } z_t^{(0)} = -h^{-1}(t)f(t). \tag{259}$$

Note that, under time evolution, the instantaneous vacuum does not evolve into the instantaneous vacuum, i.e., $U(t_2, t_1)|J_{t_1}^{(0)}, z_{t_1}^{(0)}\rangle \neq |J_{t_2}^{(0)}, z_{t_2}^{(0)}\rangle$.

The instantaneous vacuum is the starting point for the definition of the notion of adiabatic vacua of order $m$. Adiabatic vacua arise in the context of driven slowly changing systems, where one can identify a small parameter $\lambda$ characterizing the time dependence. They play an important role in quantum field theory in curved spacetime and cosmology [4, 32, 75, 76], where they are natural candidates for initial states in dynamical background geometries. Interestingly, the concept is intimately linked to the so-called Lewis–Riesenfeld invariants [77],

Table 5: *Applications.* This table summarizes and compares our methods to computed properties of bosonic and fermionic Gaussian states using Kähler structures covered in section 4.

| structure | bosons | fermions |
|---|---|---|
| von Neumann entropy | $S(\rho_{(J,z)}) = \left\| \mathrm{Tr}(iJ\,\mathrm{argh}\,iJ) + \frac{1}{4}\log\det\left(\frac{\mathbb{1}+J^2}{4}\right)\right\|$ with $\mathrm{argh}(x) = \frac{1}{4}\log\left(\frac{1+x}{1-x}\right)^2$ | |
| Restricted complex structure $J_A$ | $J_A \equiv \bigoplus_{i=1}^{N_A}\begin{pmatrix} 0 & \cosh 2r_i \\ -\cosh 2r_i & 0\end{pmatrix}$ | $J_A \equiv \bigoplus_{i=1}^{N_A}\begin{pmatrix} 0 & \cos 2r_i \\ -\cos 2r_i & 0\end{pmatrix}$ |
| entanglement entropy | $S_A = \sum_{i=1}^{N_A}\left(\cosh^2 r_i \log\cosh^2 r_i - \sinh^2 r_i \log\sinh^2 r_i\right)$ | $S_A = -\sum_{i=1}^{N_A}\left(\cos^2 r_i \log\cos^2 r_i + \sin^2 r_i \log\sin^2 r_i\right)$ |
| entanglement entropy trace formula | $S_A = \mathrm{Tr}\left(\frac{\mathbb{1}_A+iJ_A}{2}\log\left\|\frac{\mathbb{1}_A+iJ_A}{2}\right\|\right)$ | $S_A = -\mathrm{Tr}\left(\frac{\mathbb{1}_A+iJ_A}{2}\log\frac{\mathbb{1}_A+iJ_A}{2}\right)$ |
| Rényi entropy of order $n$ | $R_A^{(n)} = \frac{1}{n-1}\sum_{i=1}^{N_A}\log\left(\cosh^{2n} r_i - \sinh^{2n} r_i\right)$ | $R_A^{(n)} = -\frac{1}{n-1}\sum_{i=1}^{N_A}\log\left(\cos^{2n} r_i + \sin^{2n} r_i\right)$ |
| Rényi entropy of order 2 | $R_A^{(2)} = \frac{1}{2}\log\|\det iJ_A\|$ | $R_A^{(2)} = -\frac{1}{2}\log\det\left(\frac{\mathbb{1}_A - J_A^2}{2}\right)$ |
| relative entropy | $S(\rho\|\sigma) = \left\|\mathrm{Tr}\,iJ_\rho(\mathrm{argh}\,iJ_\sigma - \mathrm{argh}\,iJ_\rho) + \frac{1}{4}\log\det\left(\frac{\mathbb{1}+J_\sigma^2}{\mathbb{1}+J_\rho^2}\right)\right\|$ | |
| circuit complexity | $\mathcal{C}(\|J_T\rangle,\|J_R\rangle) = \sqrt{\frac{1}{8}\left\|\mathrm{Tr}\log^2(\Delta)\right\|}$ with $\Delta = J_T J_R^{-1}$ | |
| Hamiltonian | $\hat{H}(t) = \frac{1}{2}h(t)_{ab}\hat{\xi}^a\hat{\xi}^b + f(t)_a\hat{\xi}^a$ | $\hat{H}(t) = \frac{1}{2}h(t)_{ab}\hat{\xi}^a\hat{\xi}^b$ |
| generator | $K^a{}_b(t) = \Omega^{ac}h_{cb}(t)$ | $K^a{}_b(t) = G^{ac}h_{cb}(t)$ |
| equations of motion | $\dot{J}(t) = [K(t),J(t)] = K(t)J(t) - J(t)K(t)$ and $\dot{z}(t) = K(t)z(t) + \Omega^{ab}f_b(t)$ | |
| classical solutions | $J(t) = M(t)J_0 M^{-1}(t)$ and $z(t) = M(t)z_0 + M(t)\int_0^t M^{-1}(t')k(t')\,dt'$ with $k^a = \Omega^{ab}f_b(t)$ and $M(t) = \mathcal{T}\exp\left(\int_0^t K(t')dt'\right)$ | |
| ground state $\|J_0,z_0\rangle$ | $J_0 = \|K\|^{-1}K$ and $z_0 = -(h^{-1})^{ab}f_b$ | |
| ground state energy | $E_0 = c_0 + \frac{1}{4}\mathrm{Tr}(\|K\|) - \frac{1}{2}f_a(h^{-1})^{ab}f_b$ with | |
|  | $c_0 = -\frac{1}{2}h_{ab}\Omega^{ab}$ | $c_0 = \frac{i}{4}h_{ab}G^{ab}$ |
| adiabatic vacua $\|J_t^{(m)}, z_t^{(m)}\rangle$ | $\lambda \dot{J}_t^{(m)} = [K(t),J_t^{(m)}]$, $J_t^{(m)2} = -\mathbb{1}$ and $\lambda \dot{z}_t^{(m)} = K(t)z_t^{(m)} + \Omega f(t)$ | |
| vacuum subtraction | $\delta E(t)\|_{(m)} = -\frac{1}{4}\mathrm{Tr}\left(K(t)\left(J(t) - J_t^{(m)}\right)\right)$ | |

where the adiabatic state can be related to certain time dependent operators. In the context of Gaussian adiabatic states $|J_t^{(\infty)}, z_t^{(\infty)}\rangle$, this invariant operator turns out to be the respective number operator $\hat{N}_{J_t^{(\infty)}}$ defined in (93).

We introduce a notion of adiabatic vacuum for bosons and fermions defined directly in terms of Kähler structures. The notion is adapted to the time-dependent Hamiltonian (252) and to a choice of reference time $t$. We start from the definition of instantaneous vacuum (259) and introduce the ansatz

$$J_t^{(m)} = J_t^{(0)} + \sum_{n=1}^m A_n(t)\lambda^n, \tag{260}$$

$$z_t^{(m)} = z_t^{(0)} + \sum_{n=1}^m \zeta_n(t)\lambda^n, \tag{261}$$

for the adiabatic vacuum $|J_t^{(m)}, z_t^{(m)}\rangle$ at order $m$. By requiring that the following two conditions (the first due to the dynamics (256) and the second imposing that $J_t^{(m)}$ is a complex structure)

$$\lambda\,\partial_t J_t^{(m)} = [K(t), J_t^{(m)}] \quad \text{and} \quad (J_t^{(m)})^2 = -\mathbb{1}, \tag{262}$$

are satisfied at the time $t$ and at each order in $\lambda$, we can determined $J_m$ and $z_m$ by solving

algebraically the equations

$$[K, A_n] = \dot{A}_{n-1}$$
$$\{J_t^{(0)}, A_n\} = -(A_1 A_{n-1} + \ldots + A_{n-1} A_1) \tag{263}$$

evaluated at time $t$ for $A_n(t)$ in terms of $\dot{A}_{n-1}(t)$ and

$$\lambda \, \partial_t z_t^{(m)} = K(t) z_t^{(m)} + \Omega f(t). \tag{264}$$

The adiabatic vacuum of order $m$ at the time $t_0$ is then obtained as the Gaussian state $|J_{t_0}^{(m)}, z_{t_0}^{(m)}\rangle$ associated to $J_{t_0}^{(m)}$ and $z_{t_0}^{(m)}$ by setting $\lambda = 1$.

In general the series is only asymptotic in $\lambda$ and does not converge. When the series converges in the limit $m \to \infty$, we can define the *exact* adiabatic vacuum $|J_{t_0}^{(\infty)} z_{t_0}^{(\infty)}\rangle = \lim_{m \to \infty} |J_{t_0}^{(m)}, z_{t_0}^{(m)}\rangle$ at time $t_0$. In this case, the time evolution under $\hat{H}(t)$ evolves the adiabatic vacuum $|J_{t_0}^{(\infty)} z_{t_0}^{(\infty)}\rangle$ into $|J_t^{(\infty)} z_t^{(\infty)}\rangle$ at later times. Of course, this is only possible for special cases where $\hat{H}(t)$ is an analytical function of $t$.

### 4.3.4 Time-dependent vacuum subtraction

In a stable time-independent system, the vacuum energy can be simply subtracted once and for all from the energy of the system. For instance, assuming for simplicity $f_a = 0$ in (239), we have

$$\delta E = \langle J | \hat{H} | J \rangle - \langle J_0 | \hat{H} | J_0 \rangle = -\tfrac{1}{4} \text{Tr}\big(K(J - J_0)\big), \tag{265}$$

where the vacuum complex structure is $J_0 = |K|^{-1}K$ and we have used (247), (250). This vacuum subtraction corresponds to the procedure of putting the Hamiltonian in standard form and then *normal ordering* the associated creation and annihilation operators.

On the other hand, in the time dependent case (252), there is no standard notion of normal ordering but there is still a well defined notion of vacuum subtraction associated to the adiabatic vacuum $|J_t^{(m)}\rangle$ of order $m$ at the time $t$,

$$\delta E(t)|_{(m)} = \langle J(t)|\hat{H}(t)|J(t)\rangle - \langle J_t^{(m)}|\hat{H}(t)|J_t^{(m)}\rangle = -\tfrac{1}{4}\text{Tr}\big(K(t)\big(J(t) - J_t^{(m)}\big)\big). \tag{266}$$

Note that, while the complex structure of the state $|J(t)\rangle$ evolves as $J(t) = M(t, t_0)J(t_0) M^{-1}(t, t_0)$, the complex structure of the adiabatic vacuum is computed at the time $t$ directly from $K(t)$ and its time derivatives via (263). In particular, $J_t^{(m)} \neq M(t, t_0)J_{t_0}^{(m)}M^{-1}(t, t_0)$. This adiabatic subtraction is well defined for the expectation value of all operators and plays an important role in the renormalization of the energy-momentum tensor in cosmological spacetimes [4, 32, 75, 76].

The formula (235) provides us also with a tool for computing the relative entanglement entropy of a Gaussian state $|J(t)\rangle$ with respect to the adiabatic vacuum $|J_t^{(m)}\rangle$ at the time $t$,

$$S_A(J(t)\|J_t^{(m)}) = \left| \text{Tr} \, iJ_A(t)(\text{argh} \, iJ_{tA}^{(m)} - \text{argh} \, iJ_A(t)) + \tfrac{1}{4}\log\det\left(\frac{\mathbb{1} + J_{tA}^{(m)2}}{\mathbb{1} + J_A^2(t)}\right) \right|. \tag{267}$$

## 5 Summary and discussion

In applications to quantum information, Gaussian states are often described in a covariance matrix formalism [1–3]. In sections 2 and 3 we have presented a comprehensive introduction

to the description of Gaussian states in terms of Kähler structures developed in the mathematical literature on quantization [7–9] and on quantum fields in curved spacetimes [10]. Here we have adopted a language and selected aspects that are tailored to applications in quantum information and non-equilibrium physics. In parallel to [8], we characterize pure and mixed, bosonic and fermionic Gaussian states by relating them to a triangle of Kähler structures $(G, \Omega, J)$ on the classical phase space and its dual. The key insight is that bosonic and fermionic Gaussian states can be parametrized by a linear complex structure $J^a{}_b$ in a unified manner. Before discussing applications to quantum information, let us highlight what we believe to be the main advantages of describing Gaussian states in this mathematical formalism:

**Gaussian states from Kähler structures.** Our formalism is arguably best encapsulated by the equation

$$(\mathbb{1} - \mathrm{i}J)^a{}_b(\hat{\xi} - z)^b \, |J, z\rangle = 0 \,, \tag{268}$$

from which $J^2 = -\mathbb{1}$ and the compatibility conditions of $(G, \Omega, J)$ can be derived for pure Gaussian states. We showed how for both, bosonic and fermionic Gaussian states, compatible Kähler structures turn the classical phase space $V$ into a complex vector space with inner product $\langle v, u \rangle$, known as the single-particle Hilbert space. In contrast, we found that mixed Gaussian states $\rho = e^{-\hat{Q}}$ are characterized by $G$ and $\Omega$, whose incompatibility is quantified by the failure of $J^2$ to be equal to $-\mathbb{1}$.

**Phase space covariance.** We put particular emphasis on ensuring that all our equations are independent of the chosen basis of $V$ and $V^*$, what is often referred to as *covariant equations*. This is in contrast to the typical treatment, where one often chooses either the Hermitian basis (we indicate by $\overset{q,p}{=}$) or the ladder operator basis (we indicate by $\overset{a,a^\dagger}{=}$). For example, we have $\Omega \overset{q,p}{=} -\omega$ for bosons, *i.e.*, the matrix representation of inverse symplectic form $\omega$ only picks up a sign in this basis, but this relation breaks down when we move to a different basis or consider fermionic states. While we provided a comprehensive list of examples, where we give the respective equations in both bases, we were careful to present all equations as covariant tensor equations using Einstein's summation convention and Penrose's abstract index notation (see appendix A.1). In this context, we also introduced the notion of phase space covariant ladder operators $\hat{\xi}^a_\pm$, which allowed us to give a rather compact derivation of a basis-independent Wick's theorem.

**Relative complex structure $\Delta$.** When comparing two different Gaussian states $|J, z\rangle$ and $|\tilde{J}, \tilde{z}\rangle$, we found that it is natural to define the object $\Delta = -\tilde{J}J$, which we call the *relative complex structure*. It provides a basis-independent way to characterize the relationship between the two states (apart from the displacement $z - \tilde{z}$) and we derived various properties of its spectrum, its utility when constructing the Cartan decomposition and how it appears naturally when studying unitary inequivalence of Fock spaces in field theory. *It was brought to our attention that [8] defines the same object under the name of k for bosons and fermions in the context of the Cartan decomposition, also known as j-polar decomposition.*

While these methods are well-known in mathematical physics, they have not been broadly applied in quantum information and out-of-equilibrium quantum systems. We believe that this manuscript can help to establish a link between these fields by providing comprehensive review of the methods and demonstrating their versatility in practical applications. In sections 4.1, 4.2 and 4.3, we have shown how Kähler structures provide a powerful tool for studying (A) entanglement and complexity for the vacuum and the adiabatic vacuum of (B) stable and of (C) driven quantum systems in bosonic and fermionic Gaussian states. In particular, we have shown concretely how quantities such as the entanglement entropy of a Gaussian state, and its time dependence in a driven quantum system, can be expressed in terms of

Kähler structures. The table 5 provides a comprehensive overview of our results that compare bosonic and fermionic expressions side-by-side which we hope to be useful for many readers. Remarkably, various formulas (*e.g.*, for the von Neumann entropy and the circuit complexity) take the same form for bosons and fermions when expressed in terms of $J$.

We believe that the presented formalism provides a starting point for future studies of Gaussian states from a mathematical physics perspective with applications in various research field. In fact, some of our methods have already been used to study entanglement production [17, 18, 53, 54], entanglement of energy eigenstates [19, 21, 22], variational methods [24, 34] and circuit complexity [70, 71, 73]. We also expect that our results are particularly useful for the study of generalized Gaussian states, as defined in [24, 39, 40], where we allow for certain non-Gaussian unitaries to entangle bosonic and fermionic degrees of freedom in the initially unentangled Gaussian state.

As outlined in section 4.3.3, Kähler structures also provide a powerful tool to compute so-called adiabatic vacua, *i.e.*, states that change the least under the time evolution of time-dependent Hamiltonians. They play an important role in quantum field theory of curved space-time and cosmology, but also in the context of the so-called Lewis-Riesenfeld invariants [77]. The traditional approach relies on WKB approximations and works for well for translationally invariant field theories, but treating more complicated systems is difficult, when the time dependent Hamiltonian does not split over individual (momentum) degrees of freedom. The formal power series presented in this manuscript reduces the problem to solving sets of algebraic equations iteratively, whose applications to concrete models in cosmology we will present elsewhere.

Another interesting avenue for the presented formalism would be to extend it to discrete phase spaces and stabilizer states. Quantum degrees of freedom are often classified as bosonic, fermionic or as being spin. For the former two, we have the important classes of Gaussian states, which we can characterize in the unified framework based on Kähler structures presented in this manuscript. On the other hand, spin degrees of freedom with $d$ levels are also known as qudits (generalization of qubit) and there is the well-known class of so called stabilizer states [78]. They are characterized by their eigenvalues with respect to certain spin operators (Pauli matrices) and play an important role in the context of quantum computation. Over the last few years, there has been substantial evidence that stabilizer states are the analogues of Gaussian states for spin system [79, 80], but this connection has not been made mathematically precise. What is well understood is that there is a discrete phase space formulation for qudits, which largely resembles the case of bosonic Gaussian states. In particular, there is a discrete analogue of the symplectic form, which for bosons governs the commutation relations. This is peculiar as spins are neither bosonic nor fermionic. It would thus be interesting to explore if there is an equivalent fermionic phase space formulation for spins, which resembles the case of fermionic Gaussian states (positive definite form instead of symplectic form). This leads to the natural question: Can we extend our unifying framework of Kähler structures to spin systems, where the analogous structure may be suitable to parametrize stabilizer states efficiently? Finding new results in this direction will be challenging, but if successful it may be directly relevant for algorithms and error correction in quantum computing.

# Acknowledgements

We thank Ivan Agullo, Abhay Ashtekar, Ignacio Cirac, Peter Drummond, Jens Eisert, Tommaso Guaita, Alexander Jahn, Tao Shi, Nelson Yokomizo and Bennet Windt for inspiring discussions.

LH acknowledges support by VILLUM FONDEN via the QMATH center of excellence (grant No. 10059). EB acknowledges support by the NSF via the Grant PHY-1806428.

# A   Conventions and notation

In this appendix, we review the conventions and notation used in this manuscript. The goal of our formalism is to be largely self-explanatory with an easy conversion between abstract objects (vectors, tensors, operators) and their numerical representation (lists, matrices, arrays). Note that this appendix largely resembles the one in [24].

## A.1   Abstract index notation

Throughout this paper, all equations containing indices follow the conventions of so-called *abstract index notation*. This formalism is commonly used in the research field of general relativity and gravity, where differential geometry plays an important role, but we believe that it is also of great benefit in the context of Kähler structures on the classical phase space used for Gaussian states.

The formalism is suitable to conveniently keep track of tensors built on a vector space. Given a finite dimensional real vector space $V$ with dual $V^*$, a $(r,s)$-tensor $T$ is a linear map

$$T : V^* \times \cdots \times V^* \times V \times \cdots \times V \to \mathbb{R} \,. \tag{269}$$

In particular, a $(1,0)$-tensor is a vector, a $(0,1)$-tensor is a dual vector and a $(1,1)$-tensor is a linear map. Moreover, a symplectic form, a metric (*i.e.*, an inner product) and their respective duals are $(2,0)$ or $(0,2)$ tensors.[20] To keep track of the type of tensor, abstract index notation refers to the $(r,s)$-tensor $T$ as $T^{a_1 \cdots a_r}{}_{b_1 \cdots b_s}$, *i.e.*, we assign $r$ upper indices and $s$ lower indices. Typically, one chooses the indices from some alphabet to indicate which vector space, we are referring to. For the classical phase space $V$ and its dual $V^*$, we use consistently Latin letters $a, b, c$.

The key advantage of abstract index notation in the context of the classical phasespace is that it helps us to keep track of what types of tensors, we are dealing with and which contractions are allowed. Apart from vectors $X^a$ and dual vectors $w_a$, we are mostly dealing with tensors that have two indices, namely linear maps $J^a{}_b$, bilinear forms $G^{ab}$ (on $V^*$) and their inverses $g_{ab}$.

Tensors with two indices can be naturally represented as matrices, which is particularly useful for numerical evaluation. However, when representing a tensor as a matrix, the index position (up or down) is lost and so one needs to be carefully keep track of which indices can be contracted, *i.e.*, which type of matrices can be multiplied. This problem does not arise if there is a fixed metric $G$ (with inverse $g$), such that we can require all tensors to be represented in an orthonormal basis, such that $G \equiv g \equiv \mathbb{1}$. While this could be done for fermions (where $G$ is indeed fixed), it does not work for bosons where $G$ is a dynamical state-dependent object. Moreover, we believe that making $G$ and $\Omega$ explicit throughout the manuscript highlights their role in the defining the respective tensors.

For tensors with two indices, it is therefore convenient to use a shorthand notation, where adjacent tensors with suppressed indices are implied to be contracted, just as standard matrix multiplication works. Obviously, this means that only such expressions are allowed where the

---

[20] Note that symplectic form $\Omega$ and metric $G$ act on the dual phasespace $V^*$.

adjacent indices are given by one upper and one lower index. This notation is particularly useful for numerical implementation.

## A.2 Special tensors and tensor operations

In the following, we review common tensor operations and emphasize how they are defined in a basis-independent way, which is also independent of a natural identification between $V$ and $V^*$. This highlights that certain formulas involving matrix operations (such as computing determinants, traces, eigenvalues or transposes) are only well defined in certain cases, *e.g.*, if the respective matrix represents a linear map in some cases or bilinear form in other cases.

**Identity.** Every vector space $V$ comes with the canonical identity map $\delta^a{}_b$ satisfying $\delta^a{}_b X^a = X^a$. Note that the notation $\mathbb{1}^a{}_b$ would be consistent with our convention, but we stayed with the commonly used Kronecker delta. In contrast, there does not exist a canonical bilinear form. In particular, writing $\delta_{ab}$ or $\delta^{ab}$ only makes sense with respect to a specific basis choice and is thus not canonical, unless we choose a specific basis (such as orthonormal basis). In the latter case, such a choice is equivalent to choosing an additional object (such as a metric $G^{ab}$).

**Transformation rules.** An invertible linear map $M^a{}_b : V \to V$ of the vector space $V$ acts on a general $(r,s)$-tensor $T^{a_1 \cdots a_r}{}_{b_1 \cdots b_s}$ and transforms it to

$$M^{a_1}{}_{c_1} \cdots M^{a_r}{}_{c_r} T^{c_1 \cdots c_r}{}_{d_1 \cdots d_s} (M^{-1})^{d_1}{}_{b_1} \cdots (M^{-1})^{d_s}{}_{b_s}. \tag{270}$$

In particular, a vector $X^a$ transforms as $M^a{}_b X^b$, a dual vector $w_a$ as $w_b (M^{-1})^b{}_a$, a dual bilinear form $S^{ab}$ as $M^a{}_c B^{cd} (M^\intercal)_d{}^b$, a bilinear form $s_{ab}$ as $(M^{-1\intercal})_a{}^c b_{cd} (M^{-1})^d{}_b$ and a linear map $K^a{}_b$ as $M^a{}_c K^d{}_d (M^{-1})^d{}_b$. The transformation rule can be understood as active transformation, where we ask what tensor $T$ one would get if we applied $M^\intercal$ on $V^*$ and $M^{-1}$ on $V$ for all input covectors and vectors. In practice, one gets the same formula if one has already expressed $T$ as an array of numbers with respect to fixed basis of $V$ (and a dual basis of $V^*$) and asks how do the numbers of $T$ change, if we applied $M^{-1}$ on the basis vector of $V$ and $(M^{-1})^\intercal$ on the elements of the dual basis.

**Determinant.** The determinant $\det(M)$ is only well-defined for a linear map $M^a{}_b$. The determinant of a bilinear form $s_{ab}$ or $S^{ab}$ is ill defined, unless we have a reference object, such as a metric $g_{ab}$ or $G^{ab}$. Then, we can compute the determinant of the matrix of the linear maps $S^{ac} g_{cb}$ or $G^{ac} s_{cb}$.

**Trace.** The trace $\mathrm{tr}(M) = M^a{}_a$ is defined for a linear map. There is no basis-independent definition of a trace for bilinear forms $S^{ab}$ or $s_{ab}$, unless we again have a reference object, such as a metric $G^{ab}$ and its dual $g_{ab}$, which would effectively allow us to convert the bilinear forms into linear maps $S^{ac} g_{cb}$ and $G^{ac} s_{cb}$, for which we can compute the regular trace.

**Eigenvalues.** Without additional structures, we can only define eigenvalues for a linear map $M^a{}_b$, where an eigenvalue $\lambda$ associated to an eigenvector $X^a$ satisfies

$$M^a{}_b X^b = \lambda X^b. \tag{271}$$

This is well-known from linear algebra. A bilinear form $X^{ab}$ does not have intrinsic eigenvalues, but we can compute its eigenvalues relative to another bilinear form. Given a bilinear form $s_{ab}$ and a metric $G^{ab}$ or symplectic form $\Omega^{ab}$, we can define the metric or symplectic eigenvalues as the regular eigenvalues of the linear map $G^{ac} s_{cb}$ or $\Omega^{ac} s_{cb}$, respectively.

**Functions.** Given a scalar function $f$ and a linear map $M^a{}_b$, we can define the matrix function $f(M)$ in the following ways: First, if $M$ is diagonalizable, we can apply $f$ to each

eigenvalue individually. This may require $f$ to be defined on the whole complex plane to make also send for complex eigenvalues. Second, when we consider analytical functions $f$, we can also define $f(M)$ by its power series, so that $f(M)$ is even defined for non-diagonalizable $M$. Note that there is no canonical way to apply a function $f$ to a bilinear form, such as $G^{ab}$ or $\Omega^{ab}$, unless we provide a procedure to first convert the form into a linear map (*e.g.*, by contraction with an inverse bilinear form), then apply the function in the previously described way and then convert back to a bilinear form.

**Transpose.** The transpose of a linear map $M^a{}_b : V \to V$ is the dual map $(M^{\intercal})_a{}^b : V^* \to V^*$. We use the notation $(M^{\intercal})_a{}^b = M^b{}_a$, which means that we just swap the positions of the two indices, while they are the same $(1,1)$-tensor, if we do not distinguish the ordering. For our shorthand notation, it is important to keep track of the order of indices, as this is how we contract. From the perspective of abstract index notation, transpose operation is often ignored (as one sometimes would even just write $T^a_b$ without any chosen ordering), but we intentionally choose an ordering ($M^a{}_b$ vs. $(M^{\intercal})_b{}^a$), so that they can be easily converted to matrix expressions, *e.g.*, for numerical implementations. Note that the shorthand notation $\mathbb{1}^{\intercal}$ then corresponds to the Kronecker delta $\delta_a{}^b$. If one has a metric $G$ on our vector space $V$ and a linear map $M^a{}_b$, many authors refer to the linear map $G^{ac}(M^{\intercal})_c{}^d g_{db}$ as the transpose of $M^a{}_b$, whose matrix representation coincides with $(M^{\intercal})_a{}^b$ in an orthonormal basis (where $G \equiv g \equiv \mathbb{1}$). As we do not always have such a fixed reference metric, we intentionally do not use the term transpose in that way.

## A.3 Common formulas

Given a triangle of compatible Kähler structures $(G, \Omega, J)$ with inverses $(g, \omega, -J)$, we have the following relations (in abstract index and shorthand notation):

$$-J^2 = \mathbb{1} \qquad \Longleftrightarrow \qquad -J^a{}_c J^c{}_b = \delta^a{}_b \,, \tag{272}$$

$$-(J^\mathsf{T})^2 = \mathbb{1}^\mathsf{T} \qquad \Longleftrightarrow \qquad -(J^\mathsf{T})_a{}^c (J^\mathsf{T})_c{}^b = \delta_a{}^b \,, \tag{273}$$

$$-J^{-1} = J \qquad \Longleftrightarrow \qquad -(J^{-1})^a{}_b = J^a{}_b \,, \tag{274}$$

$$J\Omega J^\mathsf{T} = \Omega \qquad \Longleftrightarrow \qquad J^a{}_c \Omega^{cd} (J^\mathsf{T})_d{}^b = \Omega^{ab} \,, \tag{275}$$

$$-\Omega J^\mathsf{T} = J\Omega \qquad \Longleftrightarrow \qquad -\Omega^{ac}(J^\mathsf{T})_c{}^b = J^a{}_c \Omega^{cb} \,, \tag{276}$$

$$JGJ^\mathsf{T} = G \qquad \Longleftrightarrow \qquad J^a{}_c G^{cd}(J^\mathsf{T})_d{}^b = G^{ab} \,, \tag{277}$$

$$-GJ^\mathsf{T} = JG \qquad \Longleftrightarrow \qquad -G^{ac}(J^\mathsf{T})_c{}^b = J^a{}_c G^{cb} \,, \tag{278}$$

$$\Omega J^\mathsf{T} = G \qquad \Longleftrightarrow \qquad \Omega^{ac}(J^\mathsf{T})_c{}^b = G^{ab} \,, \tag{279}$$

$$-J\Omega = G \qquad \Longleftrightarrow \qquad -J^a{}_c \Omega^{cb} = G^{ab} \,, \tag{280}$$

$$\Omega\omega = \mathbb{1} \qquad \Longleftrightarrow \qquad \Omega^{ac}\omega_{cb} = \delta^a{}_b \,, \tag{281}$$

$$\omega\Omega = \mathbb{1}^\mathsf{T} \qquad \Longleftrightarrow \qquad \omega_{ac}\Omega^{cb} = \delta_a{}^b \,, \tag{282}$$

$$Gg = \mathbb{1} \qquad \Longleftrightarrow \qquad G^{ac}g_{cb} = \delta^a{}_b \,, \tag{283}$$

$$gG = \mathbb{1}^\mathsf{T} \qquad \Longleftrightarrow \qquad g_{ac}G^{cb} = \delta_a{}^b \,, \tag{284}$$

$$-\omega G\omega = g \qquad \Longleftrightarrow \qquad -\omega_{ac}G^{cd}\Omega_{db} = g_{ab} \,, \tag{285}$$

$$-g\Omega g = \omega \qquad \Longleftrightarrow \qquad -g_{ac}\Omega^{cd}G_{db} = \omega_{ab} \,, \tag{286}$$

$$\Omega g = J \qquad \Longleftrightarrow \qquad \Omega^{ac}g_{cb} = J^a{}_b \,, \tag{287}$$

$$-G\omega = J \qquad \Longleftrightarrow \qquad -G^{ac}\omega_{cb} = J^a{}_b \,, \tag{288}$$

$$-g\Omega = J^\mathsf{T} \qquad \Longleftrightarrow \qquad g_{ac}\Omega^{cb} = (J^\mathsf{T})_a{}^b \,, \tag{289}$$

$$\omega G = J^\mathsf{T} \qquad \Longleftrightarrow \qquad \omega_{ac}G^{cb} = (J^\mathsf{T})_a{}^b \,, \tag{290}$$

$$-\Omega^\mathsf{T} = \Omega \qquad \Longleftrightarrow \qquad -\Omega^{ba} = \Omega^{ab} \,, \tag{291}$$

$$G^\mathsf{T} = G \qquad \Longleftrightarrow \qquad G^{ba} = G^{ab} \,. \tag{292}$$

A symplectic group element $M^a{}_b \in \mathrm{Sp}(2N, \mathbb{R})$ and a symplectic algebra element $K^a{}_b \in \mathfrak{sp}(2N, \mathbb{R})$ are characterized by the following properties:

$$M\Omega M^\mathsf{T} = \Omega \qquad \Longleftrightarrow \qquad M^a{}_c \Omega^{cd}(M^\mathsf{T})_d{}^b = \Omega^{ab} \,, \tag{293}$$

$$\Omega M^\mathsf{T}\omega = M^{-1} \qquad \Longleftrightarrow \qquad \Omega^{ac}(M^\mathsf{T})_c{}^d \omega_{db} = (M^{-1})^a{}_b \,, \tag{294}$$

$$-\Omega K^\mathsf{T} = K\Omega \qquad \Longleftrightarrow \qquad -\Omega^{ac}(K^\mathsf{T})_c{}^b = K^a{}_c \Omega^b \,. \tag{295}$$

An orthogonal group element $M^a{}_b \in \mathrm{O}(2N)$ and an orthogonal algebra element $K^a{}_b \in \mathfrak{so}(2N)$ are characterized by the following properties:

$$MGM^\mathsf{T} = G \qquad \Longleftrightarrow \qquad M^a{}_c G^{cd}(M^\mathsf{T})_d{}^b = G^{ab} \,, \tag{296}$$

$$GM^\mathsf{T}g = M^{-1} \qquad \Longleftrightarrow \qquad G^{ac}(M^\mathsf{T})_c{}^d g_{db} = (M^{-1})^a{}_b \,, \tag{297}$$

$$-GK^\mathsf{T} = KG \qquad \Longleftrightarrow \qquad -G^{ac}(K^\mathsf{T})_c{}^b = K^a{}_c G^b \,. \tag{298}$$

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
