# Peer review of "Bosonic and fermionic Gaussian states from Kähler structures"

_SciPost Physics, doi:SciPost Phys. Core 4, 025 (2021)_

## Round 2 · Referee Report · Jan Dereziński (Referee 1) · 2021-2-10

• Download as PDF

Report

See report
  • validity: -
  • significance: -
  • originality: -
  • clarity: -
  • formatting: -
  • grammar: -

Author:  Lucas Hackl  on 2021-05-20  [id 1441]

(in reply to Report 1 by Jan Dereziński on 2021-02-10)

We thank the referee for the careful review of our manuscript and for the remark that our manuscript can be useful to many readers.

We understand that the main criticism of the referee refers to our claim of novelty in regards to the presented formalism and the use of Kähler structures in particular. While we are fully aware that Kähler structures have always played an important role in the mathematical machinery of quantization and its treatment in the mathematical physics literature, which we also cite in our introduction [1,2,3], we agree with the referee that we should have made clearer where some of our formulas already appeared in the mathematical physics literature (though often in very different notation). In particular, we fully agree that the treatment of quasi-free states in the excellent reference [2] heavily uses Kähler structures.

On the other hand, we believe that our manuscript is more than a mere review of existing methods scattered in the literature, but rather an application of existing mathematical methods with the goal of establishing a formalism for the unified treatment of Gaussian states that can be readily used by physicists from fields, such as quantum information, condensed matter, statistical physics, quantum gravity and high energy theory. While reference [2] presents a mathematical treatment of quantization to prepare mathematical physicists to work on quantum field theory, we entirely focus on Gaussian states and how the Kähler structures can be easily applied for practical calculations in quantum information, condensed matter and statistical physics. Finally, we also introduce several new directions (such as adiabatic vacua), which have not been considered before and for which Kähler structures offer a natural starting point.

In response to the referee's suggestions, we carefully compared our presentation with the ones contained in the mathematical physics literature [1-3] and included clear references to the respective sections to emphasize where parts of our formalism appeared in similar or equivalent form in the literature. Here, we thank the referee for pointing out several relations to reference [2], which we fully agree with. Furthermore, we also implemented all other corrections and suggestions regarding our manuscript. We have also restructured the manuscript to better highlight the new applications (Sec.IV-A,B,C) and separate them from the discussion of the relevant mathematical structures.

We are attaching a revised version with individual changes marked in red.

[1] M. de Gosson, "Symplectic Geometry and Quantum Mechanics"
[2] J. Derezinski and C. Gerard, "Mathematics of Quantization and Quantum Fields"
[3] P. Woit, "Quantum Theory, Groups and Representations"

Attachment:

PAPER__Bosonic_and_Fermionic_Gaussian_states_from_K_hler_structures.pdf

Anonymous on 2021-06-08  [id 1493]

(in reply to Lucas Hackl on 2021-05-20 [id 1441])
Category:
correction

We thank the referee for emphasizing this point, with which we agree. We do not claim novelty of the mathematical formalism (that was certainly known in the mathematical physics literature for Gaussian states) and agree with the referee that the listed points in our summary are rather key advantages that make the formalism relevant and useful also in areas outside of mathematical physics. Consequently, we see as main contribution of our manuscript (a) the attempt to make the relevant mathematical methods accessible to physicist in other fields and (b) to demonstrate their versatility for a number of applications with a particular emphasis on quantum information and out-of-equilibrium physics. We therefore rephrased some parts of our introduction and summary to emphasize this perspective based on the referee's criticism. We hope that the referee agrees with these changes (see attachment, changes in red). Apart from that we thank the referee for the interest in our treatment of adiabatic states which we intend to expand on in later works. As suggested also by the referee, we hope that our manuscript will be useful for a broad audience of physicists and thereby contributes to linking mathematical physics, quantum information and out-of-equilibrium quantum theory.

Attachment:

PAPER__Bosonic_and_Fermionic_Gaussian_states_from_K_hler_structures.pdf

Anonymous on 2021-05-30  [id 1479]

(in reply to Lucas Hackl on 2021-05-20 [id 1441])
Category:
remark

I am still convinced that the manuscript has limited novelty. The description of its novelty on pages 35-36 is exaggerated. "Gaussian states from K"ahler stuctures"--this is well known in the literature (Araki, etc., and discussed at length in Derezinski-Gerard). "Phase space covariance"--again, this is well-known. In particular, the treatment of Gaussian states Derezinski-Gerard is coordinate free . These points are not "novelty features", but rather "advantages" of this review.
The description of adiabatic states on pages 34-35 is a valuable part of the manuscript. This is a useful presentation in abstract terms of some ideas found mostly in the literature about curved spacetimes. Actually, I wish this part of the manuscript was more elaborated.
In any case, the text is in general well structured and it should be useful to many readers.

---

## Round 2 · Referee Report · Anonymous (Referee 2) · 2021-2-16

Strengths

  1. The authors work out in detail the general case of canonical quantized systems that can be exactly solved: arbitrary quadratic hamiltonians and complex structures, for arbitrary bosonic and fermionic degrees of freedom. This provides a valuable reference and starting point for a wide array of research directions, some of which are described in the final section.

  2. The approach to the material brings together a structural, coordinate-invariant view, together with detailed formulas in coordinates, making clear how things depend on coordinate choices. Other references typically are abstract, leaving it unclear how to do calculations, or just a collection of complicated formulas, leaving their structure unclear.

  3. The approach is exhaustive, working out everything in detail, providing formulas useful in a wide variety of contexts, not available elsewhere.

  4. The writing is unusually clear, easy to follow and very good at providing insight into what otherwise can be an obscure and difficult to follow formalism.

Weaknesses

  1. Most of what the authors are doing follows from basic ideas that have been known to experts for a long time. Much of what they write down that is not available elsewhere could be worked out with a moderate amount of effort by a researcher who needed the results.

Report

I recommend publication, this is a quite valuable document. It meets the journal criteria by providing a basis for future research, linking a wide variety of different research areas.

Requested changes

No changes needed.

  • validity: top
  • significance: high
  • originality: high
  • clarity: top
  • formatting: perfect
  • grammar: perfect

Author:  Lucas Hackl  on 2021-05-20  [id 1442]

(in reply to Report 2 on 2021-02-16)

We thank the referee for the careful review of our manuscript. We agree that many of the aspects of the presented formalism have been known by experts (particularly, in the mathematical physics community) for a long time and we revised our manuscript in such a way to emphasize such connections to existing literature further. At the same time, we equally believe as the referee does that our review will be of value for physicists working in different fields (quantum information, condensed matter, quantum gravity, statistical mechanics etc.) who are less familiar with the mathematical machinery of Kähler structures and are mostly interested in applying the presented unified framework to concrete problems in physics, for which we provided several compact formula tables.

We are attaching a revised version with individual changes marked in red.

Attachment:

PAPER__Bosonic_and_Fermionic_Gaussian_states_from_K_hler_str_m6qIfpb.pdf

---

## Round 3 · List of Changes

As part of our reply to the referee reports, we included a revised manuscript with changes marked in red.

---

## Editorial Decision

published